# Large-scale plasma proteomics comparisons through genetics and disease associations

Grimur Hjorleifsson Eldjarn[1,7], Egil Ferkingstad[1,7], Sigrun H. Lund[1,2], Hannes Helgason[1,2], Olafur Th. Magnusson[1], Kristbjorg Gunnarsdottir[1], Thorunn A. Olafsdottir[1], Bjarni V. Halldorsson[1,3], Pall I. Olason[1], Florian Zink[1], Sigurjon A. Gudjonsson[1], Gardar Sveinbjornsson[1], Magnus I. Magnusson[1], Agnar Helgason[1,4], Asmundur Oddsson[1], Gisli H. Halldorsson[1], Magnus K. Magnusson[1,5], Saedis Saevarsdottir[1,5], Thjodbjorg Eiriksdottir[1], Gisli Masson[1], Hreinn Stefansson[1], Ingileif Jonsdottir[1,5], Hilma Holm[1], Thorunn Rafnar[1], Pall Melsted[1,2], Jona Saemundsdottir[1], Gudmundur L. Norddahl[1], Gudmar Thorleifsson[1], Magnus O. Ulfarsson[1,6], Daniel F. Gudbjartsson[1,2], Unnur Thorsteinsdottir[1,5], Patrick Sulem[1✉] & Kari Stefansson[1,5✉]

High-throughput proteomics platforms measuring thousands of proteins in plasma combined with genomic and phenotypic information have the power to bridge the gap between the genome and diseases. Here we performed association studies of Olink Explore 3072 data generated by the UK Biobank Pharma Proteomics Project[1] on plasma samples from more than 50,000 UK Biobank participants with phenotypic and genotypic data, stratifying on British or Irish, African and South Asian ancestries. We compared the results with those of a SomaScan v4 study on plasma from 36,000 Icelandic people[2], for 1,514 of whom Olink data were also available. We found modest correlation between the two platforms. Although *cis* protein quantitative trait loci were detected for a similar absolute number of assays on the two platforms (2,101 on Olink versus 2,120 on SomaScan), the proportion of assays with such supporting evidence for assay performance was higher on the Olink platform (72% versus 43%). A considerable number of proteins had genomic associations that differed between the platforms. We provide examples where differences between platforms may influence conclusions drawn from the integration of protein levels with the study of diseases. We demonstrate how leveraging the diverse ancestries of participants in the UK Biobank helps to detect novel associations and refine genomic location. Our results show the value of the information provided by the two most commonly used high-throughput proteomics platforms and demonstrate the differences between them that at times provides useful complementarity.

The development of high-throughput proteomics platforms by Soma-Logic and Olink and their integration with genomic data has increased the depth of our understanding of the relationships between sequence variants and diseases and other traits[2–6]. This has uncovered genomic sequence variants associated with plasma protein levels (protein quantitative trait loci (pQTLs)) and biomarkers of diseases and their progression. Associations between protein levels and diseases are rarely sufficient to separate cause from effect. However, associations of pQTLs with diseases can be used for causal inference[7,8].

SomaLogic and Olink are affinity-based platforms that use binding to target proteins for measurement. Individual SomaScan assays use a single aptamer to measure the target protein. In our previous study of 35,559 Icelanders, we quantified 4,719 proteins with 4,907 SomaScan assays and performed association analyses with rich health-related and genotype information[2]. Olink is based on immunoassays that

require the binding of two distinct antibodies. Studies using Olink have either measured hundreds of proteins in tens of thousands of people[3] or thousands of proteins in about a thousand individuals[7,9]. Whether one platform or the other should be preferred in certain research settings is not well understood.

Previous studies comparing various versions of the Olink and SomaScan platforms for the analysis of plasma samples have either been limited by sample size or the small number of proteins investigated[7,10,11]. This has resulted in limited power for genetic analysis. Efforts to replicate pQTLs between platforms have been moderately successful, with between 75% and 93% of *cis* pQTLs (the pQTL is close to the gene encoding the protein) and between 52% and 64% of *trans* pQTLs (not *cis*) replicated[2,3,6,7].

Here we compare two affinity-based platforms, Olink Explore 3072 using data from the UK Biobank (UKB) generated by the UKB Pharma

[1]deCODE Genetics/Amgen, Reykjavik, Iceland. [2]School of Engineering and Natural Sciences, University of Iceland, Reykjavik, Iceland. [3]School of Technology, Reykjavik University, Reykjavik, Iceland. [4]Department of Anthropology, University of Iceland, Reykjavik, Iceland. [5]Faculty of Medicine, School of Health Sciences, University of Iceland, Reykjavik, Iceland. [6]Faculty of Electrical and Computer Engineering, University of Iceland, Reykjavik, Iceland. [7]These authors contributed equally: Grimur Hjorleifsson Eldjarn, Egil Ferkingstad. ✉e-mail: patrick.sulem@decode.is; kstefans@decode.is

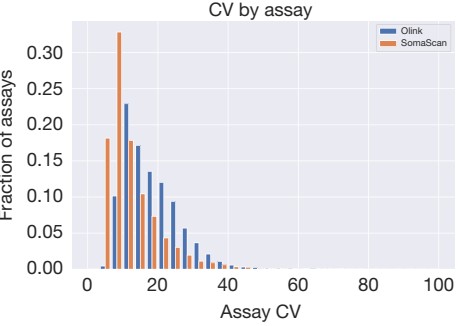

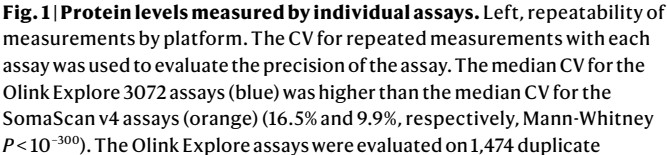

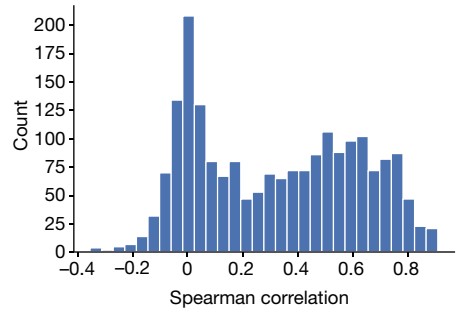

**Fig. 1 | Protein levels measured by individual assays.** Left, repeatability of measurements by platform. The CV for repeated measurements with each assay was used to evaluate the precision of the assay. The median CV for the Olink Explore 3072 assays (blue) was higher than the median CV for the SomaScan v4 assays (orange) (16.5% and 9.9%, respectively, Mann-Whitney $P < 10^{-300}$). The Olink Explore assays were evaluated on 1,474 duplicate measurements from the UKB 47K dataset, whereas the SomaScan v4 assays were evaluated on 227 duplicate measurements from the Iceland 36K dataset. Right, correlation between measurements for protein levels measured using assays on the Olink Explore 3072 and SomaScan v4 platforms in the Iceland 1K dataset (Spearman correlation), evaluated by measuring plasma samples from 1,514 individuals using both platforms.

Proteomics Project (UKB-PPP) and SomaScan v4 generated in Iceland[2] (Extended Data Fig. 1). We analysed the UKB dataset stratified on ancestry into 46,218 individuals with British or Irish ancestry (UKB-BI), 953 individuals with South Asian ancestries (UKB-SA) and 1,513 individuals with African ancestries[12] (UKB-AF). We analysed SomaScan v4 data for plasma samples from 35,892 Icelanders, where 1,514 of the samples were also measured using the Olink Explore 3072 platform. In addition to a direct comparison of measurements, we used the available UKB and Icelandic genotype and phenotype data to compare their associations with protein levels on the platforms. We demonstrate how the selection of the platform can substantially affect the results and the conclusions drawn in the context of the study of a disease.

## Comparison of precision

We calculated the CV for the assays using duplicate measurements of 1,474 samples in the UKB dataset (Olink Explore 3072) and 227 samples in the Icelandic dataset (SomaScan v4) (Fig. 1, Supplementary Tables 1 and 2). On the basis of all assays, the median CV for Olink was higher than for SomaScan (16.5% and 9.9%, respectively; Mann−Whitney $P < 10^{-300}$) (Supplementary Table 3). Restricting to assays that target the 1,823 proteins targeted on both platforms, the median CV remained higher for Olink than SomaScan (14.7% and 9.5%, respectively; Mann−Whitney $P = 6.5 \times 10^{-150}$), consistent with previous reports of SomaScan assays having lower CV on control samples than Olink assays[10,11,13]. Computing the CV of the assays using random pairs of samples ($n = 1,000$) gives a higher median CV for Olink assays than SomaScan assays (47.4% and 32.2%, respectively, Mann-Whitney $P = 1.4 \times 10^{-170}$, Supplementary Fig. 1). While this may be at least in part due to population differences, using only the Icelandic samples measured using the Olink platform and considering only the assays targeting proteins targeted on both platforms still gives a higher median CV for the Olink assays than the SomaScan assays (62.5% and 34.0%, respectively, Mann-Whitney $P = 1.4 \times 10^{-209}$) (Supplementary Notes 1–3 and Supplementary Table 4).

## Inter-platform correlation of levels

In most studies performed using SomaScan, the last step in the normalization process adjusts the median protein levels for each individual to a reference (referred to as SMP normalization), as recommended by the manufacturer, whereas some studies omit this step[2,7,10]. The results presented in this Article are based on the median-adjusted data; the unadjusted results are presented and the effect of the adjustment are discussed in Supplementary Note 4.

In the Icelandic set of 1,514 individuals with data from both platforms, the median Spearman correlation between plasma levels of 1,848 proteins measured with matching Olink and SomaScan assays was 0.33 (Fig. 1, Supplementary Tables 5 and 6, Supplementary Note 5 and Supplementary Fig. 2), consistent with previous reports based on up to around 900 proteins[7,10,11]. The distribution of the correlation coefficients has two modes, one just above 0 and the other just below 0.6. Omitting the SMP normalization step on SomaScan, we observed a higher median correlation of 0.39 between platforms. The Olink Explore 3072 consists of two assay sets: an earlier version of the platform labelled 1536, and an additional follow-up set of assays labelled Expansion. We note a substantially higher median correlation for assays in the 1536 set than in the Expansion set (0.36 and 0.27, respectively). For comparison, the median pairwise correlation between assays from the two platforms for all possible assay pairs was −0.01, and the median within-platform correlation between assays was 0.08 for the Olink platform and −0.01 for the SomaScan platform (0.42 if omitting SMP normalization). If either assay of a matching pair had a low variation in levels, the correlation between the levels of proteins measured with the matching assays tended to be low (Supplementary Notes 4−6 and Supplementary Fig. 3).

The overall variance of the levels of a protein was more concordant between matching assays on the two platforms where *cis* pQTLs were detected on both platforms (Spearman correlation 0.57 for pQTLs on both platforms, otherwise below 0.28) but did not show a clear trend for individual platforms depending on the presence or absence of *cis* pQTLs on that platform (Extended Data Fig. 2).

## Dilution and subcellular location

Both Olink and SomaScan use dilution groups for their assays based on the protein abundance in plasma, with the most abundant proteins belonging to the highest dilution groups (Supplementary Table 7). For both platforms, the correlation between protein levels was lowest in the lowest dilution group (Supplementary Table 8). Correspondingly, for the matching assays, the median CV was higher in the lowest dilution group than in the other dilution groups, although on the SomaScan platform, the CV was again higher in the highest dilution group than the intermediate group (Supplementary Table 9). A substantial number of Olink assays have levels below the limit of detection (LOD) for a large fraction of individuals, especially for proteins in the undiluted group−that is, with expected lowest abundance in plasma. The fraction of values below LOD correlated positively with CV (Spearman correlation 0.69, $P < 10^{-300}$; Supplementary Fig. 4).

According to the Human Protein Atlas, 63% of the 19,187 human proteins are intracellular, 28% are membrane proteins and the remaining 9% are secreted[14]. SomaScan v4 and Olink Explore 3072 were similarly depleted of intracellular proteins (49% and 48%, respectively) and enriched with secreted proteins (21% and 24%, respectively) ($P < 5.7 \times 10^{-158}$; Supplementary Table 10). The protein abundance as reflected by the dilution groups is lower on both platforms for assays targeting intracellular proteins.

The abundance of proteins as reflected by the dilution group is associated with the precision of measurements (Supplementary Tables 9 and 11 and Supplementary Fig. 5). Previous reports have described the effect of decreased matrix complexity on measurement, which along with the abundance may affect the precision[15].

We noted a wide range of median correlation between levels when stratifying on tissue based on enriched expression ($r$ between 0.05 and 0.64). The tissue with the lowest median correlation was the pituitary gland, followed by testis, fallopian tube, retina, skin and brain (all with $r \le 0.15$). The tissue with the highest median correlation of levels was gallbladder, followed by smooth muscle, cervix, endometrium, pancreas and salivary gland (all with $r > 0.45$; Supplementary Table 12).

## Protein–phenotype associations

We accounted for multiple testing of protein–phenotype associations using Bonferroni adjustment for the number of assays on each platform ($P < 1.0 \times 10^{-5}$ for 4,907 assays on SomaScan and $P < 1.7 \times 10^{-5}$ for 2,941 assays on Olink). Plasma protein levels have previously been shown to correlate with sample age[16]. Sample age—that is, time from blood draw to measurement—correlated significantly with levels of a considerable number of proteins on both platforms, although differences in the distribution of sample age between cohorts make direct comparison difficult (Supplementary Tables 13–15). The effects of sample age were generally small for both platforms and the correlation between effects was low (Spearman $r = 0.08$). The levels of 77% of SomaScan proteins were associated with participant age at sample collection, 64% were associated with sex and 69% were associated with body mass index (BMI). In UKB-BI, the levels of 60%, 68% and 78% of Olink proteins were associated with participant age at sample collection, sex and BMI, respectively, and the effects correlated well between ancestries in UKB (pairwise Spearman correlation greater than 0.70, 0.71 and 0.87 for participant age, sex and BMI, respectively; Extended Data Fig. 3 and Supplementary Tables 14 and 15). The Spearman correlation of participant age and sex effects between the 2,021 pairs of matching assays for proteins targeted by both Olink and SomaScan was 0.52 for participant age, 0.56 for sex and 0.43 for BMI.

We tested for associations between protein levels and 389 binary phenotypes and 208 quantitative traits in UKB, and 275 binary phenotypes and 110 quantitative traits in Iceland. In UKB, we found 303,261, 13,047 and 10,850 associations among individuals with British or Irish, South Asian and African ancestries, respectively (Supplementary Table 16). In Iceland, we found 218,503 associations (Supplementary Table 17).

As examples, we compared the associations between levels of proteins measured using the two platforms with heart failure, Alzheimer's disease and inflammatory bowel disease (IBD) (comprising Crohn's disease and ulcerative colitis). For heart failure ($n = 1,369$ cases in Iceland and $n = 676$ cases in UKB), natriuretic peptide B (BNP) (encoded by *NPPB*) had the most significant association on both platforms (UKB-BI with Olink: odds ratio (OR) = 2.25 per s.d., $P = 1.6 \times 10^{-177}$; Iceland with SomaScan: OR = 1.90 per s.d., $p = 2.0 \times 10^{-163}$), consistent with the established correlation of circulating BNP levels with heart failure[17], whereas for some proteins the results were discordant (Supplementary Note 7).

In the case of Alzheimer's disease ($n = 389$ cases in Iceland and $n = 224$ cases in UKB-BI), neurofilament light (NFL) (encoded by *NEFL*) was

targeted on both platforms and strongly associated with disease on both (rank 2 of all assays on Olink and rank 3 of all assays on SomaScan based on effect and significance), but with opposite direction (UKB-BI Olink: OR = 1.64 per s.d., $P = 1.3 \times 10^{-12}$; SomaScan OR = 0.53 per s.d., $P = 9.7 \times 10^{-52}$). The Spearman correlation between levels of NFL on the two platforms was low ($r = 0.06$) and *cis* pQTLs for the protein were not detected on either platform. Many proteoforms can be derived from a single gene, for example through differential splicing of transcripts, proteolytic cleavage and post-translational modification[18]. NFL forms polymers[19], and we do not have information about which proteoforms are measured on the two platforms. Consistent with the results for NFL on Olink, high NFL levels in cerebrospinal fluid and blood have been reported to associate with advanced Alzheimer's disease[20]. Plasma levels of NFL measured using the Olink platform and an alternative affinity-based assay (Simoa) have been reported to be strongly correlated[21,22] ($r > 0.90$ for both studies), whereas levels of NFL measured using the SomaScan platform did not correlate with Simoa measurements ($n = 231$ Icelanders, Spearman $r = 0.00$). Of note, NFL levels measured with both the Olink and SomaScan platforms correlate strongly with Alzheimer's disease, but with opposite directions of effect. Although Olink and Simoa appear to be measuring the same proteoform(s), it remains to be understood what protein or proteoform SomaScan is measuring.

For IBD ($n = 618$ cases in Iceland and $n = 900$ cases in UKB-BI), prostaglandin-H2 D-isomerase (encoded by *PTGDS*) was the most significantly associated protein with the disease on both Olink and SomaScan (Supplementary Table 18). On both platforms, IBD cases had higher plasma levels of prostaglandin-H2 D-isomerase than controls (UKB-BI with Olink: OR = 1.67 per s.d., $P = 4.7 \times 10^{-40}$; Iceland with SomaScan: OR = 1.36 per s.d., $P = 1.9 \times 10^{-13}$), consistent with the reported role of the PGD2 metabolic pathway in IBD supported by animal models[23,24]. The correlation of levels between platforms was 0.51 and *cis* pQTLs were observed for *PTGDS* on both platforms. Whereas the level of several proteins as measured on both platforms are significantly associated with IBD (for example, CXCL11 and REG3A), some associations did not replicate between the two groups (for example, interleukin-6 (IL-6), MMP12 and CLPS) (Supplementary Table 18). Of note, IBD was associated with CXCL9 levels as measured by the single Olink assay, but only one of two assays on SomaScan (Supplementary Table 18), again raising questions about what proteoform or protein is being targeted with the non-associating SomaScan assay.

## Detection of pQTLs

We updated our previous pQTL analysis in Iceland to include more sequence variants and applied the manufacturer-recommended data normalization[2] (Table 1, Extended Data Fig. 4, Supplementary Tables 19 and 20 and Supplementary Note 4). We identified 2,120 and 22,616 sentinel *cis* and *trans* pQTLs, respectively (Table 1).

Using the Olink-UKB data, we identified pQTLs stratified by ancestry, using the same approach as we previously applied to the Icelandic SomaScan data[2] ($P < 1.8 \times 10^{-9}$; Table 1, Extended Data Fig. 4, Supplementary Tables 21–23 and Supplementary Note 8). We identified 2,102, 900 and 714 sentinel *cis* pQTLs for the UKB-BI, UKB-AF and UKB-SA ancestry groups, respectively, and 24,824, 1,332 and 190 sentinel *trans* associations. Of the UKB-BI sentinel *cis* pQTL associations, 1,246 were from the initial set (1536) and 856 were from the Expansion set, whereas of the *trans* pQTL associations, 16,551 were from the 1536 set and 8,273 were from the Expansion set.

Based on Olink, 52% of sentinel *trans* pQTL associations are with a variant associating with more than 10 proteins (non-specific pQTL), whereas based on SomaScan, 63% of *trans* pQTL associations are with such a variant. Consequently, 211 out of all 2,616 proteins with at least one pQTL on Olink (8%) have associations with only non-specific

**Table 1 | Summary of pQTLs detected on the Olink Explore 3072 and SomaScan v4 platforms**

| | ALL TARGETS | | | OVERLAPPING TARGETS | | |
|---|---|---|---|---|---|---|
| Dataset | Iceland 36K non-normalized | Iceland 36K normalized | UKB-BI | Iceland 36K non-normalized | Iceland 36K normalized | UKB-BI |
| Platform | SomaScan v4 | SomaScan v4 | Olink Explore 3072 | SomaScan v4 | SomaScan v4 | Olink Explore 3072 |
| Number of assays | 4,907 | 4,907 | 2,931 | 1,954[a] | 1,954[a] | 1,832[a] |
| Number of individuals | 36,136 | 35,892 | 46,218 | 36,136 | 35,892 | 46,218 |
| Population | Iceland | Iceland | UKB | Iceland | Iceland | UKB |
| No. of assays with *cis* pQTLs (%) | 1,889 (38%) | 2,120 (43%) | 2,101 (72%) | 1,068 (55%) | 1,164 (60%) | 1,467 (80%) |
| No. of assays with *trans* pQTLs (%) | 4,437 (90%) | 4,716 (96%) | 2,528 (86%) | 1,782 (91%) | 1,889 (97%) | 1,658 (91%) |
| No. of assays with pQTLs (%) | 4,649 (95%) | 4,809 (98%) | 2,627 (90%) | 1,869 (96%) | 1,928 (99%) | 1,715 (94%) |
| No. of sentinel pQTL associations | 18,667 | 24,736 | 26,926 | 8,696 | 11,516 | 20,046 |
| No. of sentinel *cis* pQTL associations | 1,889 | 2,120 | 2,102 | 1,068 | 1,164 | 1,468 |
| No. of sentinel *trans* pQTL associations | 16,778 | 22,616 | 24,824 | 7,628 | 10,352 | 18,578 |
| No. of secondary pQTL associations | 10,564 | 14,786 | 14,232 | 6,005 | 7,889 | 11,172 |
| No. of secondary *cis* pQTL associations | 5,791 | 7,292 | 8,640 | 3,877 | 4,796 | 6,614 |
| No. of secondary *trans* pQTL associations | 4,773 | 7,494 | 5,592 | 2,128 | 3,093 | 4,558 |
| No. of sentinel *cis* pQTL associations with PAV in high LD ($r^2 > 0.8$) (%) | 636 (34%) | 710 (33%) | 696 (33%) | 371 (35%) | 408 (35%) | 476 (32%) |
| No. of sentinel *cis* pQTL associations with *cis* eQTL in high LD ($r^2 > 0.8$) (%) | 657 (35%) | 735 (35%) | 820 (39%) | 380 (36%) | 405 (35%) | 563 (38%) |
| No. of sentinel pQTL associations with MAF <0.1% | 478 (3%) | 816 (3%) | 617 (2%) | 273 (3%) | 433 (4%) | 516 (3%) |
| No. of sentinel pQTL associations with 0.1%< MAF <1% | 1,857 (10%) | 2,404 (10%) | 1,212 (5%) | 963 (11%) | 1.069 (9%) | 951 (5%) |

[a]1,823 proteins are targeted by assays on both platforms.
LD, linkage disequilibrium.

*trans* pQTLs and 1,282 out of 4,574 proteins with at least one pQTL on SomaScan (28%) had such associations only. When an assay has no *cis* pQTLs and has only non-specific *trans* pQTLs, it is possible that the targeting or measurement are not accurate. Non-specific pQTLs should be interpreted with caution.

In both the UKB-BI Olink data and Icelandic SomaScan data, more than 98% of the *cis* pQTLs associate with only one protein in *cis*. However, 34 pQTLs detected in the Olink data had multiple *cis* associations. Of these, 32 associated with fewer than 25 proteins in *cis* or *trans*—mostly 2 or 3— whereas 2 associated with more than 25 (768 and 388). On SomaScan, 34 pQTLs had multiple *cis* associations. Of these, 29 associated with less than 25 proteins in *cis* or *trans*, whereas 5 associated with more than 25 (ranging from 40 to 786) (Supplementary Tables 19–21). Co-regulation of expression in *cis* has been observed at RNA level and does not in itself detract from the *cis* pQTL as evidence for the performance of the assay[25]. However, the *cis* location of a highly pleiotropic variant could be a matter of chance and should therefore be considered in the same way as a *trans* pQTL.

The detection of *cis* pQTLs for a given protein associated positively with the fraction of assay measurements above the LOD ($P = 8.3 \times 10^{-164}$, median fraction above LOD 99.8%) as well as the fraction of assays where the median normalized protein expression (NPX) value was above the LOD (chi-squared $P = 2.7 \times 10^{-207}$), based on 24 healthy and 48 individuals with disease (information supplied by Olink) (Supplementary Fig. 6 and Supplementary Tables 11 and 24). However, we detected pQTLs for some assays with a very small fraction of values above the LOD, indicating that measurements below the LOD may still be informative.

Neither subsampling nor accounting for the different number of variants tested in the datasets affected the conclusions of the comparison between the platforms. In both datasets, we estimated a false discovery rate of 1.2% (Supplementary Note 9). Replication of pQTLs in the Icelandic dataset also assessed by Olink was somewhat lower than predicted by power analysis and may represent the effect of differences in population and sample handling and processing (Supplementary Note 10 and Supplementary Tables 13 and 25)

Secondary *cis* pQTL associations were detected for 1,702 out of 2,102 sentinel *cis* pQTLs on the Olink platform (81%) and for 1,594 out of 2,120 on the SomaScan platform (75%). Secondary *trans* pQTL associations were detected for 3,340 out of 24,824 sentinel *trans* pQTL associations on the Olink platform (13%) and 4,065 out of 22,616 on the SomaScan platform (18%).

Whereas secondary signals help to understand how genetic variation affects protein expression, the mere existence of a *cis* pQTL for a protein on a particular platform provides evidence that the assay is binding to the correct protein, even though the pQTL may in fact be the result of an epitope effect (that is, the genetic variant directly affects binding of the antibody to its epitope) and not reflect actual variation in protein levels. On the Olink platform, the majority of proteins already have a *cis* pQTL. Furthermore, the significance of most *cis* pQTLs is well above the genome-wide threshold, suggesting that the number of *cis* pQTLs is unlikely to change drastically with increased sample size, although *cis* pQTLs still provide valuable insights into the genetic control of protein expression. However, as most *trans* pQTL associations have significance close to the genome-wide threshold, expanding the sample size is likely to reveal more *trans* pQTLs (Extended Data Fig. 5).

## pQTL analysis by ancestry group

Analyses of different ancestry groups enables the assessment of greater sequence diversity and variable patterns of LD to refine association

signals to fewer variants[12]. Recently, Katz and colleagues used the Olink Explore 1536 platform to analyse the levels of 1,472 proteins in the plasma of 489 individuals with African ancestries[10]. Applying the same cut-off for significance ($P < 1.8 \times 10^{-9}$) to both datasets, for the same 1,472 proteins we detected *cis* pQTLs in the UKB-AF dataset ($n = 1,513$ individuals) for 628 proteins, whereas they detected only 307 using a 3 times smaller sample size. We validated the existence of *cis* pQTLs for 301 of their 307 proteins. Furthermore, Zhang and colleagues have reported the analysis of plasma protein levels of 1,871 individuals with African ancestries using the SomaScan platform[26] (4,437 targeted proteins analysed). Of these, 1,746 proteins are also targeted by Olink Explore 3072, which we used to analyse the UKB-AF group ($n = 1,513$). Applying the same cut-off for significance ($P < 1.8 \times 10^{-9}$) to both datasets and considering the 1,746 overlapping targets, we detect *cis* pQTLs for a similar number of proteins using Olink (667) as they did using SomaScan (671) in these two sets of similar population size. Of these, 417 proteins have *cis* pQTLs in both studies.

We find that around 32% and 4% of the top *cis* pQTLs identified in the UKB-AF and UKB-SA ancestry groups, respectively, were variants absent from or extremely rare in the UKB-BI ancestry group (Supplementary Tables 26 and 27). For example, the predicted loss-of-function variant rs28362286 (p.Cys679Ter) in *PCSK9*, which has been associated with low levels of low-density lipoprotein cholesterol[12,27] (−0.92 s.d. per copy), was carried by 1 in 50 participants with African ancestries but was almost absent from other participants and associated with 2.1 s.d. lower levels of PCSK9 ($P = 6.1 \times 10^{-33}$). The lower levels we observed are consistent with reports of the stop-gained variant rs28362286 (p.Cys679Ter) preventing the secretion of PCSK9 (ref. 28). Also, the sickle cell anaemia variant Gly7Val in the *HBB* gene[29] (which encodes the β-subunit of haemoglobin) was seen almost exclusively among participants with African ancestries (minor allele frequency (MAF) =7.5%), where it associated in *trans* ($P = 2.9 \times 10^{-12}$) with 0.50 s.d. higher levels of *HMOX1*, which encodes haem oxygenase-1, an enzyme that degrades free hemin. Hemin is released intravascularly in sickle cell disease and is a known inducer of HMOX1 (ref. 30).

In the UKB-AF group, *cis* pQTLs are in high LD with fewer variants on average (12) than in the UKB-BI or UKB-SA groups (37 and 29, respectively), consistent with greater sequence diversity and lower LD in populations with African ancestries than in other populations[31]. Out of 893 proteins with *cis* pQTLs in both UKB-BI and UKB-AF, for 324 proteins (36%), the top *cis* pQTLs in the UKB-BI and UKB-AF populations are in high LD ($r^2 > 0.8$ between the two variants) in UKB-BI. For 62 of these proteins, substantial refinement of the *cis* pQTL locus was achieved in UKB-AF, where the top *cis* pQTL in the UKB-AF group is in high LD with 5 or fewer variants but with 15 or more variants in the UKB-BI group (Fig. 2a). For example, rs6794768 is the top *cis* pQTL for SERPINI2 in UKB-BI and UKB-AF groups, but in UKB-AF, the signal is refined to markedly fewer variants than in the former (Fig. 2b). At the CD58 locus, the sentinel *cis* pQTLs in the UKB-BI and UKB-AF data were in high LD ($r^2 = 0.96$ in UKB-BI). However, the number of highly correlated variants is much smaller in the UKB-AF group than in the UKB-BI group (3 versus 37 variants). Since the pQTL in UKB-BI associates with multiple sclerosis, the refinement allowed in UKB-AF of the pQTL signal indicates the potential gain from investigating disease correlation with the same variant in a population of African origin (Fig. 2c). This would require us to determine the association with multiple sclerosis among individuals with African ancestries.

## pQTL comparison between platforms

In both the Icelandic and UKB-BI cohorts, the sample size is sufficiently large that the number of proteins with a *cis* pQTL is not likely to change much by increasing it (Extended Data Fig. 5). *Cis* pQTLs were present for 2,101 (71%) Olink assays and 2,120 (43%) SomaScan assays (Table 1). Thus, whereas a very similar number of proteins had *cis* pQTLs on the two platforms (2,093 on Olink and 2,044 on SomaScan), a larger fraction of proteins on Olink than on SomaScan had *cis* pQTLs. On both platforms, most assays had pQTLs: *trans* pQTLs were present for 2,528 (86%) and 4,716 (95%) Olink and SomaScan assays, respectively. There were more *trans* than *cis* sentinel pQTLs associations on both platforms, but a larger number of secondary associations in *cis* than in *trans* (Table 1 and Supplementary Tables 21 and 19).

The fraction of assays with *cis* pQTLs varied depending on several factors including dilution group, subcellular location and CV, in both the Icelandic and UKB-BI datasets (Extended Data Fig. 6 and Supplementary Table 28). On both platforms, more abundant proteins—as reflected by the assays requiring greater dilution—were more likely to have a *cis* pQTL. On both platforms, assays targeting secreted proteins were more likely to have *cis* pQTLs than assays targeting intracellular proteins, with assays targeting membrane proteins falling in between. On both platforms, the fraction of assays with *cis* pQTLs went up with assay precision, as reflected by CV.

When we restricted our analysis to matching assays for 1,848 unique proteins targeted on both platforms, the Olink assays were more likely to have a *cis* pQTL (80% of 1,864) than the SomaScan assays (58% of 1,994) (Table 1). We note that correlation between assays targeting the same protein is substantially higher when we observe a *cis* pQTL on both Olink and SomaScan ($r = 0.48$) than when we observe a *cis* pQTL on neither ($r = 0.17$) or on one only ($r = 0.11$) (Supplementary Table 29).

The large number of whole-genome-sequenced individuals in the two study populations, followed by imputation, enabled us to detect associations with rare sequence variants (Table 1). On Olink, 2,505 variants with a MAF below 0.1% (10% of 25,147 variants) associated with protein levels, and 1,596 variants (8% of 19,225 variants) did so on SomaScan.

## Relationship of pQTLs between platforms

The fraction of sentinel *cis* pQTLs in high LD ($r^2 > 0.80$) with protein-altering variants (PAVs) (33% for both Olink and SomaScan) or *cis* expression quantitative trait loci (eQTLs) (39% and 35% for Olink and SomaScan, respectively), for the gene encoding the targeted protein, was similar on the two platforms (Supplementary Tables 21 and 19). We previously concluded that the presence of a PAV and the absence of eQTL could be evidence of the association resulting from an epitope effect[2] (Supplementary Note 11). We observe very similar results on both platforms. We conclude that the 23% of sentinel Olink *cis* pQTLs and 24% of sentinel SomaScan *cis* pQTLs that are in high LD with a moderate-effect PAV but not with a *cis* eQTL are likely to be caused by epitope effects and may not in fact reflect variation in protein levels (Fig. 3 and Extended Data Fig. 5). When attempting to replicate sentinel *cis* pQTLs between platforms, we observed that when *cis* pQTLs are in high LD with PAV, the correlations between effect estimates were lower in both directions of replication, suggesting that both Olink and SomaScan assays are similarly susceptible to epitope effects caused by PAVs (Extended Data Fig. 7, Supplementary Note 12, Supplementary Fig. 7 and Supplementary Tables 30–34).

Although replication of pQTLs between platforms was similar to previous reports in smaller studies[2,3,6,7] (Supplementary Note 12), the presence of multiple independent signals at the same locus makes the comparison of pQTLs complicated, as the sentinel signal in one cohort may be a secondary signal in the other. To establish correspondence between pQTLs on the Olink and SomaScan platforms, we checked whether the sentinel variant detected on one platform was in high LD ($r^2 > 0.8$) with any of the pQTLs (sentinel or secondary) at the same locus (within 5 Mb) on the other platform. In the UKB-BI Olink data, 581 (40%) out of 1,468 sentinel *cis* pQTL signals had a corresponding pQTL in the Icelandic SomaScan data, and in 434 cases (30%) the pQTL was the sentinel signal at the locus. In the SomaScan data, 559 (48%) out of 1,164 sentinel *cis* pQTL signals had a corresponding pQTL in

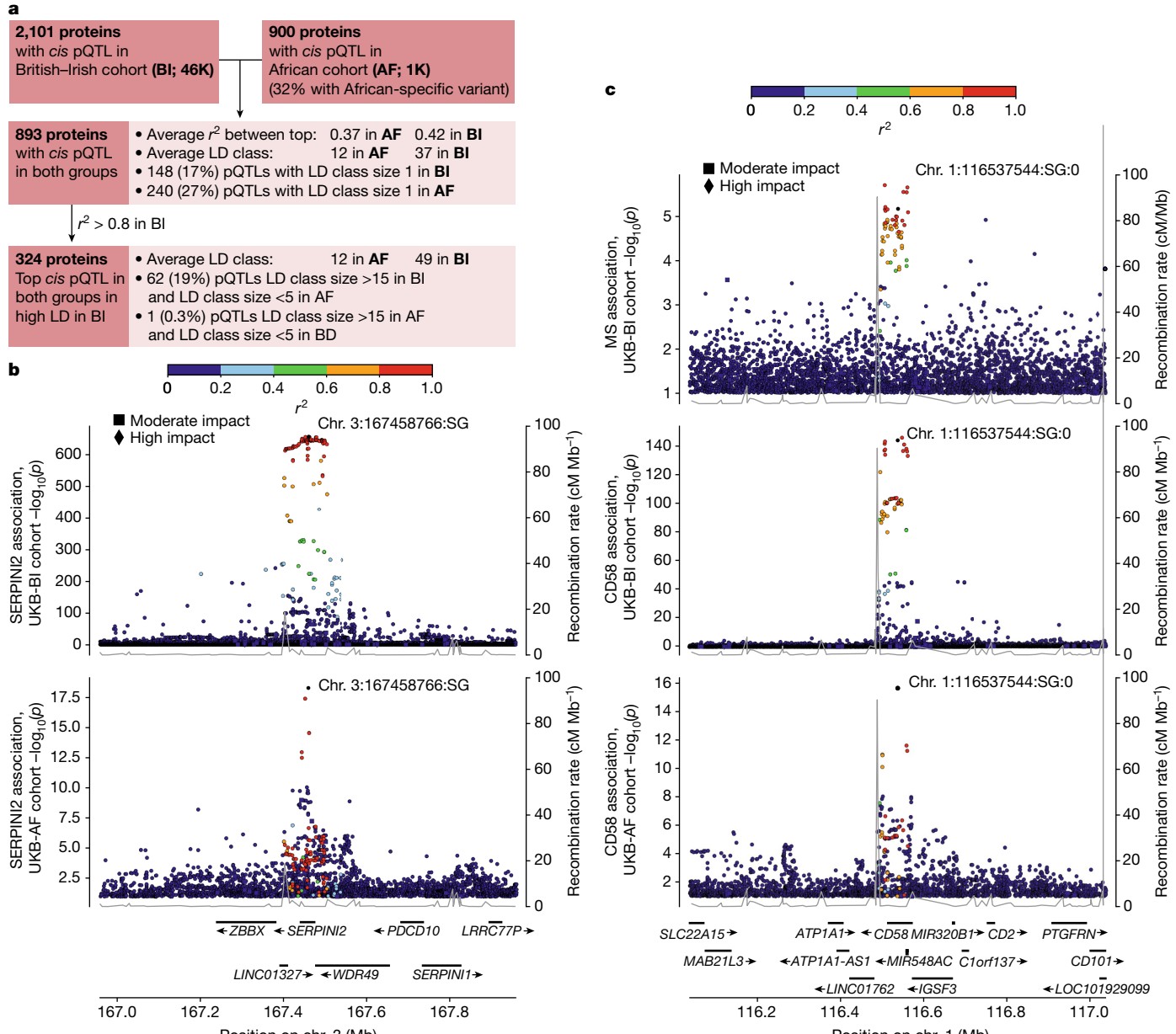

**Fig. 2 | Using different ancestry groups for locus refinement. a**, Using the more granular LD structure of the UKB-AF pQTLs to refine the location of pQTLs detected in the UKB-BI dataset. **b**, Locus plot of the sentinel *cis* pQTL for SERPINI2 in UKB-BI (top) and UKB-AF (bottom) ancestry groups. Although the sentinel *cis* pQTL for SERPINI2 is the same in the UKB-AF and UKB-BI groups, the LD class to which the variant belongs is much smaller in the UKB-AF group. This enables a more precise determination of which variant truly affects the

protein levels. **c**, Locus plot of the association at the *CD58* locus of the association with multiple sclerosis (MS) (top), and the sentinel *cis* pQTLs for CD58 in UKB-BI (middle) and UKB-AF (bottom). The locus refinement enabled by the smaller LD class in the UKB-AF group suggests that the disease association could be similarly refined. **b,c**, *P* values based on two-sided likelihood ratio test and not adjusted for multiple comparisons.

the UKB-BI Olink data, and in 449 cases (39%) the pQTL was the sentinel signal at the locus. Of the sentinel *trans* pQTL signals detected in the UKB-BI Olink data, 1,855 (10%) out of 18,578 had a corresponding pQTL in the Icelandic SomaScan data, and in most of the cases (1,777 (10%)) the pQTL was the sentinel signal at the locus. Of the sentinel *trans* pQTL signals detected in the Icelandic SomaScan data, 1,918 (19%) out of 10,352 had a corresponding pQTL in the UKB-BI Olink data, and in most of the cases (1,828 (18%)) the pQTL was the sentinel signal at the locus. Proteins having a *cis* pQTL on both platforms were more likely to have corresponding sentinel *trans* pQTLs (Supplementary Table 35 and Supplementary Note 13). Thus, even when pQTLs for a protein are detected on both platforms, they are not necessarily the same. When the

sentinel *cis* pQTLs on the two platforms are in high LD, the correlation between levels is higher than when they are not (median correlation 0.55 versus 0.49, Mann–Whitney *P* = 6.9 × 10⁻⁶) (Supplementary Table 29).

## Pleiotropic pQTLs

Some pQTLs are pleiotropic (that is, associated with a large number of proteins). A total of 46 and 35 pQTLs were associated with more than 50 proteins on Olink and SomaScan, respectively (Supplementary Table 36). Eight such pleiotropic Olink pQTLs did not associate with any protein in the SomaScan data and an additional 11 associated with fewer than 10 proteins. Conversely, one of the SomaScan pQTLs did not

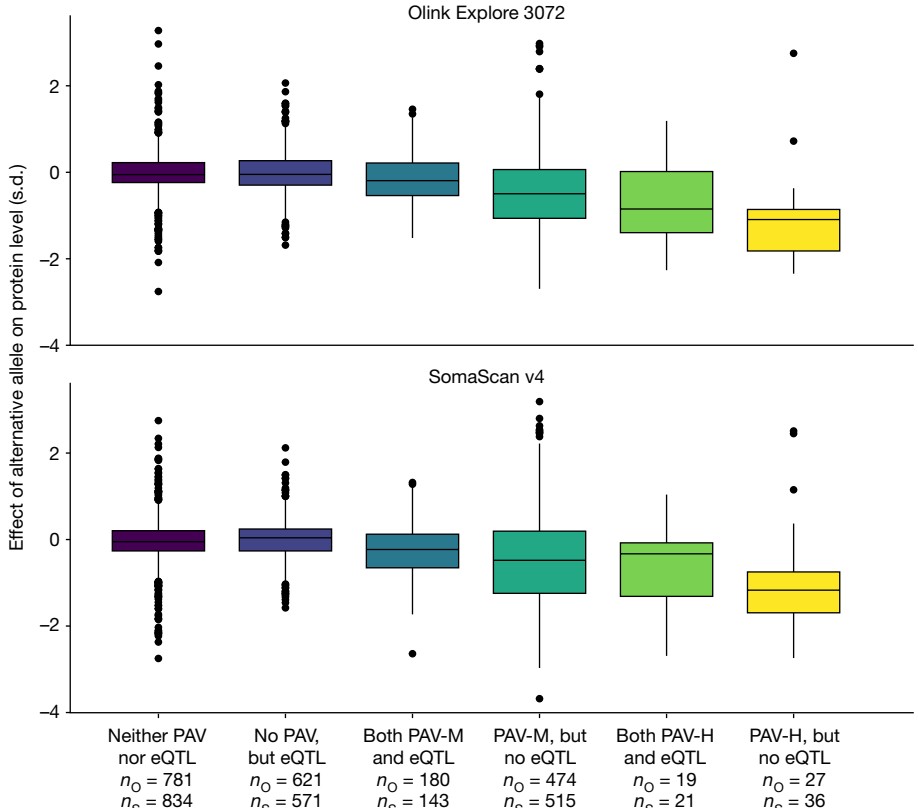

**Fig. 3 | Effect of alternative alleles.** Effect on each platform of alternative alleles broken down by the presence or absence of PAVs or *cis* eQTLs in high LD. PAV-M, moderate-impact PAV; PAV-H, high-impact PAV; $n_O$, number of *cis* pQTLs detected with Olink; $n_S$, number of *cis* pQTLs detected with SomaScan. Box plots show the median and lower and upper quartiles; whiskers extend to 1.5 times the interquartile range; points beyond whiskers are plotted individually.

associate with any Olink proteins and an additional ten associated with fewer than ten proteins (Supplementary Table 37). A number of the pQTLs that are pleiotropic on SomaScan but not on Olink are close to complement factor genes (for example, *C3*, *CFD* and *CFH*). This could be because the SomaScan sample processing or measurement may interact with complement proteins, as previously contemplated in the case of CFH[32]. Other pQTLs are pleiotropic on Olink but not on SomaScan, such as variants in *PNPLA3* and *FADS1*—genes involved in the regulation of fat in liver.

Other such differences between studies and/or platforms due to variants associated with platelet counts have been noted and have been suggested to be at least partially owing to differences in sample handling and storage[2,33] (Supplementary Note 14).

## pQTLs and disease-associated variants

The establishment of a relationship between a variant associating with a disease and a *cis* pQTL makes it likely that the variant is at least in part mediating risk through the associated protein. We use three methods to establish such a relationship: high LD ($r^2 > 0.8$) between a pQTL and a disease-associated variant, inclusion of a disease-associated variant in the subset of variants in the credible set in high LD with the pQTL, and for specific examples where the necessary statistics are available, statistical colocalization (posterior probability (PP)).

For the Olink-UKB-BI data, there were 2,409 pairs of genome-wide association study (GWAS) catalogue variants and *cis* pQTLs where the GWAS catalogue variant was in high LD with the pQTL and included in the 95% credible set for the pQTL, whereas for the Icelandic Soma-Scan data there were 1,597 such pairs. In addition, there were 529,604 and 196,836 such pairs for *trans* pQTLs detected in the Olink-UKB-BI

data and the Icelandic SomaScan data, respectively (Supplementary Tables 38–40 and Supplementary Note 15).

On Olink and SomaScan, counting only the unique pQTLs yields 403 and 359 *cis* pQTLs, respectively, and 2,830 and 1,782 *trans* pQTLs, respectively, where at least one disease or trait is related as described above to the levels of at least one protein.

For proteins targeted on both platforms, we demonstrate in the Olink-UKB-PPP data examples of independent replication of the relationship between *cis* pQTLs and disease-associating variants previously discovered using SomaScan[2]. These include a variant associating with lower SULT2A1 levels and less risk of gallstones and a variant associating with lower CHRDL2 levels and less risk of colorectal cancer (Supplementary Note 16).

Several *cis* pQTLs that are in high LD with a disease-associating variant were observed only on one platform. At the *IL10* locus, the minor allele (A) of rs3024493 (MAF = 15%) and its correlate rs3024505 ($r^2 = 1.00$) associate with increased risk of IBD[34]. However, the variants have not been reported to be in high LD with coding variants, sentinel *cis* eQTL or *cis* pQTL, but *IL10* has been considered the most likely candidate gene in the region on the basis of its anti-inflammatory function and the fact that IL-10-deficient mice develop chronic enterocolitis[35]. In the UKB data, the IBD risk allele of rs3024493 is the top *cis* pQTL for IL-10 and associates with lower plasma level of IL-10 ($P = 1.4 \times 10^{-52}$, effect = −0.14 s.d., PP = 0.98), but not with any other protein measured using Olink (Fig. 4). By contrast, no *cis* pQTL was detected for IL-10 using the Icelandic SomaScan data. Of note, the missense variant Ser159Gly (rs3135932-G; MAF = 17%) in *IL10RA* associates in *trans* with greater IL-10 levels measured on Olink (effect = 0.26 s.d.; $P = 2.6 \times 10^{-195}$). Ser159Gly also associates in *cis* with higher IL10RA levels (effect = 0.18 s.d.; $P = 6.5 \times 10^{-94}$; Extended Data Fig. 8). *IL10RA* is the most commonly

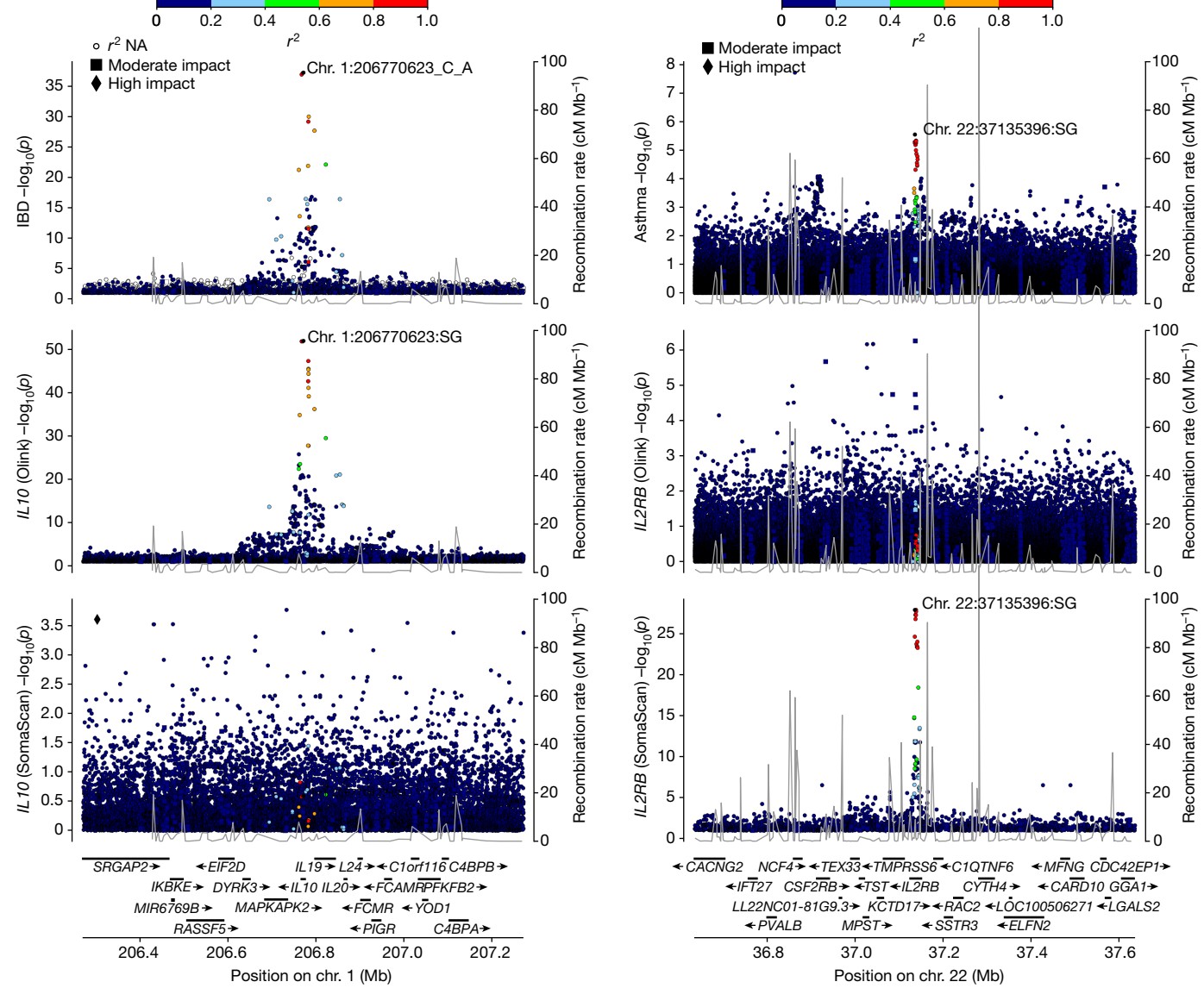

**Fig. 4 | pQTLs that are detected on one platform only and their relationship with disease-associated variants.** Left, association at the *IL10* locus between sequence variants and IBD (top) and levels of IL-10 as measured using Olink (middle) and SomaScan (bottom). The IL-10 protein is targeted by assays on both platforms, but no *cis* pQTLs were observed using the SomaScan platform. Right, association at the *IL2RB* locus between sequence variants and asthma (top) and levels of IL2RB as measured by Olink (middle) and SomaScan (bottom). The colour code indicates the $r^2$ values for each variant with the labelled one. The IL2RB protein is targeted by assays on both platforms, but no *cis* pQTLs were observed using the Olink platform. *P* values based on a two-sided likelihood ratio test and not adjusted for multiple comparisons. NA, not applicable.

reported gene for monogenic IBD[36] and in a meta-analysis of 20,295 IBD cases, we found Ser159Gly to associate with greater risk of IBD (OR = 1.05, $P = 1.6 \times 10^{-5}$). Additionally, we measured IL-10 levels with an enzyme-linked immunosorbent assay (ELISA) and observed a high correlation with levels measured on Olink ($r = 0.81$), but not on SomaScan ($r = 0.04$) (Extended Data Fig. 9).

The sentinel *cis* pQTL for tumour necrosis factor ligand superfamily member 11 (TNFSF11) (also known as RANKL) using Olink (chr. 13:42478744; MAF = 47%, $P = 10^{-75}$; effect = 0.12 s.d.) is in high LD ($r^2 = 0.99$) with a variant associated with primary biliary cirrhosis[37] (PBC) (rs9533122; $P = 6.0 \times 10^{-13}$). The minor allele associates with a reduction in PBC risk by 14% and higher TNFSF11 level, but is not in LD with a PAV or a *cis* eQTL. No *cis* pQTLs were detected for TNFSF11 using SomaScan, and the correlation of TNFSF11 levels between the two platforms is low ($r = 0.02$). TNFSF11 is involved in establishing self-tolerance and as such may have a role in autoimmune diseases such as PBC[38].

The sentinel *cis* pQTL for ERBB4 on Olink associates with higher ERBB4 levels (rs6735267; MAF = 27%, $P = 10^{-198}$, effect = 0.22 s.d.) and a highly correlated variant ($r^2 = 0.99$) associates with lower BMI[39] (rs7599312; $P = 2 \times 10^{-23}$, effect = −0.01 s.d., PP = 1.00). No *cis* pQTLs were detected for ERBB4 on SomaScan, and the correlation between the levels of ERBB4 measured on the two platforms was low ($r = -0.02$). We did not observe any eQTL or PAV in high LD with these variants. *ERBB4* encodes a receptor tyrosine kinase expressed in liver and pancreas, and ERBB4 disruption has been linked to impaired glucose tolerance and reduced insulin response in mice, supporting its possible effect on obesity[40].

At the locus of *GRP* (encoding gastrin-releasing peptide and neuromedin-C), the minor alleles of variants in high LD—including rs7243357 and rs9957145 (MAF = 17%)—associate with a lower BMI[39] and less risk of type 2 diabetes[41]. However, the variants have not been reported to be in high LD with coding variants, sentinel *cis* eQTL or *cis* pQTL. In the UKB data, the minor allele of rs9961404 is the top *cis*

pQTL for gastrin-releasing peptide (GRP) and associates with higher plasma level of GRP as measured on Olink ($P = 2.1 \times 10^{-568}$, effect = 0.46 s.d.) and is highly correlated with rs7243357 and rs9957145 ($r^2 = 0.90$ and 0.99, PP = 0.96). The GRP levels based on Olink did not correlate with levels on SomaScan as measured by two aptamers ($r = -0.01$ and $r = 0.00$) and no *cis* pQTL was detected on SomaScan. Notably, the peptides encoded by *GRP* have multiple roles including the release of gastrin from the stomach and the control of food intake[42]. A similar correlation of variants with GRP levels on Olink and its association with a diabetes-associated variant has been previously noted[9]. Other such examples can be found in Supplementary Note 17.

At the *IL2RB* locus (encoding the beta subunit of the interleukin-2 receptor), the minor allele (A) of rs228953 (MAF = 47%), associates with lower eosinophil count and less risk of asthma but the variant has not been reported to be in high LD with a coding variant[43,44]. In Iceland, the minor allele of rs228958 is the top *cis* pQTL for IL2RB, associating with higher plasma level as measured on SomaScan ($P = 1.3 \times 10^{-28}$, effect = 0.09 s.d.) as well as in a report and in high LD with rs228953 (ref. 32) ($r^2 = 0.93$, PP = 0.78) (Fig. 4). The IL2RB levels based on Olink have a weak correlation with levels on SomaScan ($r = 0.14$) and no *cis* pQTL was detected on Olink, consistent with IL2RB being one of the assays on the Olink platform with the highest fraction of measurements below the LOD (79%). IL2RB constitutes one of the receptor subunits for IL-2 and IL-15 and dysregulated signalling of both of these immunoregulatory cytokines has been linked to asthma and allergy[45,46].

In some instances, we detected *cis* pQTLs for a given protein on both platforms, but they had little correlation and only one of them was in high LD with a disease-associated variant. For example, the minor allele of the sentinel *cis* pQTL for CD58 associated with lower protein levels on Olink and less risk of multiple sclerosis (PP = 0.98). Of interest, the CD58 pQTL can be substantially refined by cross-ancestry analysis. However, the sentinel *cis* pQTL on SomaScan was in low LD ($r^2 = 0.04$) with the one on Olink and did not associate with multiple sclerosis (Fig. 2, Extended Data Fig. 8 and Supplementary Note 18).

In addition to our observation of *cis* pQTLs associating with diseases and other traits possibly shedding light on pathogenesis, we also noted such association for a large number of *trans* pQTLs. The correlation of a *trans* pQTL with a variant associating with a disease can be interpreted in three ways. First, the change in protein levels may be a consequence of the disease predisposed by the variant. Second, the variant may be affecting the disease risk through a protein encoded by the gene at the variant locus affecting another protein in the same pathway, reflected in plasma protein levels and the *trans* pQTL. Third, the variant may affect the protein levels and the disease risk independently of each other.

Similar to the logic underlying Mendelian randomization, when all variants associating significantly with the levels of a given protein also associate proportionally with the risk of a particular disease, we propose that the protein has a role in the pathogenesis of the disease[47]. When all variants associating significantly with a given disease also associate proportionally with the levels of a particular protein in plasma (*trans* pQTLs), and in the absence of the conditions described above, we propose that the change in protein levels is a consequence of the disease. We note that often, the *trans* pQTLs associate with a protein with enriched expression in the tissue affected by the disease.

The proteins affected by these *trans* pQTLs can point to potential biomarkers of diseases. Using SomaScan data, we have previously noted that variants associating with psoriasis also associate with levels of DEFB4A, a protein highly expressed in skin, pointing to a potential disease biomarker[2]. Of these variants, the variant most significantly associated with disease and protein levels at the *IL12B* locus (rs12188300) was also detected as *trans* pQTL of DEFB4A in plasma based on Olink. The levels of DEFB4A are highly correlated between the two platforms ($r = 0.81$) and *cis* pQTLs are observed on both platforms.

Similarly, several *trans* pQTLs for PRSS2 at different loci are in high LD with diabetes-associating variants. PRSS2 encodes trypsinogen, a protein highly expressed in exocrine pancreas, and the *trans* pQTLs may reflect damage of the pancreas among individuals with diabetes[48] (Supplementary Tables 38 and 39). The levels of PRSS2 are highly correlated between the two platforms ($r = 0.78$) and *cis* pQTLs are observed on both platforms.

To assess which *trans* pQTLs associating with disease are likely to correspond to an interaction between proteins in the same pathway, we assessed whether each protein affected by a *trans* pQTL interacts with the protein encoded by the closest gene to the variant according to the STRING database[49]. For about 9% of the *trans* pQTLs in high LD with a disease- or trait-associated variant, the two proteins of interest are known to interact. For instance, we note that a variant in TLR3 associates with autoimmune thyroid disease[50] and levels of IFNL1 in *trans*, consistent with the fact that TLR3 is known to activate IFNL1 (ref. 51). The *trans* pQTL with IFNL1 strongly supports the role of TLR3 at the variant locus in the pathogenesis of autoimmune thyroid disease.

Examples of proteins targeted only by one of the platforms and related to disease-associated variants imclude ITGA11 on Olink and GREM1, ASIP and STAT3 on SomaScan (Supplementary Note 19 and Supplementary Fig. 8).

## Discussion

The amount of data generated using Olink Explore 3072 in the UKB and SomaScan v4 in Iceland allows the identification of a large number of pQTLs as well as associations between phenotypes and protein levels in plasma, enabling the comparison of the platforms. In these two datasets of similar sample sizes, we observe differences in the detection of pQTLs. We directly compared measurements on the two platforms for twice the number of proteins considered in previous studies and show a modest correlation, consistent with recent reports[7,10,11]. Finally, we demonstrated how these differences can affect the conclusions drawn from the integration of proteomics and genetics in the study of diseases.

The large number of measurements on the Olink platform (50,000 individuals) enables the detection of pQTLs for more than 2 times the number of proteins reported in previous studies using smaller sample sizes[7,9]. On both platforms, a substantial fraction of proteins with a *cis* pQTL is targeted by that platform only. On the Olink platform, the majority of proteins already have a *cis* pQTL, suggesting that the increased sample size may yield diminishing returns in terms of the number of new *cis* pQTLs detected. Overall, we detected *cis* pQTLs in plasma for 3,129 unique proteins out of the 5,814 proteins targeted by at least 1 platform. Although not all proteins are expected to have a *cis* pQTL in plasma, the large fraction of proteins with no *cis* pQTLs (around half) suggests that not all proteins can be easily measured in plasma. Consistently, a number of proteins have a large fraction of measurements below the LOD as estimated on the Olink platform, particularly proteins with low expected abundance in plasma, as reflected by the dilution group. In addition, proteins in lower dilution groups, corresponding to lower expected abundance, showed lower correlation between platforms and a lower fraction of *cis* pQTLs, suggesting less reliable measurements.

Using the presence of *cis* pQTLs on the two platforms and the correlation in protein levels between them, the proteins targeted by the platforms can be organized into tiers by confidence (Extended Data Fig. 10 and Supplementary Table 29). Of all proteins targeted by either platform, about 500 had *cis* pQTLs on both platforms and strong correlation between protein levels as measured by the 2 platforms. These can be said to be measured with high confidence on the two platforms (tier 1). About 2,600 had a *cis* pQTL on at least 1 of the platforms but either lacked a *cis* pQTL on one platform, were not highly correlated between platforms, or both (tier 2). Finally, about a further 3,000 did not have a *cis* pQTL on either platform (tier 3). We believe that this classification of proteins can be useful for prioritizing the orthogonal validation of the assays—for example, by mass spectrometry.

Whereas we observed a similar total number of proteins having a *cis* pQTL on the 2 platforms, when considering the 1,848 proteins targeted by both platforms, we observe a greater number of *cis* pQTLs using Olink in UKB than using SomaScan in Iceland (80% versus 60%). Given the enrichment of secreted and abundant proteins in the collection of proteins currently targeted on the platforms compared with the Human Protein Atlas, we expect proteins currently targeted by neither of the two platforms to be even more challenging to measure in plasma[14] (Supplementary Note 20).

Even where *cis* pQTLs provide evidence that both platforms are measuring the targeted protein, in more than half of the cases, the top associated variants were in low LD. Although *cis* pQTLs provide strong evidence that the protein being measured is in fact encoded by the gene of interest, they do not indicate which proteoform is being measured. The difference in pQTLs between the two platforms is consistent with proteoforms being differentially targeted by the platforms, as suggested by previous work on smaller sample sizes, both in terms of individuals and proteins[7]. Proteoforms encoded by the same gene may participate in different biological processes and therefore have different associations with diseases and other traits. Furthermore, some *cis* pQTLs may correspond to epitope effects rather than protein levels, particularly when the *cis* pQTL correlates with a coding variant and in the absence of a correlated eQTL[2,7]. The differences that we note in plasma protein levels or in the pQTLs between platforms influence the results from integrating protein levels and genetics in the study of diseases.

Around a third of the *trans* pQTLs associating with many proteins on one platform are not pleiotropic in the other. This may be caused by sequence variants interacting with pre-analytical variables, including sample handling and processing, or with the measurements themselves. Such differences in pleiotropic *trans* pQTLs have previously been noted between studies using the same platform but different protocols[32].

Large-scale proteomics studies have been performed predominantly in populations with European ancestry and more recently in some non-European populations[10,26,52,53]. Leveraging the diverse ancestry composition of the UK Biobank cohort measured on Olink, we detected associations between protein levels and ancestry-specific variants. In addition, variants associated with protein levels are, on average, in high LD with three times fewer variants in the African ancestries group (UKB-AF) than in the British or Irish ancestry group (UKB-BI). The lower average LD in UKB-AF enables substantial locus refinement of association signals in a large number of cases. Similar observations of locus refinement were made in 2022 using SomaScan in groups with European and African ancestries[26]. We have shown that performing cross-ancestry signal refinement on pQTLs may also assist in refining associated disease signals, even if the diseases are studied in only one of the ancestry groups.

## Limitations

Although both the Olink and SomaScan platforms are affinity-based, they differ in nature, as one is based on antibodies and the other on aptamers. This may affect how proteins are quantified in complex samples such as plasma. The biochemical properties of orthogonal assays used for validation may need to be considered in the context of the properties of the two platforms.

Protein concentration varies between tissues and sample types[14]. The current work is limited to plasma and some of the results and conclusions may be specific to this sample type. The analysis of other sample types using these platforms requires separate assessment; such assessment has begun for some sample types, such as cerebrospinal fluid[54,55]. Analysis of pQTLs in different sample types in large datasets is likely to be highly informative.

Although the current study attempts to assess the proteome of individuals with non-European ancestry, the sample size is still limited.

The differences in genetic association with protein levels between the ancestries are of high interest and our results suggest that larger sample sizes in the cohorts with non-European ancestries will further our understanding of these differences.

Our study suggests that some of the differences between platforms may lie in their sensitivity to different proteoforms, but the contribution of the various proteoforms remains to be studied.

The SMP normalization of the SomaScan data has a considerable effect on downstream analysis, as reflected by lower correlation with Olink measurements, higher fraction of assays with *cis* pQTLs, and differences in associations (Supplementary Note 4). The full extent of the effects of the SMP normalization warrants further study.

The platforms may be differently affected by epitope effects. We have not systematically performed assays such as ELISA or used other methods for each of the proteins or proteoforms studied, as such validation is currently difficult to perform at scale. Where the results from the platforms are discordant, further studies are required to determine which platform to believe, although evidence such as orthogonal validation or the existence of *cis* pQTLs can provide some insight.

## Conclusion

Both platforms are expanding the number of targeted proteins, and it can be predicted that proteins with evidence of *cis* pQTLs in plasma on one platform can guide the selection of proteins on the other platforms towards those that have the highest chance of successful measurement in that medium. Thus, each of the two platforms could select 1,000 additional proteins with *cis* pQTLs documented on the other platform. In addition, there are reasons to believe that the two platforms may measure different proteoforms for up to 500 proteins. We foresee that future versions of proteomics assays will target specific proteoforms encoded by a given gene instead of being referred to as targeting a single version of a protein.

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

## Methods

### Study populations

Plasma samples collected from 54,265 UKB participants at their baseline visit were measured using Olink Explore 3072 as a part of UKB-PPP (UK Biobank application number 65851). All participants provided informed consent. A large majority of the samples were randomly selected across the UK Biobank, and only those were used for the analysis presented here. Many GWASs using the UKB data[56] have been based on a prescribed European ancestry subset of 409,559 participants who self-identified as 'white British'[57]. To better leverage the value of a wider range of UKB participants, we defined three cohorts encompassing 450,690 individuals, based on genetic clustering of microarray genotypes informed by self-described ethnicity and supervised ancestry inference[12]: 431,805 individuals with British or Irish ancestry (UKB-BI, 46,218 with Olink data), 9,633 individuals with African ancestries (UKB-AF, 1,513 with Olink data) and 9,252 individuals with South Asian ancestries (UKB-SA, 953 with Olink data).

Samples likely to be incorrectly labelled were identified based on individual predictions of sex by protein levels, and of protein levels by genotypes. Whole plates or individual rows or columns of samples, identified as being majority likely incorrectly labelled, were excluded from the UKB-PPP data. From the Expansion set of assays, this resulted in the exclusion of 13 whole plates and five rows or columns of samples, in total 1,179 samples. From the 1536 set of assays, this resulted in the exclusion of four whole plates and seven rows or columns of samples, in total 404 samples. Furthermore, in the 1536 set of assays, a single panel was excluded for two plates, affecting 174 samples.

We measured the plasma protein levels of 35,892 Icelanders using SomaScan v4 (ref. 2). All participants who donated samples gave informed consent, and the National Bioethics Committee of Iceland approved the study, which was conducted in agreement with conditions issued by the Data Protection Authority of Iceland (VSN_14-015). Personal identities for the participants' data and biological samples were encrypted by a third-party system (Identity Protection System), approved and monitored by the Data Protection Authority. In addition, we measured 1,514 of these Icelanders with the Olink Explore 3072 platform using the same plasma sample.

We used 1,474 and 227 additional duplicate measurements of samples to evaluate assay precision for the Olink Explore (UKB sets) and SomaScan (Iceland 36K) platforms, respectively. For samples that were measured more than twice, two of the measurements were chosen at random.

### External data sources

URLs for external data used are as follows: the GWAS catalogue (https://www.ebi.ac.uk/gwas/), the GTEx project (https://gtexportal.org/home/), the Human Protein Atlas (https://www.proteinatlas.org/), STRING database (https://string-db.org/; file name: 9606.protein.actions.v11.txt.gz) and UniProt (https://www.uniprot.org/).

### Software and data processing pipelines

We used the following publicly available software in conjunction with the algorithms described above. BamQC (v1.0.0, https://github.com/DecodeGenetics/BamQC), GraphTyper (v2.7.1, v1.4, v2.7.2, https://github.com/DecodeGenetics/graphtyper), GATK resource bundle (v4.0.12, gs://genomics-public-data/resources/broad/hg38/v0), Svimmer (v0.1, https://github.com/DecodeGenetics/svimmer), popSTR (v2.0, https://github.com/DecodeGenetics/popSTR), Admixture (v1.3.0, https://dalexander.github.io/admixture), Dipcall (v0.1, https://github.com/lh3/dipcall), RTG Tools (v3.8.4, https://github.com/RealTimeGenomics/rtg-tools), bcl2fastq (v2.20.0.422, https://support.illumina.com/sequencing/sequencing_software/bcl2fastq-conversion-software.html), Samtools (v1.9, v1.3, https://github.com/samtools/samtools), samblaster (v0.1.24, https://github.com/GregoryFaust/samblaster), BWA (v0.7.10 mem, https://github.com/lh3/bwa), GenomeAnalysisTKLite (v2.3.9, https://github.com/broadgsa/gatk), Picard tools (v1.117, https://broadinstitute.github.io/picard), Bedtools (v2.25.0-76-g5e7c696z, https://github.com/arq5x/bedtools2), Variant Effect Predictor (release 100, https://github.com/Ensembl/ensembl-vep), BOLT-LMM (v2.1, https://data.broadinstitute.org/alkesgroup/BOLT-LMM/downloads), IMPUTE2 (v2.3.1, https://mathgen.stats.ox.ac.uk/impute/impute_v2.html), dbSNP (v140, https://www.ncbi.nlm.nih.gov/SNP), BiNGO (v3.0.3, https://www.psb.ugent.be/cbd/papers/BiNGO/Download.html), Cytoscape (v3.7.1, https://cytoscape.org/download.html), COLOC (v5.1.0.1, https://github.com/chr1swallace/coloc). The genomics and pQTL processing pipelines have been extensively described previously[2,12]. To process data generated on the Olink platform, we used Olink Explore (v1.9.0, https://www.olink.com/products-services/data-analysis-products/npx-explore/). Data were analysed and figures generated using Python (version 3.9.1), along with packages numpy (version 1.20.3), scipy (version 1.7.1), matplotlib (version 3.4.3), and pandas (version 1.3.0), and R (version 3.6.0).

### Proteomic platforms

The Olink Explore 3072 proximity extension assay (PEA) platform is based upon an in-solution binding of two polyclonal antibody pools to a target protein and subsequent hybridization and enrichment of two unique single-stranded DNA probes to create a double stranded barcode unique for the antigen[58]. The platform consists of 2,941 immunoassays targeting 2,925 proteins. Each assay is based on a pair of polyclonal antibodies. The antibodies bind to different sites on the target protein and are labelled with single-stranded complementary oligonucleotides. If matching pairs of antibodies bind to the protein, the attached oligonucleotides hybridize, and are then measured using next-generation sequencing[59,60]. Olink Explore 3072 consists of 8 panels of 384 assays analysed by next-generation sequencing. Four of those panels make up a previous iteration of the platform, Olink Explore 1536, which can be considered a subset of the Olink Explore 3072, along with the Expansion set. The Olink measurements were based on the NPX values recommended by the manufacturer, which include normalization[58].

The UKB plasma samples were measured at Olink's facilities in Uppsala Sweden. All samples were randomized and plated by the UK Biobank laboratory team prior to delivery. Samples were processed across three NovaSeq 6000 Sequencing Systems. Extensive quality control measures and normalization of protein concentration was performed at Olink's facilities, producing NPX values for each protein per participant. NPX is Olink's relative protein quantification unit on a $\log_2$ scale.

The Olink measurements of the Icelandic plasma samples were performed at deCODE's facility in accordance with the Olink Explore manual[61]. Quality control measures were the same as used by Olink for the UK Biobank samples.

The SomaScan platform utilizes a surface bound enrichment of proteins alongside a universal polyanionic competitor to prevent transient non-specific interactions[62]. SomaScan v4 consists of 4,907 aptamer-based assays targeting 4,719 proteins. Aptamers are short, single-stranded oligonucleotides that bind to protein targets. The bound aptamers are then quantified using DNA microarray technology[62,63]. In most studies performed using SomaScan, the last step in the normalization process adjusts the median protein levels for each individual to a reference[5,64]. As this can affect the correlation of protein levels to other factors, some studies omit this step[2]. We refer to the former data as normalized and the latter as non-normalized. In addition to using non-normalized SomaScan protein measurements as we had done before[2], we also applied SomaLogic's SMP normalization[64] and performed all analyses using both non-normalized and normalized data. Comparison of the two normalization methods can be found in Supplementary Note 4.

We refer to the outcome of a particular assay as the level of the protein, noting that the assay may not in fact measure the targeted protein.

Both Olink and SomaScan use dilutions of plasma samples to compensate for different concentrations of proteins in plasma[13,59,62]. For the set of proteins targeted by both platforms, the two platforms are generally in agreement on the placement of proteins into low, intermediate or high dilution groups (Supplementary Tables 1, 2 and 7).

## Genotyping and imputation

The whole genomes of 150,119 UKB participants were sequenced to a median of 32.5× using Illumina technology[12]. Sequence variant calling was performed using GraphTyper[65]. In addition, all UKB participants were single-nucleotide polymorphism (SNP) genotyped with Affymetrix SNP chips[66,67]. After filtering, the sequence variants along with the phased SNP chip data by Bycroft et al.[57] were used to create a haplotype reference panel. Sequence variants were then imputed into the chip-genotyped samples using tools and methods described previously[68,69]. The genotyping and imputation of the UKB dataset have previously been described in greater detail[12]. We restricted our analysis to variants with MAF >0.01% and imputation information >0.9, resulting in 57.7 million variants in the UKB-BI, 36.5 million variants in the UKB-SA and 68.6 million variants in the UKB-AF datasets.

The whole genomes of 63,118 Icelanders were sequenced to a median of 32× using Illumina technology[68]. Sequence variants were called using GraphTyper[65]. In addition, the samples were SNP genotyped with Illumina SNP chips and long-range phased, and the data was used to impute genotypes. In total, 173,025 Icelanders were SNP genotyped, long-range phased and imputed based on the sequenced datasets. Where genotypes for an individual were missing for association studies, they were inferred using genealogic information if possible. The imputation learning set was based on whole-genome sequencing of 15% of Icelanders, which allowed rare variant imputation. The genotyping and imputation on the Icelandic dataset have been previously described in greater detail[50]. We restricted our analysis to variants with MAF >0.01% and imputation information >0.9, resulting in 33.5 million variants. Other software tools used for various tasks in the genotyping pipeline were BamQC, GATK resource bundle, Svimmer, popSTR, Admixture, Dipcall, RTG Tools, bcl2fastq, Samtools, samblaster, BWA, GenomeAnalysisTKLite, Picard tools, Bedtools, Variant Effect Predictor, IMPUTE2, dbSNP, BiNGO and Cytoscape.

## Phenotypes

In UKB we used health care records to identify the diagnosis of a disease or disease category, both prior and post plasma collection, based on the first three letters of the corresponding ICD10 code. When the number of individuals diagnosed exceeded 50, we estimated the association of protein levels with disease diagnosis. This resulted in 324, 29 and 20 case–control phenotypes for UKB-BI, UKB-AF and UKB-SA, respectively. In addition, we had measurements of 208, 56, and 60 quantitative traits in UKB-BI, UKB-AF and UKB-SA respectively with at least 50 individuals measured for each trait. The quantitative traits were measured at the same time as the plasma was collected, when available.

In Iceland we used health care records to construct lists of disease diagnoses, both prior and post plasma collection. This resulted 275 case–control phenotypes. We furthermore had measurements of 110 quantitative traits from various sources, in general not measured at the same time as the plasma was collected.

## Protein–phenotype associations

We estimated the association of proteins levels with quantitative traits using linear regression. We estimated the association of protein levels with a prior or past disease in UKB and Iceland using logistic regression. All analyses were adjusted for the sex and age of the individual at the time of plasma collection, and in addition, quantitative measures were inverse normal transformed.

## Annotation of assay targets

We assigned genomic coordinates to assay targets using UniProt IDs[70] for each assay provided by the manufacturer. Out of 4,963 valid assays on the SomaScan platform (excluding non-human proteins and assays marked as defective by the manufacturer), this resulted in 4,961 assays getting assigned the genomic coordinates of their intended targets. Out of 2,941 valid assays on the Olink Explore platform, this resulted in 2,923 assays getting assigned the genomic coordinates of their intended targets.

We identified assays targeting the same protein using their UniProt IDs. This resulted in 2,023 pairs of assays targeting 1,848 UniProt IDs; 1,864 Olink assays and 1,994 SomaScan assays (Supplementary Table 4).

## Assay precision

Following Olink[58], we assumed a log-normal distribution of protein levels. On the logarithm scale, denoting the mean protein level with $\mu$ and variance with $\sigma^2$, the mean and variance of protein levels will be $e^{\mu+\sigma^2/2}$ and $(e^{\sigma^2}-1)e^{2\mu+\sigma^2}$. The CV is defined as the s.d. divided by the mean and therefore equals $\sqrt{e^{\sigma^2}-1}$ assuming a log-normal distribution.

To evaluate the precision of the assays, we estimated the CV for the available duplicate measurements and the expectation of the CV under the assumption that the two duplicates were independent of each other, that is, if the repeated measurements were not of the same sample, but of samples selected at random from the population (Supplementary Fig. 1). We used the robust median absolute deviation estimator to estimate the s.d. of the repeated measurements on the log-scale and inserted this estimate into the formula for the CV above (Fig. 1, Supplementary Fig. 1 and Supplementary Tables 1 and 2).

## Relative evaluation of batch effects between platforms

Both Olink and SomaScan use repeated measurements of control samples, specific to the platform, for quality control. When using two measurements of the same control sample on the same plate to evaluate the CV, the evaluation does not include the inter-plate variation, while the CV estimated assuming that the samples are not measured on the same plate but chosen at random from the set of all samples does include inter-plate variation. Comparing the CV computed from the repeated control samples between the two platforms can therefore help comparison in terms of batch effects, with values closer to one suggesting that the platform is less susceptible to batch effects and closer to zero that the platform is more so (Supplementary Note 2, Supplementary Fig. 9).

## Correlation of assays across platforms

We calculated correlations between protein levels measured in the same samples using Spearman correlation.

## pQTL analysis

We carried out pQTL analysis in the same way as we have previously described[2]. The following three sections briefly describe this process.

## Genome-wide association study

We rank-inverse normal transformed the measurements for each assay and adjusted them for age, sex and sample age. We standardized the residuals using rank-inverse normal transformation and used the standardized values as phenotypes for genome-wide association testing using a linear mixed model (BOLT-LMM[71]). We used LD score regression to account for inflation in test statistics due to cryptic relatedness and stratification[72].

We computed P values using a likelihood ratio test and adjusted for multiple testing by using the same significance threshold ($1.8 \times 10^{-9}$) as in our previous study on the Icelandic dataset[2].

We defined a pQTL association to be *cis* if the pQTL was located within 1 Mb of the transcription start site for the gene that encodes the target protein, as reported by UniProt, and *trans* otherwise.

Of the 2,941 assays on the Olink Explore 3072 platform, data from UKB for 2,931 assays were used for GWAS analysis.

The number of variants we test in Iceland (33.5 million) is about 40% lower than in UKB (57.7 million). The difference is largely due to very rare variants. However, the difference between them would result in a multiple testing correction threshold in UKB of $8.7 \times 10^{-10}$ instead of $1.8 \times 10^{-9}$. A total of 153 (1%) of the *cis* pQTLs are between those two thresholds and 1,608 (5%) of the *trans* pQTLs.

For replication between platforms, the *P* value threshold is 0.05, with the requirement that initial and replication associations are in the same direction.

## Conditional analysis

We performed recursive conditional analysis separately for each assay and each chromosome based on individual-level genotypes. For computational efficiency, we restricted this analysis to the candidate set of sequence variants associating with the assay with a $P < 5 \times 10^{-6}$. If the variant, v1, with the lowest *P* value had $P < 1.8 \times 10^{-9}$, we removed v1 from the candidate set and the association of all other variants in the candidate set was recomputed, conditional on v1. If any variant in the candidate set had $P < 1.8 \times 10^{-9}$, we assigned the label v2 to the variant with the lowest *P* value, removed v2 from the candidate set, and calculated the conditional association of the variants remaining in the candidate set given v1 and v2. We repeated this process until no variant in the candidate set had $P < 1.8 \times 10^{-9}$. Conditional analysis for two assays did not finish for all secondary signals but did return values for sentinel pQTLs.

We observe that 92% and 97% of secondary variants have an $r^2$ below 0.2 and 0.5, respectively, to the primary variant on Olink (based on $r^2$ calculated in the UK Biobank British and Irish set).

In addition, we estimated significance and effect based on a joint model of all variants at the locus to the phenotype for the variants selected in the stepwise model. When jointly estimating the effect on a protein at a locus, and examining pQTL associations at loci that contain more than 1 variant associated to a protein, 96% and 92% of the associations detected using SomaScan and Olink, respectively, remained significant when using the same genome-wide significance threshold as in the stepwise model (that is, $1.8 \times 10^{-9}$).

## Merging pQTLs

We considered sequence variants from the conditional analysis to belong to the same region if they were within 2 Mb of each other. Furthermore, we considered the major histocompatibility complex (MHC) region (build 38 chr. 6:25.5–34.0MB) as a single region. We refer to the most significant variant in each region as the sentinel variant for the assay in the region, and other variants as secondary variants.

We used the 'LD-based clumping approach' proposed by Sun et al.[6] to identify pQTLs associating with multiple assays: we considered variants associating with a different assay to belong to the same pQTL if they are in high LD ($r^2 > 0.8$).

## pQTL replication

For replication between platforms, the *P* value threshold was 0.05, with the requirement that initial and replication associations were in the same direction.

## Power calculation

For a given *P* value threshold *P*, sample size *N*, effect size *β*, and MAF *f*, the probability of rejecting the null hypothesis of no association is given by $1 - F(X - 1(1 - P), 2N\beta^2 f(1 - f))$, where $X^{-1}(\cdot)$ denotes the inverse cumulative distribution function (inverse CDF) of the chi-squared distribution with one degree of freedom, while $F(a, b)$ denotes the CDF of the non-central chi-squared distribution with one degree of freedom for quantile *a* and non-centrality parameter[73] *b*.

## PAV annotation of pQTL variants

For each pQTL, we tested whether the variant itself and variants in high LD ($r^2 > 0.8$) could affect the coding sequence of genes or their splicing, as described previously[2].

Based on SomaScan, 40% of variants with *cis* pQTL and 28% of variants with *trans* pQTL are in high LD with a PAV ($r^2 > 0.80$), and 44% of variants with *cis* pQTL and 38% of variants with *trans* pQTL are in high LD with *cis* eQTL ($r^2 > 0.8$).

Based on Olink, 39% of variants with *cis* pQTL and 23% of variants with *trans* pQTL are in high LD with a PAV ($r^2 > 0.80$), and 47% of variants with *cis* pQTL and 41% of variants with *trans* pQTL are in high LD with *cis* eQTL ($r^2 > 0.8$).

Thus, when considering the neighbouring genes within ±1 Mb, we note that *cis* pQTLs are more likely to be in high LD with a PAV or *cis* eQTL on both platforms compared to *trans*. Similar results were observed on both platforms and when restricting to assays measuring proteins targeted by both platforms.

In addition, for *cis* pQTLs we also report if the PAV or *cis* eQTL is for the gene encoding the targeted protein (Supplementary Tables 21 and 19).

## *Cis* eQTL and *cis* pQTL

For each *cis* pQTL, we tested whether the variant itself and variants in high LD ($r^2 > 0.8$) corresponded to one or more top *cis* eQTLs based on 73 tissues and 17 sources including the GTEx project, using the same methods and data as described previously[2].

## Integration of pQTL and disease associations

We calculated $r^2$ values (based on the Icelandic population for SomaScan-Iceland and the UKB-BI population for Olink-UKB) between all sentinel pQTL variants and top (most significantly associated) variants per Mb bin and per experimental factor ontology (EFO) term reported in the NHGRI-EBI GWAS catalogue[74] (downloaded 7 April 2022), using the same methods as described previously[2].

## Relationship between pQTLs and disease-associated variants

We identified all variants reported in the NHGRI-EBI catalogue of human GWAS[74] (excluding proteomics studies) in high LD ($r^2 > 0.8$) with sentinel pQTLs based on Olink-UKB-BI data and Icelandic SomaScan data (Supplementary Tables 38 and 39). For each sentinel pQTL association, we also identified a 95% credible set of variants (variants that most parsimoniously explain regional association[75]) likely to include the causal variant[76]. We then checked whether GWAS catalogue variants in high LD with the pQTL variant (with $r^2 > 0.8$ with the pQTL) were included in the credible set. In addition to high LD between the disease-associated variant and both the pQTL and a variant in the credible set, for the highlighted examples, we estimated the posterior probability of statistical colocalization for the variants associating with disease and protein levels when they were not identical and when we had access to the necessary statistics[77].

## Disease and pQTL colocalization

To test for colocalization of the pQTL signals with signals in other traits we used the COLOC software package implemented in R[77]. Using summary statistics for the pQTL A and the trait B—that is, effects and *P* values—we calculated Bayes factors for each of the variants in the associated region for the two traits and used COLOC to calculate the posterior probability for two hypotheses: (1) that the association with the pQTL A and the trait B are independent signals (PP3) and (2) that the association with the pQTL A and the trait B are due to a shared signal (PP4). Prior probabilities for COLOC were left at default.

## Protein subcellular locations

Protein subcellular locations were determined using annotations from the Human Protein Atlas[14], using the same approach as in Sun et al.[6] where proteins annotated as 'membrane' by the Human Protein Atlas were considered to be membrane proteins, proteins annotated in the Human Protein Atlas as 'secreted' (but not 'membrane') considered to be secreted proteins, while other proteins were considered to be intracellular.

## IL-10 ELISA

Blood was collected in EDTA tubes that were inverted 4–5 times and then centrifuged for 10 min at 3,000*g* at 4 °C. Plasma samples were frozen in aliquots at −80 °C. Plasma aliquots were allowed to thaw on ice and kept away from light during defrosting. Before measurement, the aliquots were mixed by inverting the tubes a couple of times and then centrifuged for 10 min at 3,220*g* at 4 °C.

IL-10 in plasma was measured by using MSD V-PLEX Human IL-10 (cat: K151QUD) according to the manufacturer's protocol (Meso Scale Diagnostics).

## NFL ELISA

Plasma samples were measured in duplicates with commercially available Simoa NF-light Advantage (SR-X) kit (Quanterix, cat. 103400). Samples were diluted 4:1 and incubated with 25 µl anti-NF-light immunocapture beads and 20 µl biotinylated detector antibody at 30 °C and 800 rpm for 30 min. Following the incubation, the bead-immunocomplexes were washed and resuspended before being incubated with 100 µl streptavidin-labelled β-galactosidase at 30 °C and 800 rpm for 10 min. After a second washing step, the bead-immunocomplexes and resorufin β-D-galactopyranoside were loaded onto an SR-X instrument (Quanterix) for processing and analysis.

## Reporting summary

Further information on research design is available in the Nature Portfolio Reporting Summary linked to this article.

## Data availability

Whole-genome sequencing data, genotype data and phased and imputed data for the UK Biobank dataset, as well as the proteomics data, can be accessed via the UKB research analysis platform (RAP) (https://ukbiobank.dnanexus.com/landing). The UK Biobank Resource was used under application number 65851. The Icelandic genomic data and proteomics data have been described previously[2]. Although these individual-level data cannot be shared as dictated by the Icelandic law, we are open to collaborations on these topics, as we have been in the past. GWAS summary statistics for all 2,931 Olink assays and all 4,907 SomaScan assays are available at https://www.decode.com/summarydata/. Other data presented in this study are included in this publication and its Supplementary Information.

## Code availability

The code central to power analysis and calculation of CV can be found at https://github.com/DecodeGenetics/proteomics_comparison. pQTL identification was carried out using publically available software[71], as described in the Methods section.

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

**Acknowledgements** This research has been conducted using the UK Biobank Resource under Application Number 65851. The authors received no specific funding for this work.

**Author contributions** The paper was written by G.H.E., E.F., P.S., S.H.L., U.T. and K.S. with input from H. Helgason, O.T.M., P.I.O., F.Z., S.A.G., A.H., G.L.N., G.T., M.O.U. and D.F.G. The study was conceptualized by P.S., U.T., K.S., G.H.E., E.F., S.H.L., D.F.G., A.H., G.L.N., G.T. and M.O.U. Data were provided by S.S., H.S., I.J., H. Holm, T.R. and J.S. pQTL analysis was done by E.F., G.H.E., M.I.M. and S.H.L. Disease and phenotype correlation was done by S.H.L. and G.H.E. Assay correlation and CV analysis was done by G.H.E. Identification of mislabelled samples in the UKB-PPP data was performed by T.E. and M.O.U. Additional analysis was done by B.V.H., T.E., G.L.N., G.T., M.O.U., D.F.G. and P.S. Statistical analysis was overseen by D.F.G. Additional analysis, informatics and support were provided by G.H.E., E.F., S.H.L., H. Helgason, B.V.H., P.I.O., F.Z., S.A.G., G.S., M.I.M., A.O., G.H.H., M.K.M., T.E., G.M., P.M., G.L.N., G.T. and M.O.U. Laboratory experiments were run by O.T.M., K.G. and G.L.N. Disease information was provided by T.A.O., S.S., H.S., I.J., H. Holm, T.R., P.S. and K.S. The study was supervised by P.S. and K.S.

**Competing interests** All authors are employees of deCODE Genetics, a wholly-owned subsidiary of Amgen.

**Additional information**
**Correspondence and requests for materials** should be addressed to Patrick Sulem or Kari Stefansson.

**Extended Data Fig. 1 | Properties of the data sets used in the proteomics analysis.** Proteomics measurements on the UKB data set were performed on the Olink Explore 3072 platform, while measurements on the Iceland 36K data set were performed on the SomaScan v4 platform. The Iceland 1K data set is a subset of the Iceland 36K data set, on which the same samples were measured using the Olink Explore 3072 platform in addition to the SomaScan v4 platform. Measurements of duplicated samples were used to evaluate precision of the assays. *Not all samples could be assigned to an ancestry group.

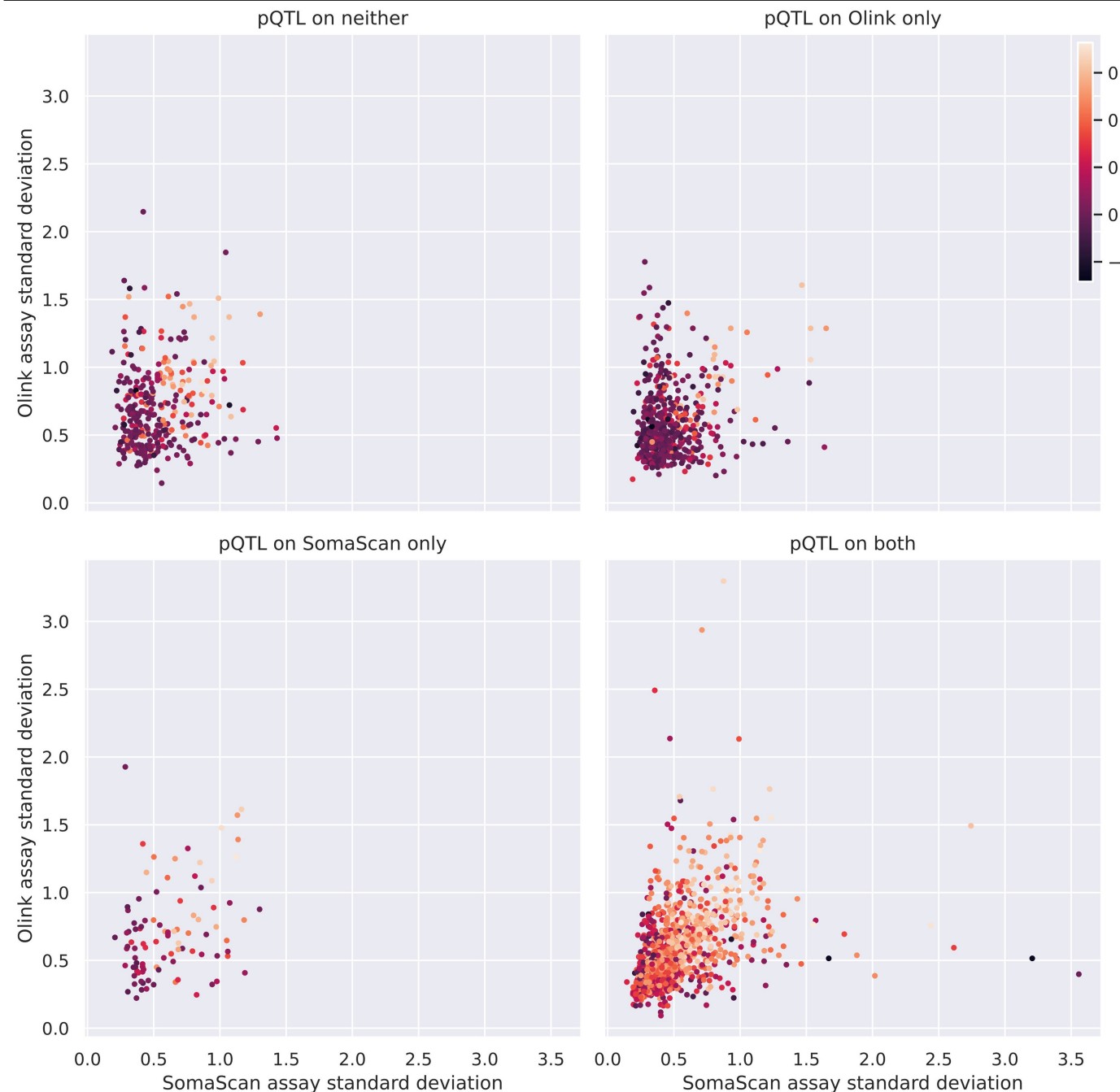

**Extended Data Fig. 2 | Variance of assays targeting the same protein.** The variance of matching SomaScan and Olink assays stratified by the presence of *cis* pQTLs and colored by the correlation of levels.

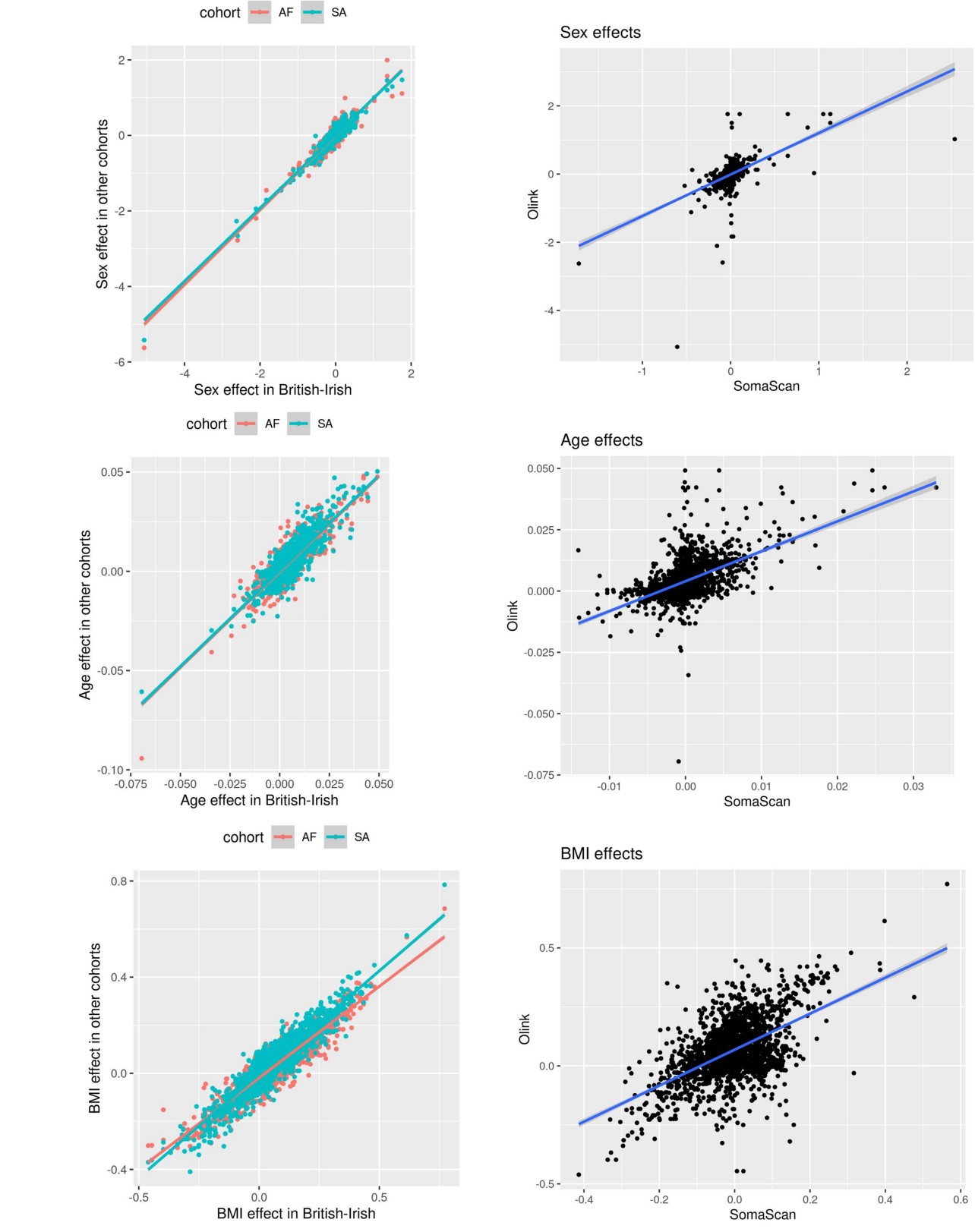

**Extended Data Fig. 3 | Correlation of sex, participant age and BMI effects.**
Left: The correlation of sex, participant age and BMI effects on protein levels
between different cohorts in the UKB data set. Right: The correlation of sex,
participant age and BMI effects on protein levels between the Olink and
SomaScan platforms.

## Olink

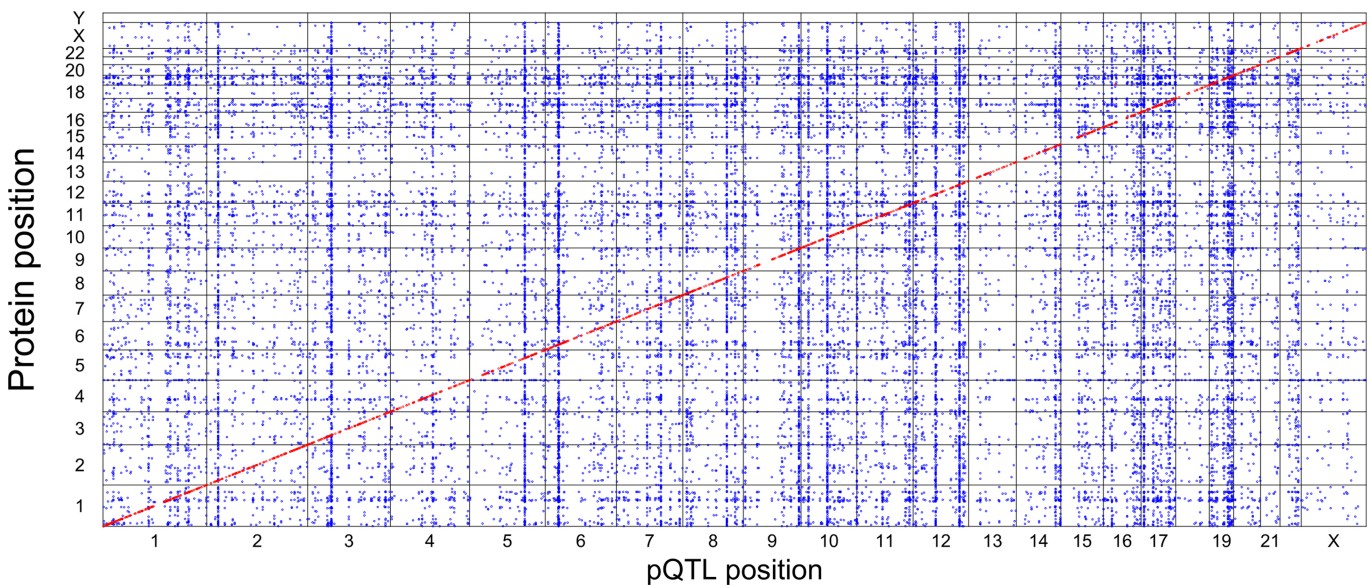

## SomaScan

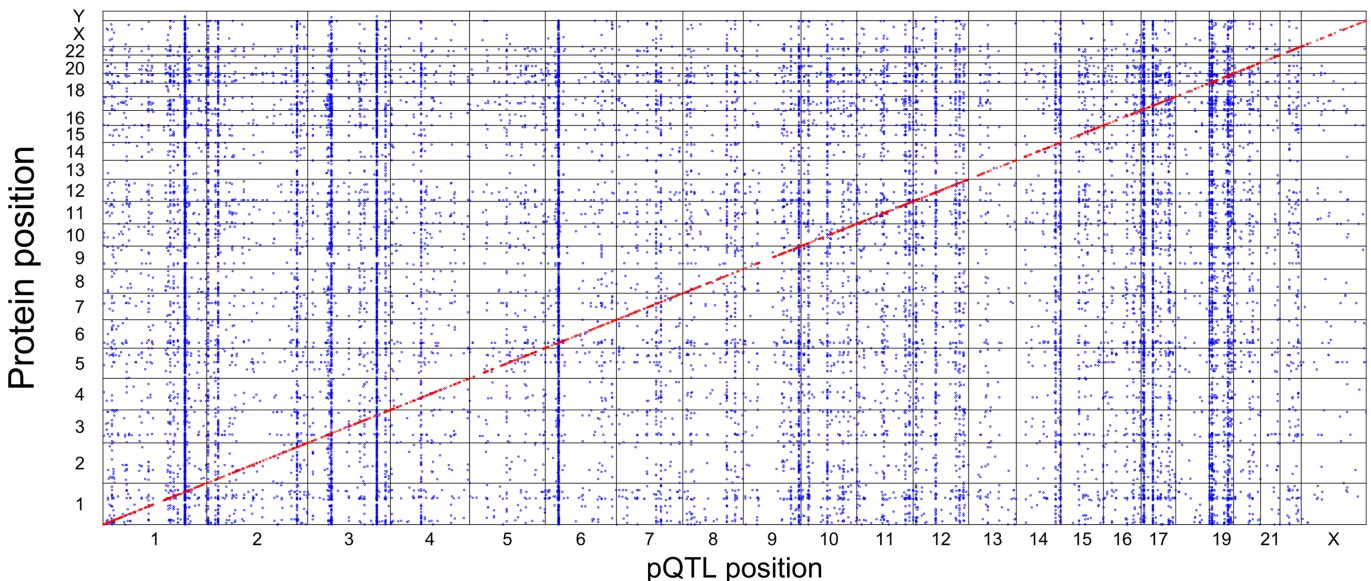

**Extended Data Fig. 4 | Genomic map of pQTLs.** Genomic locations of all sentinel pQTLs (*cis*, red; *trans*, blue) on the Olink platform (UKB-BI, top) and the SomaScan platform (Iceland 36K, bottom). The *x*-axis indicates the position of the pQTLs, and the *y*-axis indicates the gene encoding the protein with the associated levels.

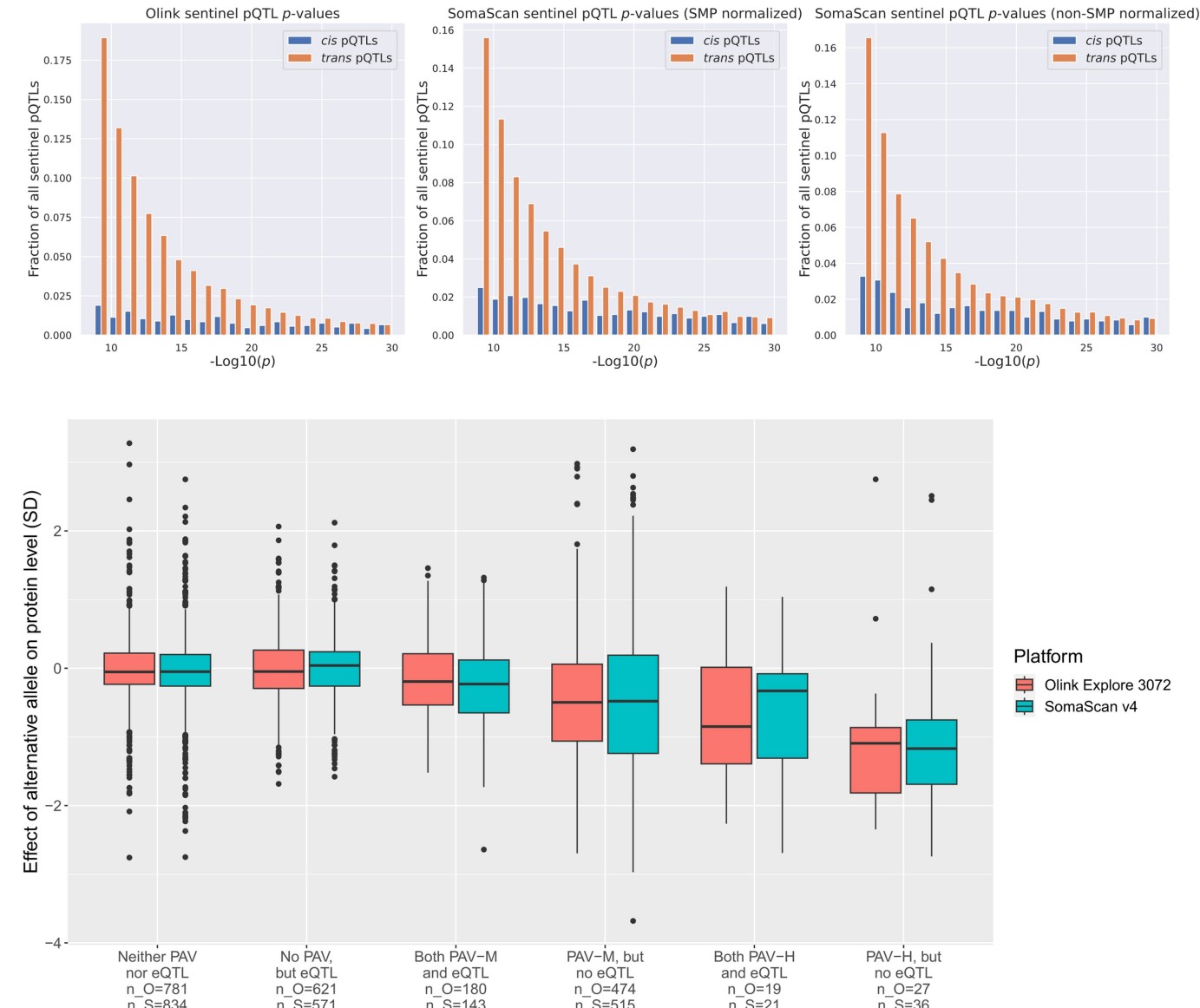

**Extended Data Fig. 5 | Significance of pQTLs and effect of alternative allele.**
Top: significance of detected pQTLs in the UKB-BI and Iceland 36K data sets.
For all platforms and populations, at the population size, a relatively much
higher number of *trans* pQTLs than *cis* pQTLs have significance close to the
threshold. *p*-values were based on two-sided significance tests and not
corrected for multiple comparsions. Bottom: Effect of alternative allele broken
down by presence or absence of PAV or *cis* eQTL in high LD. PAV-M: moderate
impact PAV, PAV-H: high impact PAV, n_O: number of *cis* pQTLs detected with
Olink, n_S: number of *cis* pQTLs detected with SomaScan. Box plots show
the median and lower and upper quartiles; whiskers extend to 1.5 times the
interquartile range; points beyond whiskers are plotted individually.

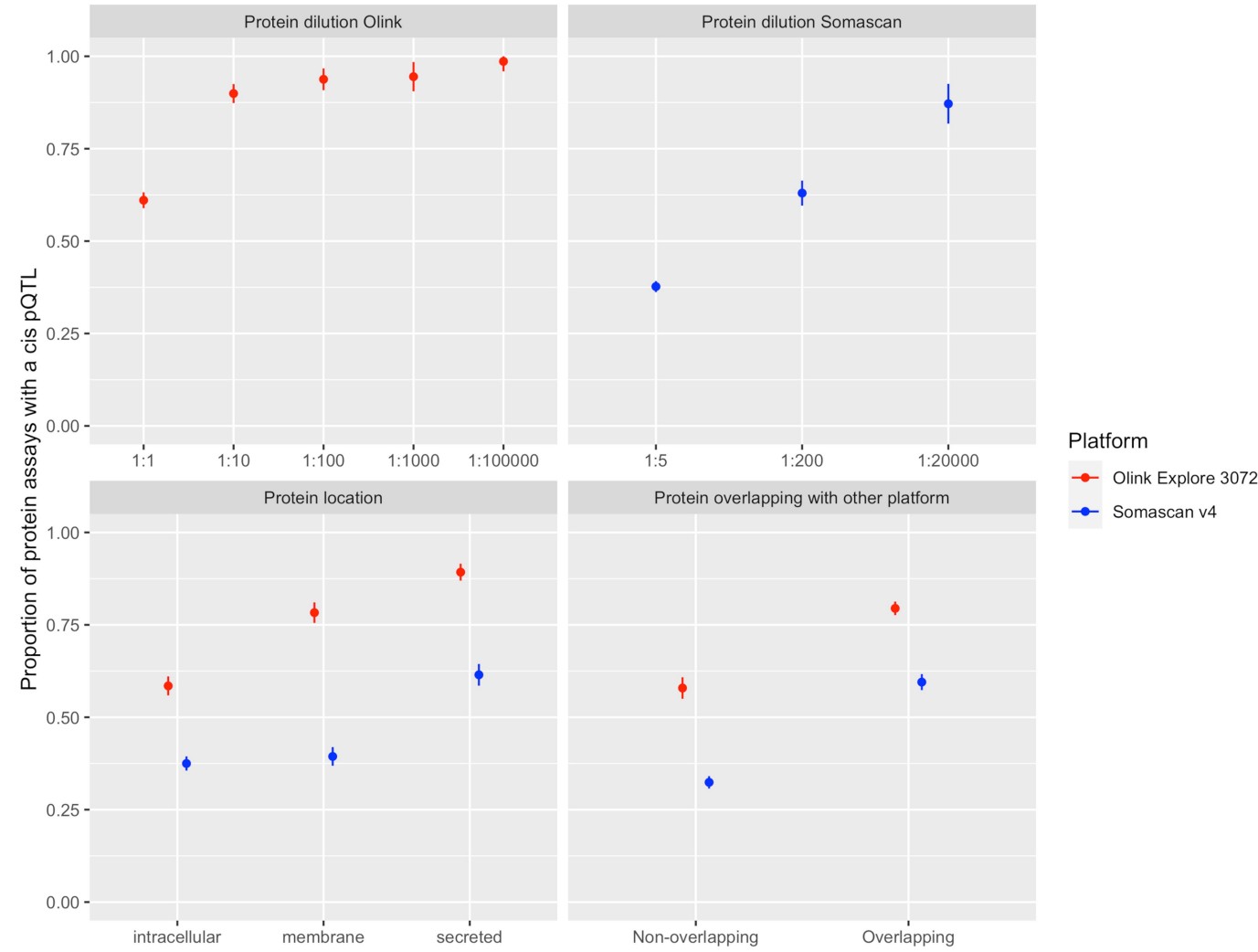

**Extended Data Fig. 6 | Proportion of protein assays that have a *cis* pQTL for subgroups of proteins defined by protein dilution, protein cellular location and overlap between platforms.** The plot show point estimates with 95% confidence interval. Centre points show proportion of *cis* pQTLs in each group. Top left panel based on 1964, 526, 257, 127 and 72 proteins with dilutions 1:1, 1:10, 1:100, 1:1000, and 1:1000 dilution, respectively. Top right panel based on 3981, 778 and 148 proteins with dilutions 1:5, 1:200 and 1:20000, respectively. Bottom left panel based on 1409 intracellular, 839 membrane and 698 secreted Olink proteins (red points); and 2419 intracellular, 1434 membrane and 1054 secreted SomaScan proteins (blue points). Bottom right panel based on 1100 Olink proteins non-overlapping and 1846 Olink proteins overlapping with SomaScan (red points); and 2951 SomaScan proteins non-overlapping and 1956 SomaScan proteins overlapping with Olink (blue points).

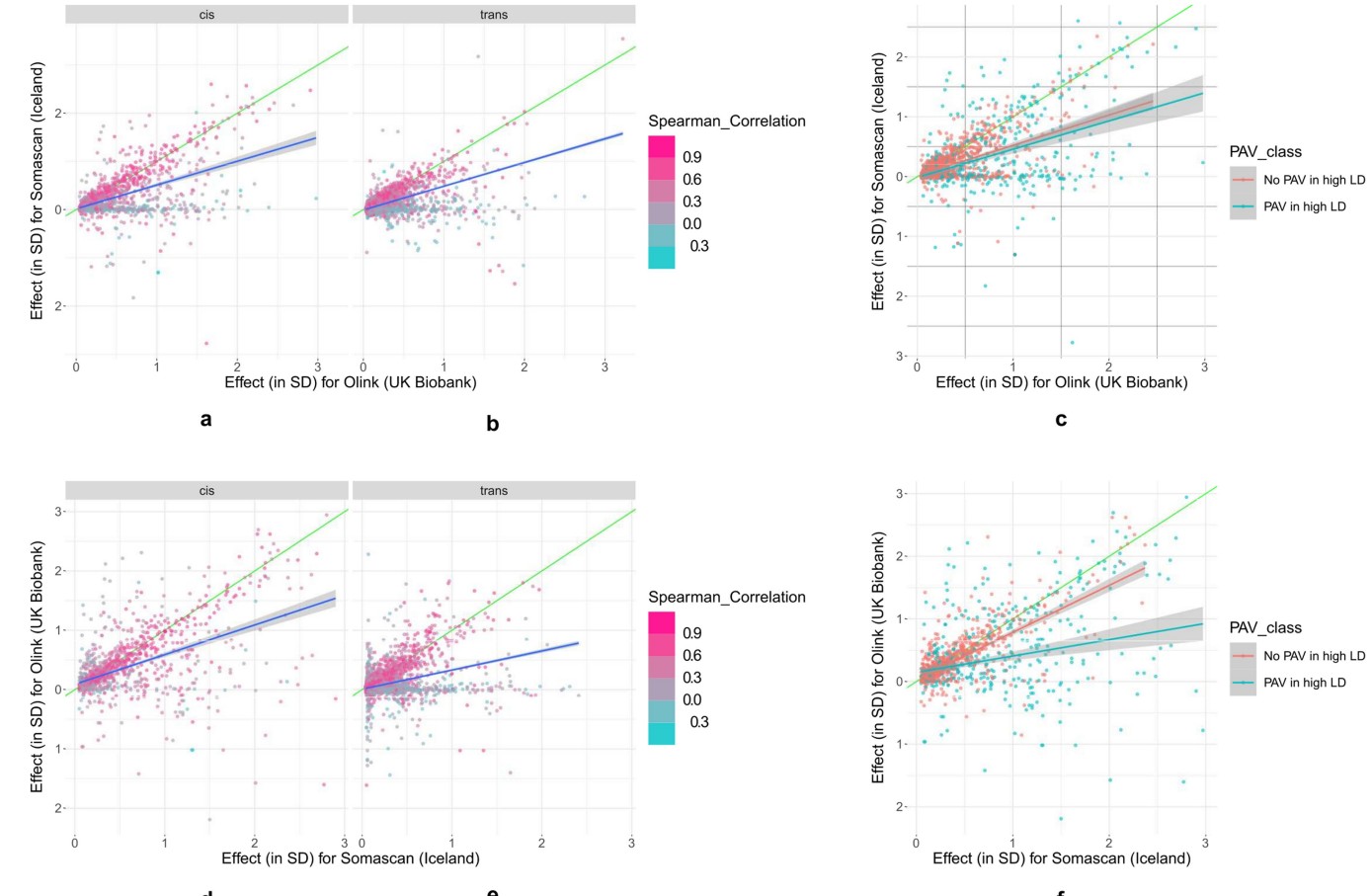

**Extended Data Fig. 7 | Replication of pQTLs between platforms.**

**a, b:** Replication of sentinel *cis* (a) and *trans* (b) pQTLs detected using Olink Explore (UK biobank) in normalized SomaScan v4 (Iceland) data. For each pQTL, the plot shows the effect (in units of SD) in SomaScan v4 (y-axis) vs the effect in Olink Explore (x-axis). The assays were matched on the UniProt ID of their targeted protein. Each point is colored based on the Spearman correlation between measured protein levels using normalized SomaScan v4 and Olink Explore. The green lines show values where the effect is equal based on SomaScan v4 and Olink Explore, while the blue lines show a linear regression estimate with shaded 95% pointwise confidence intervals. **c:** Replication of sentinel *cis* pQTLs detected using Olink Explore (UK Biobank) in normalized SomaScan v4 (Iceland) data, stratified on whether the *cis* pQTL is in high LD with PAV (red) or not (blue). For each pQTL, the plot shows the effect (in units of SD) in SomaScan v4 (y-axis)) vs the effect in Olink Explore (x-axis). Each point is colored based on whether or not the associated variant has a protein-altering variant in high LD ($r^2 > 0.80$). The green line shows values where the effect is equal based on SomaScan v4 and Olink Explore, while the blue and red lines show linear regression estimates with shaded 95% pointwise confidence

intervals for each group (PAV in high LD; No PAV in high LD). **d, e:** Replication of sentinel *cis* (d) and *trans* (e) pQTL associations detected using normalized SomaScan v4 (Iceland) in Olink Explore (UK biobank) data. For each pQTL association, the plot shows the effect (in units of SD) in SomaScan v4 (x-axis) vs the effect in Olink Explore (y-axis). Each point is colored based on the Spearman correlation between measured protein levels using SomaScan v4 and Olink Explore. The green lines show values where the effect is equal based on SomaScan v4 and Olink Explore, while the blue lines show a linear regression estimate with shaded 95% pointwise confidence intervals. **f:** Replication of sentinel *cis* pQTL associations detected using normalized SomaScan v4 (Iceland) in Olink Explore (UK biobank) data, stratified on whether the *cis* pQTL is in high LD with PAV (blue) or not (red). For each pQTL association, the plot shows the effect (in units of SD) in SomaScan v4 (x-axis) vs the effect in Olink Explore (y-axis). Each point is colored based on whether or not the associated variant has a protein-altering variant in high LD ($r^2 > 0.80$). The green line shows values where the effect is equal based on SomaScan v4 and Olink Explore, while the blue and red lines show linear regression estimates with shaded 95% pointwise confidence intervals for each group (PAV in high LD; No PAV in high LD).

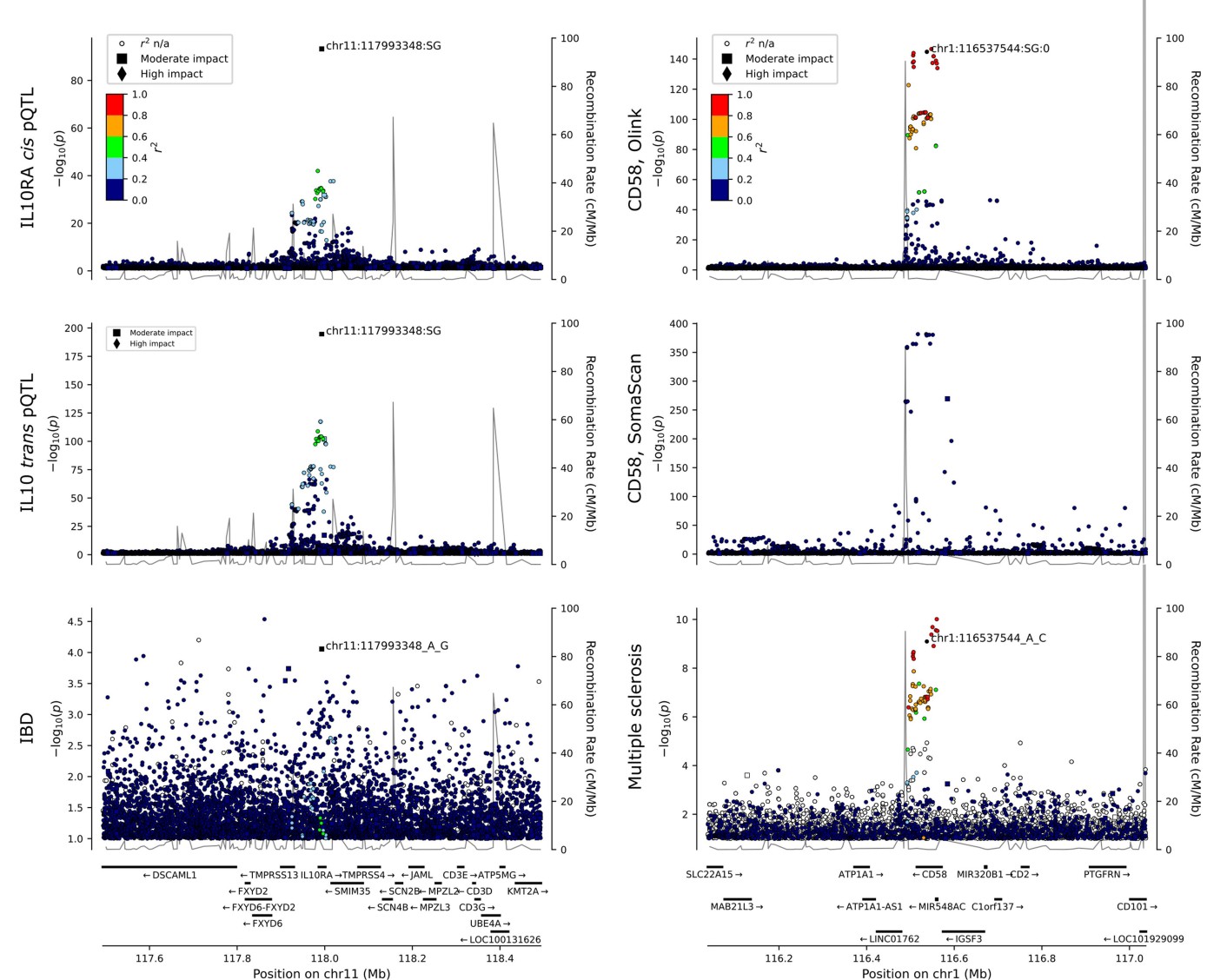

**Extended Data Fig. 8 | Association with protein levels and disease risk.**
Left: Association at *IL1ORA* locus between variants and protein levels of IL10RA
and IL10 measured using Olink Explore and IBD risk. All r² are shown to the same
variant. Right: Association at *CD58* locus between variants and CD58 levels
measured using Olink Explore, CD58 levels measured using SomaScan v4 and
multiple sclerosis risk. All r² are shown to the same variant. *p*-values were based
on a two-sided likelihood ratio test and not adjusted for multiple comparisons.

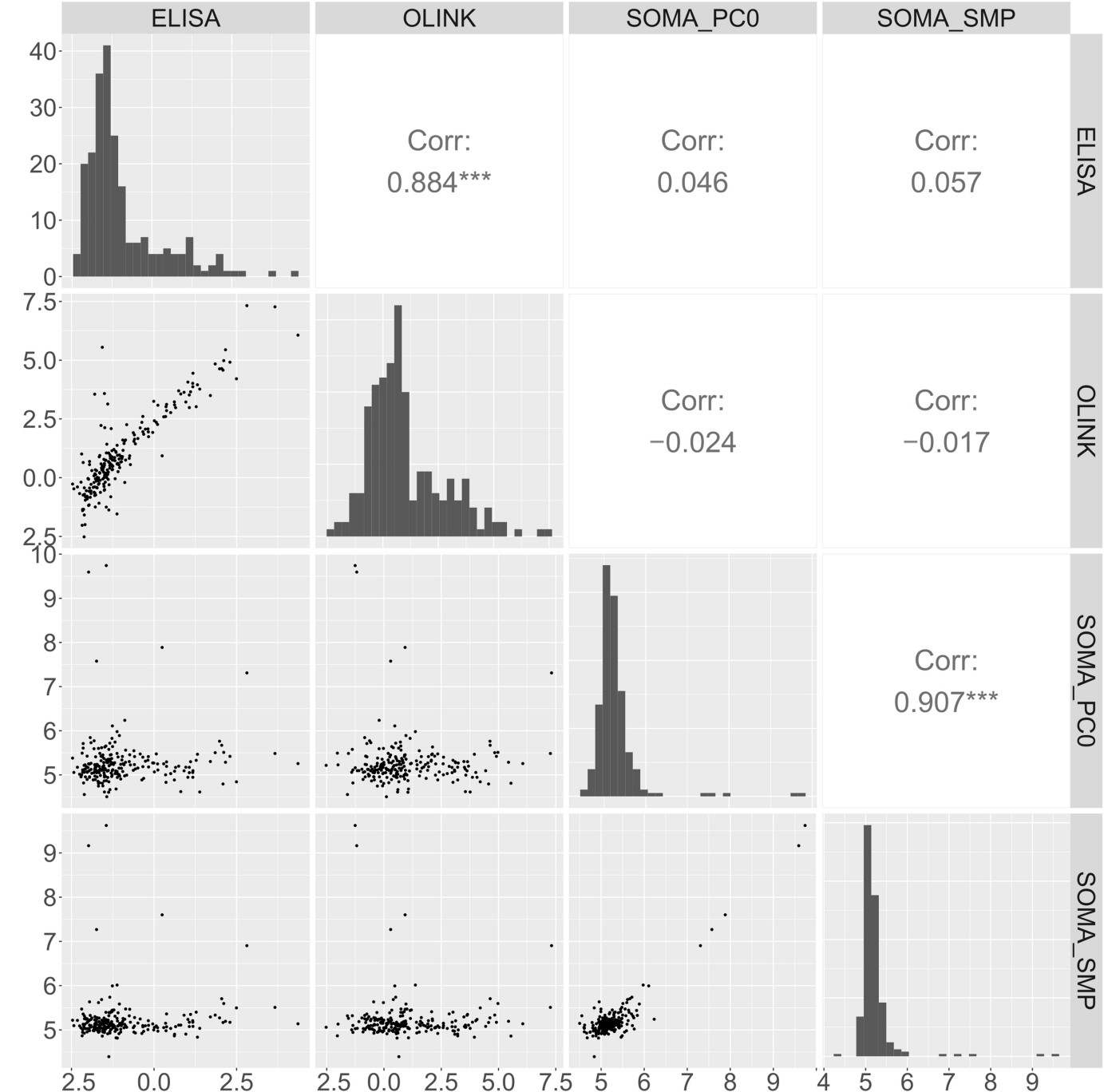

**Extended Data Fig. 9 | Comparison of IL10 measurements.** Protein levels of IL10 as measured by ELISA compared with measurements from Olink Explore 3072, SomaScan v4 non-normalized (SOMA_PC0) and normalized (SOMA_SMP). '***' represents p < 0.001, based on a two-sided t-test, not corrected for multiple comparisons. The exact p-values were $1.5 \times 10^{-75}$ for the correlation between OLINK and ELISA data and $1.1 \times 10^{-85}$ for the correlation between SOMA_PC0 and SOMA_SMP data.

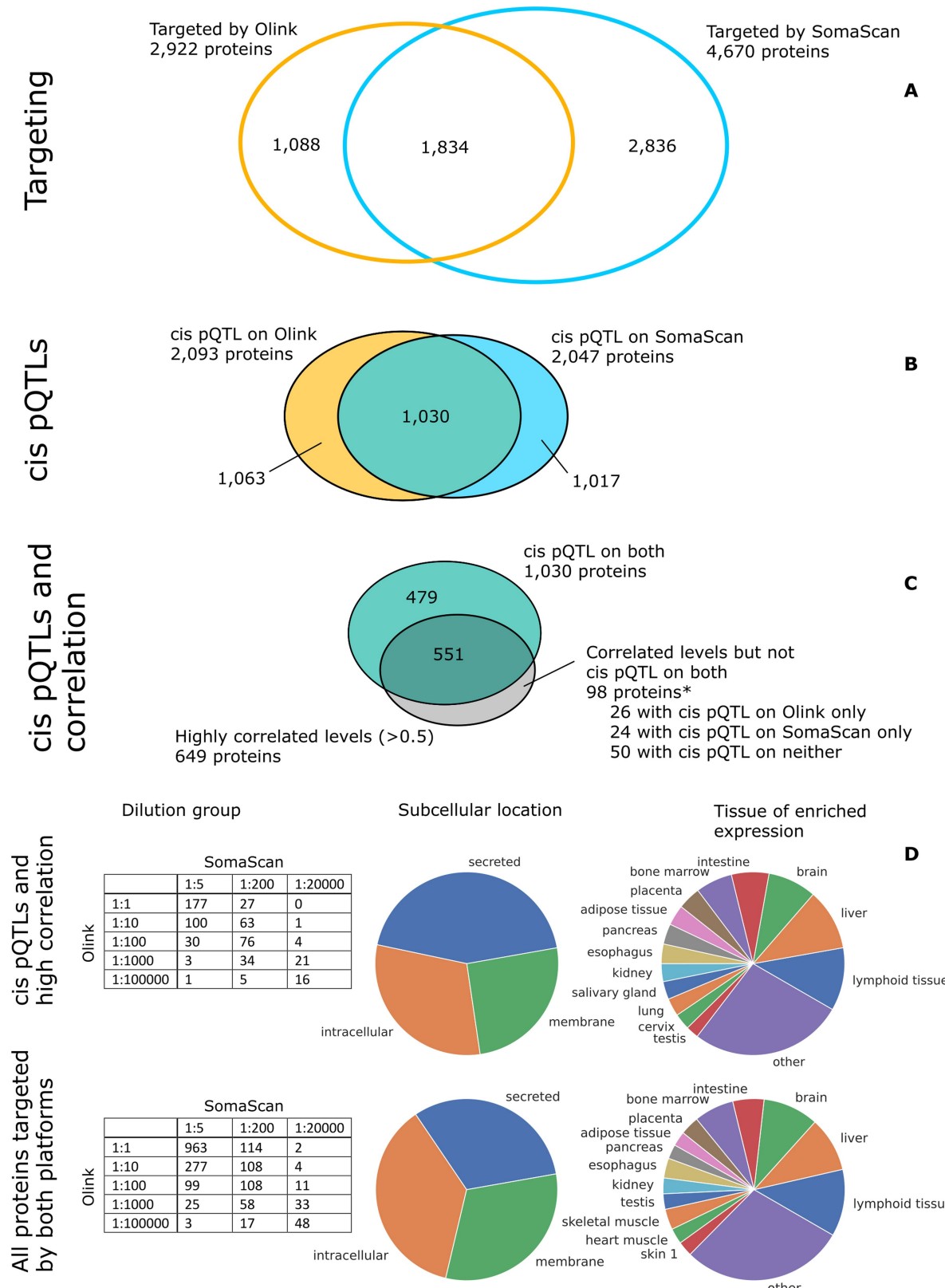

**Extended Data Fig. 10** | See next page for caption.

**Extended Data Fig. 10 | Using complementarity to assess the performance of assays.** The complementarity of the two platforms, along with the correlation of genomic information, can be used to assess the evidence for the targeting of the assays. **A)** The platforms target around 6,000 proteins in total, with about 2,000 proteins targeted by both platforms. **B)** *Cis* pQTLs provide evidence that about 2,000 proteins on each platform, with about 1,000 unique to each platform. **C)** For about 500 proteins that have a *cis* pQTL on both platforms, the correlation between levels measured using the two platforms is low. Supplementary Table 29 contains columns indicating presence or absence of *cis* pQTLs as well as the correlation between matching assays on the two platforms, making useful information to evaluate the performance of the assays easily accessible. Numbers in the figure refer to unique proteins, while the rows of the table correspond to pairs of assays. *As multiple assays targeting the same protein can differ in performance, the same protein may belong to more than one subset. **D)** Expected abundance (as reflected by dilution groups), subcellular locations, and tissue of enriched expression for the Tier 1 proteins (top) and all proteins targeted by both platforms (bottom).

# Reporting Summary

## Statistics

For all statistical analyses, confirm that the following items are present in the figure legend, table legend, main text, or Methods section.

| n/a | Confirmed | |
|---|---|---|
| ☐ | ☒ | The exact sample size (*n*) for each experimental group/condition, given as a discrete number and unit of measurement |
| ☐ | ☒ | A statement on whether measurements were taken from distinct samples or whether the same sample was measured repeatedly |
| ☐ | ☒ | The statistical test(s) used AND whether they are one- or two-sided *Only common tests should be described solely by name; describe more complex techniques in the Methods section.* |
| ☐ | ☒ | A description of all covariates tested |
| ☐ | ☒ | A description of any assumptions or corrections, such as tests of normality and adjustment for multiple comparisons |
| ☐ | ☒ | A full description of the statistical parameters including central tendency (e.g. means) or other basic estimates (e.g. regression coefficient) AND variation (e.g. standard deviation) or associated estimates of uncertainty (e.g. confidence intervals) |
| ☐ | ☒ | For null hypothesis testing, the test statistic (e.g. *F*, *t*, *r*) with confidence intervals, effect sizes, degrees of freedom and *P* value noted *Give P values as exact values whenever suitable.* |
| ☒ | ☐ | For Bayesian analysis, information on the choice of priors and Markov chain Monte Carlo settings |
| ☒ | ☐ | For hierarchical and complex designs, identification of the appropriate level for tests and full reporting of outcomes |
| ☐ | ☒ | Estimates of effect sizes (e.g. Cohen's *d*, Pearson's *r*), indicating how they were calculated |

*Our web collection on statistics for biologists contains articles on many of the points above.*

## Software and code

Policy information about availability of computer code

| | |
|---|---|
| Data collection | We used the following publicly available software for data collection in the sequence processing pipeline: BamQC (v1.0.0, https://github.com/DecodeGenetics/BamQC), RTG Tools (v3.8.4, https://github.com/RealTimeGenomics/rtg-tools), bcl2fastq (v2.20.0.422, https://support.illumina.com/sequencing/sequencing_software/bcl2fastq-conversion-software.html), To process data generated on the Olink platform: Olink Explore (v1.9.0, https://www.olink.com/products-services/data-analysis-products/npx-explore/) |
| Data analysis | We used the following publicly available software in conjunction with the algorithms described above. GraphTyper (v2.7.1, v1.4, v2.7.2, https://github.com/DecodeGenetics/graphtyper), GATK resource bundle (v4.0.12, gs://genomics-public-data/resources/broad/hg38/v0), Svimmer (v0.1, https://github.com/DecodeGenetics/svimmer), popSTR (v2.0, https://github.com/DecodeGenetics/popSTR), Admixture (v1.3.0, https://dalexander.github.io/admixture), Dipcall (v0.1, https://github.com/lh3/dipcall), Samtools (v1.9, v1.3, https://github.com/samtools/samtools), samblaster (v0.1.24, https://github.com/GregoryFaust/samblaster), BWA (v0.7.10 mem, https://github.com/lh3/bwa), |

GenomeAnalysisTKLite (v2.3.9, https://github.com/broadgsa/gatk),
Picard tools (v1.117, https://broadinstitute.github.io/picard),
Bedtools (v2.25.0-76-g5e7c696z, https://github.com/arq5x/bedtools2),
Variant Effect Predictor (release 100, https://github.com/Ensembl/ensembl-vep),
BOLT-LMM (v2.1, https://data.broadinstitute.org/alkesgroup/BOLT-LMM/downloads),
IMPUTE2 (v2.3.1, https://mathgen.stats.ox.ac.uk/impute/impute_v2.html),
dbSNP (v140, https://www.ncbi.nlm.nih.gov/SNP),
BiNGO (v3.0.3, https://www.psb.ugent.be/cbd/papers/BiNGO/Download.html),
Cytoscape (v3.7.1, https://cytoscape.org/download.html),
COLOC (v5.1.0.1, https://github.com/chr1swallace/coloc).

Data was analyzed and figures generated using Python (version 3.9.1), along with packages numpy (version 1.20.3), scipy (version 1.7.1), matplotlib (version 3.4.3), and pandas (version 1.3.0), and R (version 3.6.0).

The code central to power analysis and calculation of CV ratio can be found at https://github.com/DecodeGenetics/proteomics_comparison

For manuscripts utilizing custom algorithms or software that are central to the research but not yet described in published literature, software must be made available to editors and reviewers. We strongly encourage code deposition in a community repository (e.g. GitHub). See the Nature Portfolio guidelines for submitting code & software for further information.

## Data

Policy information about availability of data

All manuscripts must include a data availability statement. This statement should provide the following information, where applicable:
- Accession codes, unique identifiers, or web links for publicly available datasets
- A description of any restrictions on data availability
- For clinical datasets or third party data, please ensure that the statement adheres to our policy

WGS, genotype data, phased and imputed data for the UK Biobank data set, as well as the proteomics data, can be accessed via the UKB research analysis platform (RAP), https://ukbiobank.dnanexus.com/landing. The UK Biobank Resource was used under application number 65851.
The Icelandic genomic data and proteomics data have been described in our previous publication2.  While these individual-level data cannot be shared as dictated by the Icelandic law, we are open to collaborations on these topics, as we have been in the past.

GWAS summary statistics for all 2,931 Olink assays and all 4,907 SomaScan assays are available at https://www.decode.com/summarydata/.

Other data presented in this study are included in this publication (and its Supplementary Information).

URLs for other external data used are as follows:
the GWAS Catalog (https://www.ebi.ac.uk/gwas/),
the GTEx project (https://gtexportal.org/home/),
the Human Protein Atlas (https://www.proteinatlas.org/),
STRING database (https://string-db.org/, file name: 9606.protein.actions.v11.txt.gz),
UniProt (https://www.uniprot.org/).

## Research involving human participants, their data, or biological material

Policy information about studies with human participants or human data. See also policy information about sex, gender (identity/presentation), and sexual orientation and race, ethnicity and racism.

| | |
|---|---|
| Reporting on sex and gender | Self-reported sex of subjects was recorded at enrollment. Statistical analyses are adjusted for sex based on self-reported sex. |
| Reporting on race, ethnicity, or other socially relevant groupings | The UK Biobank data set is stratified by ancestry into subjects of British and Irish, South-Asian, and African ancestry, based on genomic information as described in Halldorsson, B. V. et al. The sequences of 150,119 genomes in the UK Biobank. Nature 607, 732–740 (2022). |
| Population characteristics | The population characteristics of the Icelandic data set have been described in Ferkingstad, E. et al. Large-scale integration of the plasma proteome with genetics and disease. Nat. Genet. 53, 1712–1721 (2021). The population characteristics of the UK Biobank have been described in Sudlow, C. et al. UK Biobank: An Open Access Resource for Identifying the Causes of a Wide Range of Complex Diseases of Middle and Old Age. PLOS Med. 12, e1001779 (2015). |
| Recruitment | See 'Population characteristics'. |
| Ethics oversight | All participants who donated samples to the Icelandic proteomics data set gave informed consent. The study was approved by the National Bioethics Committee of Iceland and conducted in agreement with conditions issued by the Data Protection Authority of Iceland (VSN_14-015). Personal identities of the participants were encrypted by a third-party system (Identity Protection System) approved and monitored by the Data Protection Authority. The scientific protocol and operational procedures of the UK Biobank were reviewed and approved by the North West |

Research Ethics Committee (REC Reference Number: 06/MRE08/65). Data for this study were obtained and research conducted under UKB application license numbers 24898, 68574 and 65851

Note that full information on the approval of the study protocol must also be provided in the manuscript.

# Field-specific reporting

Please select the one below that is the best fit for your research. If you are not sure, read the appropriate sections before making your selection.

☒ Life sciences          ☐ Behavioural & social sciences          ☐ Ecological, evolutionary & environmental sciences

For a reference copy of the document with all sections, see nature.com/documents/nr-reporting-summary-flat.pdf

# Life sciences study design

All studies must disclose on these points even when the disclosure is negative.

| | |
|---|---|
| Sample size | In Iceland, all individuals with available plasma samples have been analyzed by Somascan proteomics platform. The sample size in Iceland is comparable to the one used in the UK with Olink, where there majority of assays have cis pQTL association. Samples for measurement using the Olink platform in the Icelandic data set were chosen pseudo-randomly from those previously measured using the SomaScan platform. No statistical analysis was performed to choose sample size for that subset. |
| Data exclusions | We had two preestablished exclusion criteria. First, Non-human proteins were excluded because the aim of this study is to assess human biology. Second, deprecated assays according to the manufacturers of the Olink and SomaScan assays were excluded from the analysis. These assays are excluded from the beginning and are not taken into account in the counts. Assays mapping to multiple genes were excluded from the Somascan platform since when multiple proteins can be measured at the same time, it is difficult to classify associations as cis or trans. Individuals with evidence of incorrect labelling based on the comparison of their protein levels with genotypes and phenotypes were excluded from all the analyses. |
| Replication | We attempted replication in two different ways: a) between Olink and Somascan platforms and b) within Olink platform. We attempted to replicate pQTLs detected in the British and Irish subset of the UK Biobank data set using the Olink platform in Icelandic data set using the SomaScan platform and vice versa. We attempted to replicate the associations detected in UK Biobank using Olink on a subset of individuals with available plasma proteomics in Iceland also using Olink. |
| Randomization | We performed an observational study where we test association between protein level and sequence variants or phenotypes using two platforms and compare the results. There is no intervention in this study. The protein levels, genotypes and phenotypes are solely observed but never assigned as would be the case in an interventional study (e.g. clinical trial). Thus, randomization is not applicable to this study. |
| Blinding | The protein levels, genotypes and phenotypes were measured or assessed completely independently of each other. |

# Reporting for specific materials, systems and methods

We require information from authors about some types of materials, experimental systems and methods used in many studies. Here, indicate whether each material, system or method listed is relevant to your study. If you are not sure if a list item applies to your research, read the appropriate section before selecting a response.

## Materials & experimental systems

| n/a | Involved in the study |
|---|---|
| ☒ ☐ | Antibodies |
| ☒ ☐ | Eukaryotic cell lines |
| ☒ ☐ | Palaeontology and archaeology |
| ☒ ☐ | Animals and other organisms |
| ☒ ☐ | Clinical data |
| ☒ ☐ | Dual use research of concern |
| ☒ ☐ | Plants |

## Methods

| n/a | Involved in the study |
|---|---|
| ☒ ☐ | ChIP-seq |
| ☒ ☐ | Flow cytometry |
| ☒ ☐ | MRI-based neuroimaging |

