## [Peer Review File · Nature]

Manuscript Title: Comparison of large-scale plasma proteomics through genetics and disease

Reviewer Comments & Author Rebuttals

Reviewer Reports on the Initial Version:

Referees' comments:

Referee #1 (Remarks to the Author):

In this manuscript, Eldjarn and Ferkingstad et. al. describe associations between genetic variants and >1400 plasma proteins among UK Biobank participants, and compare their findings to previously reported genetic associations among the Icelandic population. This work represents a large-scale comparison of different proteomic platforms (affinity vs. aptamer), and highlights similarities and differences in their utility for novel genetic discovery. This work also identifies genetic variants which contribute to both circulating protein levels and risk of common diseases.

The manuscript is well organized and written and easy to follow. But it is very dense with results. There are many locations throughout, however, where the authors are cautioned to be more precise with their terminology, in particular with respect to reporting the results of hypothesis testing. Additionally, while the detail and rigor of many of the analyses and comparison conducted throughout the manuscript are appreciated, there are number of sections that should be moved to the supplement. These are indicated in the Minor Comments below as they affect the readability and flow of the paper but not the scientific interpretation of the work; nonetheless they are important comments. Finally, the manuscript seems to be lacking final conclusions.

The work is presented, and reads as, the tour de force of technical comparison between the two platforms that it is. The authors are to be commended for their detailed and rigorous comparison of the platforms. In the current presentation, however, the manuscript is little more than this and seems to offer few, if any, novel biological insights.

Major:

1. How do the differences in power relating to sample size between the UKB-BI and Icelandic cohorts, and assayed proteins between SomaScan v4 and Olink Explore, the later through its impact on multiple testing, affect the results of both the association testing and pQTL discovery?
2. A GWS threshold of $1.8e-9$ was used for the pQTL analysis based on the number of variants tested in the Icelandic data set. This is somewhat non-standard. What was used in the UKB-BI data and why? It is not clear in the methods but appears to be the same threshold in the results. (line 161). Given this is based on the number of variants in the Icelandic data set it seems inappropriate to apply to the UKB data. Additionally, using this approach, one would expect the threshold is also different between UKB subpopulations.
3. Was any attempt made to correct for number of proteins tested in the pQTL analysis? And why or why not?
4. The authors report ~10 fold more trans pQTL than cis pQTL for the Icelandic and UKB-BI analyses and ~3 fold fewer trans pQTL than cis pQTL for the UKB-AFR and UKB-SA populations, while the differences in the absolute numbers are likely due to power, not is not obvious that the ratio of cis to trans pQTL should differ as they do. Can the authors explain this?
5. What is the functional annotation of the common and rare variant cis- and trans-pQTL? It would be interesting to compare these rates both with respect to each other and across platforms.
6. The authors make some comparisons in pQTL discovery across genetic ancestry, noting examples of PCSK9 and HBBB pQTLs common among African-ancestry participants. These specific variants have been well recognized – were there novel ancestry-specific pQTLs that have not been previously reported?

7. The authors note that replication of immunoassay pQTLs identified in UKB-BI among Icelanders was lower than expected. Do the authors think this is due to differences in population structure? Is this a phenomenon unique to pQTLs or does it represent a larger difference between UKB-BI and Icelandic populations? In addition to the UKB-BI/Icelander findings, it would be helpful to know the degree of pairwise cross-ancestry replication within other combinations of UKB and Icelander populations (eg. what proportion of UKB-BI pQTLs are nominally identified in UKB-AFR/SAS and vice-versa; presenting a heatmap/matrix would be intuitive).
8. Line 263 – “Cis pQTLs on SomaScan and Olink and their colocalization with disease-associated variants”. In this section the authors report “colocalization” of cis pQTLs with disease-associated variants. As presented, it appears that the authors consider “colocalization” to represent the presence of a disease-associated variant that is in high linkage disequilibrium ($r^2 > 0.80$) with a sentinel pQTL. However, the “colocalization” terminology more conventionally refers to the formal statistical test of the same name, used to evaluate the presence of a shared casual variant influencing protein levels and disease risk (see PMID: 24830394, PMID: 34587156, PMID: 33536417). This “colocalization” terminology is subsequently repeated in the following section. The authors should formally test for colocalization (eg. using coloc, HyPrColoc, or similar tools) between pQTLs and disease-associated variants. Additionally, the association of the identified proteins with disease would be further strengthened by utilizing Mendelian Randomization based approaches.
9. Section “Proteins targeted by both platforms” – this section appears to focus on cis pQTLs uniquely identified by the immunoassay. Are there cis pQTLs identified uniquely by the aptamer-based method with meaningful disease associations?
10. More clarification about what is new, biological importance
11. Expand on implications for study of proteomics/genomics in general
12. The manuscript ends in a rather abrupt fashion. Limitations and Conclusions are needed.

Minor:

13. It is probably more accurate to say “for casual inference” than “to determine causality” on line 46.
14. Lines 52 to 55 are better suited for methods.
15. In the introduction the authors qualitatively describe differences in the proportion of intracellular/secreted proteins captured by each assay in comparison to the overall proteome reported in the Human Protein Atlas. Are the authors able to statistically compare these differences (Eg. by chi-square)?
16. The detail in lines 61-68 is probably better for the discussion while a short summary that previous cross replication attempts have had moderate success but been largely limited by sample size and the targeted nature of the Olink platform.
17. The results presented on page 5 are interesting, but a bit less detail is needed in the main text. Also the paragraphs would flow better in the following order “Comparison of precision..”, “Effect of dilution on precision..” and then “Between-platform correlation..”.
18. The authors use correlation and association interchangeable in many instances throughout the paper where better precision in language could be employed. For example on line 109-112 they use the term correlation to refer to the association between protein levels and both age and sex. While one could do a formal correlation with both, it seems more likely that linear and logistic regression were used rather than correlation methods. They should be more precise in the results and more detailed in the text.
19. A more important example of the above issues is on line 130-142 (paragraph on “correlations between levels of proteins measured using the two platforms...”) – the authors again refer to testing correlations between protein levels and incident heart failure and Alzheimer’s disease; however, as the authors are implying a causal association (eg. protein levels impact disease incidence), the variables are not interchangeable and should be reported as “associations” rather than “correlations.”
20. In the same analysis, it is best to avoid the use of prediction unless the authors have conducted AUC analysis. Rather they appear to simply be referring to magnitude of the effect estimate and should just say so.

21. In Figure 1, consider adding the number of sequence variants tested in the Iceland 1K population (presumably 33M?).
22. Given the differences in the analysis methods and thresholds used between the current analysis and the UKB-PPP analysis, which the authors point out, this comparison is informative but not really that interesting. It should be moved to the supplement.
23. One line 174-175 it is probably more accurate to say that the lead variant at some of the pQTLs had a higher MAF rather than just that the pQTL was more common.
24. In that the authors did not verify false discovery, they likely should refer to the section in Lines 190 as "theoretical" or "estimated" false discovery rates. This could also be moved to the supplement.
25. Line 246 – "Thus, even when cis pQTLs for a protein are detected by both platforms, they are not necessarily the same." The authors note that only 45% of cis pQTL are in high LD across platforms – can the authors speculate why this may be the case? Is this primarily driven by differences in population structure between UKB-BI and Icelanders, or due to differences in the assays?
26. Line 248 - Several pQTLs were highly pleiotropic. At these pleiotropic loci the authors may consider performing multi-trait colocalization using HyPrColoc (PMID: 33536417) to better estimate the number of proteins which truly share each pQTL.
27. Consider describing the proteomic platforms more generically as immunoassay- and aptamer-based methods (rather than by brand name) throughout.

Referee #2 (Remarks to the Author):

In "Comparison of large-scale immunoassay- and aptamer-based proteomics through genetics and disease" Eldjarn et al. describe the comparisons of Olink and SomaLogic data sets originating from the currently largest population studies. They anchor their investigations on the genetic associations of both platforms and cohorts and dissect their comparison of cis- and trans-pQTLs, ethnicities, phenotype associations, and features of the target proteins and assay characteristics. The work uses own data and those presented by the UKB Pharma project in a pre-print from Sun et al. (DOI: 10.1101/2022.06.17.496443). The manuscript is well written and sufficiently powered. It addresses an important aspect of connecting and comparing proteomics data obtained by different methods and independent cohorts based on genetic and phenotypic information.

Previous studies have compared the leading affinity proteomics methods at a more focused and single cohort scale. This includes the cited work by Pietzner et al. (Ref 7), and recent work by Katz et al. (not cited; DOI: 10.1126/sciadv.abm5164), both highlighting similar differences between the technologies. In the latter work, a focus was on accuracy and precision, leading to an assessment that "protein target specificity and phenotypic associations were better for the Olink platform, but measurement precision and proteome coverage were better for SomaLogic". Given the scale of the presented study, a bigger focus should therefore be set on aspects that cannot be accurately investigated in underpowered studies, such as the trans-pQTLs.

Concerns to be addressed:

While SomaLogic's approach uses a slow-off rate and a single binder selected due to its capabilities of holding on to an intended target, the Olink approach uses two complementary antibodies that reveal a signal when the antibodies bind a common target of interest (in proximity). Besides high-performance reagents, the stringency of an assay's specificity depends on two factors that govern the accessibility to the target of interest:

- 1) Aspects relating to the target proteins themselves - such as abundance, structure, interaction, or modifications.
- 2) Aspect relating to external factors (sample matrix), such as abundance and frequency of other proteins with similar epitopes.

Since neither Olink nor SomaLogic provides data on the actual epitopes and interfering factors,

other means are needed to infer sources of off-target binding. It would be highly interesting to investigate if the available genetic data be used to gain more insights into off-target interference, e.g., by studying the pleiotropy or trans-pQTLs that won't match expected pathways or processes.

As observed by the authors, the two platforms concur better for higher abundant proteins measured in diluted samples. This can be explained by a lower complexity of the sample matrix that leads to a decrease in external interference factors while target protein-related factors become dominant. It is known that the more abundant plasma proteins are secreted (DOI: 10.1126/scisignal.aaz0274). Hence there is an intrinsic bias in why secreted proteins appear to perform better. The authors should therefore evaluate their data based on the organ that serves as the main source of the proteins in the blood. This could be very informative when connecting the blood protein data to specific diseases.

As another useful outcome, the authors should list and rank all proteins they deem comparable on both platforms. Hence they could be annotated as platform-independent protein assays. These targets should become known to the community so they can be used and prioritized in studies aiming to expand on the current proteomics datasets. The Sun et al. work mentioned expanding on their work by using MS to quantify proteins in an affinity-free manner. Such efforts will likely inform about some of the high confidence protein assays, while lower abundant proteins will less likely be reached from the get-go.

In the section following line 143, the authors discuss NFL as a case with inverse directions of the effect. Did these two assays overlap in their pQTLs or differ in assay performance? In a previous study, Hong et al. presented a similar case where two antibodies were used in the same multiplexed assay against the abundant protein HRG. They found effects of PAVs, inverse trends of genetic associations, and consequently, low correlation of the data with differences in phenotypic association (DOI: 10.26508/lsa.202000817). Hence, the NFL case would benefit from more details to understand why differences have been observed.

Beyond looking at proteins 1-by-1, other means to compare data are by investigating the inter-protein correlation of all proteins in each platform. Did the authors observe any shift from zero, and if particular proteins or dilution sets present differently? Considering that proteins with common pQTLs do not always correlate well on a protein level, it would be interesting to test why this is so. Would, on the other hand, highly correlating proteins always reveal common qQTLs? Can the authors simulate a protein correlation threshold at which the genetic association becomes distinct from another or common for both?

One concern is using multiple $N=2$ comparisons to determine CV values for each platform. Even though a large number of comparisons ($N > 400$) gives some confidence, it misses judging variability in repeated measurement of single samples with $N > 3$ and preferably across multiple batches of the data. Batch size and numbers have likely influenced the outcome of the analysis that - partly - contradicts those made by Katz et al.. No longitudinal samples have been studied to determine the variability of resampling, which might be an even more important aspect when considering where the value of proteomics may lie over genomics. The authors should also perform inter-class correlations between the duplicated samples to evaluate the shift from the line of identity and discuss how the scales of the two methods affect their assessments.

Another informative measure to judge protein-level data is to investigate the heterogeneity of variance. The authors should present the variance from Olink and SomaLogic data for a few representative cases with concordant data and pQTL mapping. The authors should also discuss how differences in scales affect these measures.

The authors present and discuss the SomaLogic as normalized and unprocessed. Even though this is a relevant topic, it distracts from the core comparison and should therefore be relocated to the supplementary. Olink data can also be processed and used as inverse transformed data, including

or excluding data below LOD. This possibility has though not been covered.

Instead, the authors should present how pre-analytical factors, such as age-of-sample, influence the two assays. More interestingly, can the authors investigate how the data comparisons would appear if both data sets were adjusted by age, sex, and BMI?

A core of the comparison is understanding why differences and commonalities were observed. The presented study does not provide further insights or molecular evidence into such aspects. Using an antibody-based dual binder ELISA to validate the exemplified target is a good effort. Still, it favors one platform over the other due to the nature of the assay and affinity reagents used. Enriching proteins from plasma via a pull-down is an alternative strategy to corroborate affinity binding, but caution has to be taken considering common contaminants that will co-precipitate. It is worth discussing that the rebinding effects occur when surface-bound affinity enrichment, such as for SomaLogic's assays, is being used and that this differs from in-solution binding, such as those used in Olink's proximity assays.

Other points:

Many figures were pasted into the documents at a highly inconvenient resolution. It was difficult to judge these on correctness and detail. Instead of Excel, all figures should be reproduced in the more appropriate software, like R.

Line 26: revise to "1,075 samples were also..."

Referee #3 (Remarks to the Author):

Summary of the key results

The authors present a technical tour de force comparing the currently most comprehensive assays to measure plasma proteins, namely the SomaScan v4 platform and the Olink Explore 1536 platform. They present consistency and divergence of both technologies based on a large set of phenotypic and genetic associations in more than 30k participants, representing the largest study currently performed. They highlight technical factors to account for missing replication of phenotypic and genetic associations, and further, that while the Olink assays seems to provide better performance for targeted proteins, the SomaScan platform distinguishes by the larger protein coverage. While these results are interesting, they do not really represent an advance in our understanding of how those protein technologies can inform biology or clinical translation of genetic findings.

Validity

The conclusions drawn from the study are mostly well justified by the data presented. My greatest concerns are:

- 1) The presentation of cis-pQTLs as orthogonal validation for target specificity sounds obvious, cis-pQTLs may equally likely strongly affect assay performance and it is currently unclear to which extend this explains the reproducibility of phenotypic associations.
- 2) The authors, correctly, claim that cis-pQTLs can help establish causality, but present only a rather simple cross-referencing exercise to variants reported in the GWAS catalog. This is one of essential messages of this paper and greatly determines its value to the scientific community. The authors would need to present more rigorous statistical tests, including statistical colocalization and Mendelian randomization to validate their claims.

Further, it is unclear how replication of pQTLs across assays has been defined when comparing between DeCODE and UKB.

Minor:

- The reporting of pQTL fractions assumes that the additional proteins on the SomaScan assay would be similarly beneficially measured using Olink, which is unlikely to be the case, pls report such fractions only for matching pairs of proteins

Originality and significance

While this work is certainly the largest and most comprehensive of its kind, the conclusions drawn from the data largely align with what has been reported previously (e.g., Raffield et al. *Proteomics* 2020; Pietzner et al. *Nature Communications* 2021; Katz et al. *Science Advance* 2022), but the authors sometimes miss to acknowledge those findings.

It is further unclear to me, why this work has been submitted separately and was not combined with what has already been published by the UKB-PPP. It appears highly redundant to have both papers separately. The method comparison, ancestry-specific analysis and a substantially improved disease follow-up would have addressed the major shortcomings of the work by Sun et al. (Sun et al. *BioRxiv* 2022) and would in its totality, as a single paper, provide the conceptual advance that both papers are currently lacking.

Data & methodology

While the authors have certainly access to the largest data set to compare both platforms, the influence of differences in sample handling and study characteristics between UKB and DeCODE is not sufficiently presented, investigated and discussed, which strongly limits the conclusion drawn from the data.

Appropriate use of statistics and treatment of uncertainties

Statistical analyses have been overall well defined and executed. However, as already outlined above, linkage between protein and phenotypes using genetic instruments is certainly a subject that needs improvement. For example, the term 'colocalization' is used without proper justification of a statistical test.

The comparison between this study and the previous UKB-PPP preprint is rather confusing, as both analyze the very same protein data and I think it needs more information on why a surprisingly large fraction of pQTLs do not agree (using large genomic, $\pm 2\text{Mb}$, is a poor definition of replication). What role do rare variants play? How do those differences affect phenotypic investigations? What explains the massive difference in trans-pQTLs (16k vs 9k)?

Please provide coloc pairs of protein and phenotype by unique variant or highly related cluster thereof and further separated by cis and trans to judge the impact of pleiotropic variants. Further, were previous proteomic studies excluded from the GWAS catalog data? It is particularly irritating that the authors begin the section correctly noting the evidence for cis but the vast majority of the presented numbers afterwards relates to trans associations.

A fundamental statistical difference between the UKB-PPP and the present effort is the identification of secondary signals. Although the authors use the same method as previously described, I have some concerns how valid exact conditional analysis in the context of massive effect sizes work. How strongly are secondary signals correlated with sentinel signals, and how much do marginal effect sizes drop once the authors compute a joint model for the entire locus? Given that many downstream analyses using the provided data will include a MR framework of

some sort, validity of instrument selection is an important issue.

Minor:

- I do not understand the very first sentence of the results. Isn't the CV already a measure of precision? This data is also somewhat contradicting to a recent study by Katz et al. (Science Advance 2022) who showed higher precision of the SomaScan assay and it remains unclear how much differences in sample handling and cohort characteristics might have contributed to this data. How does this comparison look within DeCODE only?
- Please do not use the term 'predicted' for effect estimates in Cox-models as it is not clear whether protein measurements really help to identify more future patients.
- When reporting the fraction of trans-pQTLs, it would be important to understand how large the contribution from unspecific trans-pQTLs might be, that are the sole associations for possibly very many SOMAmers.

Conclusions

The conclusions drawn by the authors are mostly well justified. I would, however, suggest refining the statement about orthogonal validation of protein targets by cis-pQTLs. Strong cis-pQTLs might very well obscure protein measurements as has been shown previously (Pietzner et al. 2021).

Suggested improvements

- 1) A general weakness of the study are the limited insights from these amazing data sets. Further analysis on 1) what characterizes genuine pQTLs, 2) how can differences between both assays highlight relevant biology, or 3) affect downstream phenotypic studies would be needed to provide an advance in our understanding in which situations to trust which assay, but more importantly how those assays can yield biologically and clinically relevant insights.
- 2) The availability of different ancestry is a clear strength of this study, but the current presentation of results remains superficial showing only two well-known examples. This section can certainly improve.
- 3) Although the authors compare the two largest sets available, some of the difference might be driven by ancestry-specific effects between deCODE and UKB, and comparing UKB-PPP results to a matching European study with SomaScan might help to understand those differences better.
- 4) A major message of this study is that the higher coverage of the SomaScan assays is its surplus. With another 1.5k proteins on the horizon for UKB, to which the authors will have preferential access, I am wondering whether some of the conclusions drawn here will still hold and whether it might not be better to wait and have a fairer comparison also in terms of coverage.
- 5) Some text of the Supplemental Notes is repetition from the main text.

Minor:

- Many figures are of poor quality or miss proper legends. For example, the black background on figure 4 does not really help to distinguish points or figure 5 should rather be a stacked locus zoom plot instead of a 2x2 panel.

References

References look appropriate, although the authors seem to overly emphasize their own previous work.

Clarity and context

The authors have done a good job in pulling many statistical analyses together. However, this is also a general weakness of the work, as the reader is left with this number without meaningful interpretation. Most importantly, to really represent a conceptual advance this work needs to dig

deeper into phenotypic and ancestry-specific links towards the proteins investigated.

1) The presentation and discussion around the normalized vs non-normalized data for SomaScan is somewhat vague and would benefit from a clearer presentation. At least to me, it looks like presenting data on normalized results only and discussing non-normalized results in the Supplement would improve the paper.

Author Rebuttals to Initial Comments:

Note that we have renumbered the points raised by the referees (referee number.question number, ex. Q1.1). Where the referees included their own numbers, they have been kept but with added parentheses (ex. Q1.4 (1.)).

Referee #1 (Remarks to the Author):

In this manuscript, Eldjarn and Ferkingstad et. Al. describe associations between genetic variants and >1400 plasma proteins among UK Biobank participants, and compare their findings to previously reported genetic associations among the Icelandic population. This work represents a large-scale comparison of different proteomic platforms (affinity vs. aptamer), and highlights similarities and differences in their utility for novel genetic discovery. This work also identifies genetic variants which contribute to both circulating protein levels and risk of common diseases.

Q1.1 The manuscript is well organized and written and easy to follow. But it is very dense with results. There are many locations throughout, however, where the authors are cautioned to be more precise with their terminology, in particular with respect to reporting the results of hypothesis testing.

We thank the reviewer for their encouragement. We have tried our best to follow the reviewer comment of being more precise in terminology.

Q1.2 Additionally, while the detail and rigor of many of the analyses and comparison conducted throughout the manuscript are appreciated, there are number of sections that should be moved to the supplement. These are indicated in the Minor Comments below as they affect the readability and flow of the paper but not the scientific interpretation of the work; nonetheless they are important comments.

We have followed the reviewer's advice and moved several sections to the Supplement to improve readability.

Q1.3 Finally, the manuscript seems to be lacking final conclusions. The work is presented, and reads as, the tour de force of technical comparison between the two platforms that it is. The authors are to be commended for their detailed and rigorous comparison of the platforms. In the current presentation, however, the manuscript is little more than this and seems to offer few, if any, novel biological insights.

We have now fully rewritten the discussion and conclusions and have adapted them to reflect the updated analysis using Olink Explore 3072 and the changes based on the reviewers' comments. Throughout the text we have added further examples to enhance the biological insights offered by the paper.

The discussion covers the following points.

1. In total, 5,814 proteins are targeted by at least one of the two platforms. Of these, we detected *cis* pQTL associations is detected in plasma for 3,129 on at least one of the platforms. Specifically, a total of 2,093 out of 2,925 proteins on Olink and 2,044 out of 4,716 proteins on SomaScan have *cis* pQTLs on their respective platforms, indicative of the platform successfully detecting and measuring the protein in plasma. In total, 54% of the

proteins targeted by at least one platform have a *cis* pQTL. This suggests that not all proteins currently targeted by the platforms can be easily measured in plasma. A number of proteins, in particular intracellular proteins, may have too low concentration in plasma to be detectable. Non-secreted and less abundant proteins are less likely to have *cis* pQTLs and they have lower correlation between platforms. Given the enrichment of secreted and abundant proteins in the collection of proteins currently targeted by the platforms, we would expect proteins targeted by neither of the two platforms, which are largely non-secreted or less abundant, to be even more challenging to measure in plasma.

2. Among the 1,848 proteins targeted by both platforms, there was a larger fraction of *cis* pQTLs on Olink than on SomaScan (80% vs 60%). By assessing the direction of effect of the alternative alleles we find no evidence that the *cis* pQTLs detected on the two platforms were differentially affected by binding artefacts (section *Cross-platform replication*, page 13). This suggests that the Olink platform more often detects and measures these proteins in plasma than the SomaScan platform.
3. In cases where *cis* pQTLs provide evidence that both platforms are measuring the targeted protein, the variants driving the associations are often different. This diversity in pQTLs is consistent with diverse proteoforms being differentially targeted by the two platforms and the two platforms being affected differently by artefact associations.
4. Overall, correlation of protein levels between platforms is modest with a median of 0.33 and two modes centered at 0.0 and 0.6. The differences between the plasma protein levels and pQTLs between platforms affect the integration of protein level measurements into the study of diseases.
5. Leveraging the diverse ethnical composition of the UK Biobank population, we detected associations between protein levels and ancestry-specific variants. In addition, the associated variants are on average in high LD with 37 variants in the European ancestry group compared to 12 variants in the African ancestry group. The lower average linkage disequilibrium in individuals of African ancestry allows us to demonstrate substantial refinement of association signal across ancestries.
6. Around a third of pleiotropic trans pQTLs on one platform are not pleiotropic in the other. This may be caused by sequence variants interacting with pre-analytical variables, including sample processing, or with the protein measurements themselves.
7. Proteins with *cis* pQTLs in plasma on one platform are good candidates for addition to the other platform. For each of the two platforms, we identified between 1,000 and 1,500 additional proteins/proteoforms which should be measurable based on the other platform.
8. While *cis* pQTLs provide strong evidence that the protein being measured is in fact encoded by the gene of interest, they do not indicate which proteoform is being measured. In more than half of the cases where *cis* pQTLs were observed on both platforms, the sentinel variants are not in high LD. This can be either because the assays are measuring different proteoforms, or the measurements are affected by different artefacts between the platforms.
9. Some *cis* pQTL on both platforms might correspond to binding artefacts rather than protein levels, particularly when this *cis* pQTL corresponds to a coding variant and in the absence of a correlated eQTL. We conclude that *cis* pQTL are equally likely to be artefactual on both platforms.

10. The current work is limited to plasma samples and the analysis of other sample types using these platforms will need separate validation. Thus, measurement evidence for other sample types, such as CSF, would be beneficial.

Major:

Q1.4: (1.) How do the differences in power relating to sample size between the UKB-BI and Icelandic cohorts, and assayed proteins between SomaScan v4 and Olink Explore, the later through its impact on multiple testing, affect the results of both the association testing and pQTL discovery?

To address the reviewer's question, we performed a power analysis showing that reducing the sample size in the UK biobank to be equal to the sample size in Iceland ($n=35,892$) would be expected to result in the detection of slightly fewer sentinel *cis* pQTL and one third fewer *trans* pQTL associations: if the sample size in UKB were 35,892, we would expect to detect 2,021 of the sentinel *cis* associations (96% of the 2,102 detected in the full set of individuals), and 16,399 of the sentinel *trans* associations (66% of the 24,824). When restricting to assays measuring proteins targeted by both platforms, we would expect to detect 1,423 of the sentinel *cis* associations (97% of 1,468), and 12,315 of the sentinel *trans* associations (66% of 18,578).

The comparisons we performed based on pQTLs focused on *cis* pQTLs, the difference in sample sizes between Iceland and the UKB would therefore not impact our conclusions substantially since the fraction of assays with *cis* pQTLs on Olink in the UK on the overlapping set would go from 80% to 78% compared to 60% on Somascan in Iceland. We have added this result to the Supplementary Note S6 and are quoting it in the main text (section *Detection of pQTLs*):

"We performed a power analysis showing that reducing the sample size in the UK Biobank to be equal to the sample size in Iceland ($n=35,892$) would be expected to result in the detection of 3% fewer sentinel *cis* pQTL and 34% fewer *trans* pQTL associations."

Furthermore, we estimated the pQTL false discovery rate taking into account the number of assays and variants tested in each group as 1.2% and report on page 10 of the manuscript.

Q1.5: (2.) A GWS threshold of $1.8e-9$ was used for the pQTL analysis based on the number of variants tested in the Icelandic data set. This is somewhat non-standard. What was used in the UKB-BI data and why? It is not clear in the methods but appears to be the same threshold in the results. (line 161). Given this is based on the number of variants in the Icelandic data set it seems inappropriate to apply to the UKB data. Additionally, using this approach, one would expect the threshold is also different between UKB subpopulations.

We test 33.5M variants in Iceland which is about 40% fewer than in UKB (57.7M). The difference is largely due to very rare variants. However, the difference between them would result in a multiple testing correction threshold in UK of 8.7×10^{-10} instead of 1.8×10^{-9} . A total of 153 associations (1%) of the *cis* pQTLs are between those two thresholds and 1,608 (5%) of the *trans* pQTLs. This information has been added to the Methods section of the manuscript.

The number of variants also varies between ancestral groups in UKB. However, this would also not result in very different thresholds (6.9×10^{-10} AF, 1.2×10^{-9} SA).

Q1.6: (3.) Was any attempt made to correct for number of proteins tested in the pQTL analysis? And why or why not?

Correcting for the number of tested proteins would render the comparison asymmetrical as the SomaScan platform measures roughly 1.7 times the number of proteins that the Olink platform measures. Instead of correcting for the number of proteins we estimated the false discovery rate (FDR) based on our results as mentioned in point Q1.4 above and found it to be 1.2% for both platforms.

Q1.7: (4.) The authors report ~10 fold more trans pQTL than cis pQTL for the Icelandic and UKB-BI analyses and ~3 fold fewer trans pQTL than cis pQTL for the UKB-AFR and UKB-SA populations, while the differences in the absolute numbers are likely due to power, not is not obvious that the ratio of cis to trans pQTL should differ as they do. Can the authors explain this?

As previously observed by us and others, *cis* pQTL associations are on average much more significant than *trans* pQTL associations (Sun 2018: *cis* median $p=2 \times 10^{-72}$, *trans* median $p=7 \times 10^{-42}$; Extended Data Figure 7), meaning that with smaller sample size, relatively fewer *trans* pQTLs are detectable than *cis* pQTLs.

Q1.8: (5.) What is the functional annotation of the common and rare variant cis- and trans-pQTL? It would be interesting to compare these rates both with respect to each other and across platforms.

For each pQTL association, we assessed if the variant was in high LD with a protein altering variant or a *cis* eQTL for the neighbouring genes (within 1Mb) and have provided this annotation in the pQTL tables (Supplementary Tables S10-S12). For *cis* pQTLs, we have previously shown that the presence of PAV and absence of *cis* eQTL in high LD for the gene encoding the targeted protein possibly reflects protein binding artefacts. We had already included the number and fraction of PAV and *cis* eQTL for the gene of interest for detected *cis* pQTL in Table 1.

For *trans* pQTL associations it is not clear which gene at the location of the variant should be assessed for PAV or *cis* eQTL. If considering all the neighbouring genes within 1Mb, *trans* pQTL are less likely to be in high LD with a PAV or *cis* eQTL on both platforms.

Based on SomaScan, 40% of variants with *cis* pQTL and 28% of variants with *trans* pQTL are in high LD with a PAV ($r^2 > 0.80$), and 44% of variants with *cis* pQTL and 38% of variants with *trans* pQTL are in high LD with *cis* eQTL ($r^2 > 0.80$).

Based on Olink, 39% of *cis* pQTL variants and 23% of *trans* pQTL variants are in high LD with a PAV ($r^2 > 0.80$), and 47% of *cis* pQTL variants and 41% of *trans* pQTL variants are in high LD with a *cis* eQTL ($r^2 > 0.80$).

Thus, when considering the neighbouring genes within 1Mb, *cis* pQTLs are more likely to be in high LD with a PAV or *cis* eQTL on both platforms compared to *trans* pQTLs. When we restricted the analysis to assays measuring proteins targeted by both platforms, the results were also similar for both platforms.

Most of these numbers are now present in Table 1. In response to the reviewer comment, we have added a sentence on the fraction of sentinel *cis* and *trans* pQTL variant with or PAV or *cis* eQTL in high LD for nearby genes for pQTLs on Olink and SomaScan (See Supplementary Tables S10-S12). In addition, for *cis* pQTLs we also report if the PAV or *cis* eQTL is for the gene encoding the targeted protein (Supplementary Tables S10-S12).

Q1.9: (6.) The authors make some comparisons in pQTL discovery across genetic ancestry, noting examples of PCSK9 and HBBB pQTLs common among African-ancestry participants. These specific

variants have been well recognized – were there novel ancestry-specific pQTLs that have not been previously reported?

As suggested by the reviewer, we have now placed more emphasis on the analysis of the different ancestry groups. The chapter *pQTL analysis in different ancestry groups* has been restructured and expanded to include the following points.

The updated manuscript includes results on the fraction and number of ancestry-specific pQTLs. We report that 1/3 of the pQTLs detected in the African ancestry group involve variants that are very rare or absent in the European ancestry group. A much smaller fraction of the pQTLs in Europeans and South Asians correspond to ancestry-specific variants. There are currently much fewer disease associations reported in GWAS among individuals of African ancestry. Here we present ancestry-specific pQTLs which will be useful once more disease associations are detected in those groups.

In addition, LD classes of pQTLs are substantially smaller in the African ancestry group compared to the European group, in many cases allowing a refinement of the signal detected in Europeans. We have catalogued all these refinements (Supplementary Table S17). We provide two notable examples of *cis* pQTL refinement at *SERPINI2* and *CD58*. At the *CD58* locus we speculate that the pQTL refinement from European to African allows similar refinement of the corresponding disease association to multiple sclerosis.

Q1.10: (7.) The authors note that replication of immunoassay pQTLs identified in UKB-BI among Icelanders was lower than expected. Do the authors think this is due to differences in population structure? Is this a phenomenon unique to pQTLs or does it represent a larger difference between UKB-BI and Icelandic populations?

The observed replication rate for *cis* pQTLs in Iceland using Olink (n=1,514) of the ones detected in UKB-BI using the same platform is lower than the expected value (74% vs 85%, ratio of 0.87). We note that the expected value represents the best-case scenario and is based on the assumption that the only thing different is the sample size. The ratio of observed vs expected *trans* pQTL replication is smaller (19% vs 35%, ratio of 0.54). We have added a sentence underscoring this fact and speculation about the impact of population differences on this replication (section *Detection of pQTLs*).

We have previously reported that when testing SomaScan-detected pQTLs in the Interval study in a much larger set in Iceland, only 83% of pQTLs replicated (Feringstad et al., PMID: 34857953). It has been speculated that difference in sample processing and preparation might affect the detection between groups using the same platform².

Q1.11: In addition to the UKB-BI/Icelander findings, it would be helpful to know the degree of pairwise cross-ancestry replication within other combinations of UKB and Icelander populations (eg. what proportion of UKB-BI pQTLs are nominally identified in UKB-AFR/SAS and vice-versa; presenting a heatmap/matrix would be intuitive).

In order to address this, we have now added Supplementary Tables S17 and S18 where we show correlation of the top pQTL in European with the top pQTL in African and South Asian ancestry groups (r^2 in BI and r^2 in AF or r^2 in SA).

Overall, 30% of pQTL in African were with variant absent or very rare in British Irish, whereas the contrary was almost never observed. We noted that when the variants were in high LD, the LD class was on average smaller in African than British Irish and that often the pQTLs are quite different between ancestral groups (all r^2 between top pQTLs are reported in Supplementary Table S17)

Q1.12: (8.) Line 263 – “Cis pQTLs on SomaScan and Olink and their colocalization with disease-associated variants”. In this section the authors report “colocalization” of cis pQTLs with disease-associated variants. As presented, it appears that the authors consider “colocalization” to represent the presence of a disease-associated variant that is in high linkage disequilibrium ($r^2 > 0.80$) with a sentinel pQTL. However, the “colocalization” terminology more conventionally refers to the formal statistical test of the same name, used to evaluate the presence of a shared causal variant influencing protein levels and disease risk (see PMID: 24830394, PMID: 34587156, PMID: 33536417). This “colocalization” terminology is subsequently repeated in the following section. The authors should formally test for colocalization (eg. using coloc, HyPrColoc, or similar tools) between pQTLs and disease-associated variants.

Our aim was to assess the relationship between pQTLs and as many reported variants for association to disease and traits as possible. The GWAS catalogue has through years become the most exhaustive library of such association. However, in most instances only the variant most significantly associated with disease or trait is reported. This would not suffice to use the colocalization approaches mentioned by the reviewer.

In addition to the approach that we had taken based on r^2 between top pQTL and disease- or trait-associated variants, which can be applied to any reported association in the GWAS catalogue (Sun et al., PMID: 29875488), we have now also adopted a second approach to address the reviewer’s concerns. This is based on determining the credible set for the pQTLs and checking if the reported disease variant is part of this credible set (Maller et al, PMID: 23104008).

For the examples that we quote in the text and where we have access to full data, we have also performed statistical colocalization using the COLOC package implemented in R. (Giambartolomei et al, PMID: 24830394)

Q1.13: Additionally, the association of the identified proteins with disease would be further strengthened by utilizing Mendelian Randomization based approaches.

The scope of the manuscript is already very broad and we consider such analysis as Mendelian Randomization to be outside its scope. We emphasize that summary statistics will be made available through deCODE Genetics and individual genotypes, protein levels, and phenotypes will be accessible through application to the UK Biobank allowing further analysis.

Q1.14: (9.) Section “Proteins targeted by both platforms” – this section appears to focus on cis pQTLs uniquely identified by the immunoassay. Are there cis pQTLs identified uniquely by the aptamer-based method with meaningful disease associations?

We have now added examples that illustrate *cis* pQTL uniquely identified by each platform. We have added an example of a *cis* pQTL identified only on SomaScan, where an IL2RB *cis* pQTL unique to SomaScan associates with asthma risk and eosinophil levels (page 16).

Q1.15: (10.) More clarification about what is new, biological importance

We have completely rewritten the discussion, which should address the reviewer’s comments.

Concerning the novelty of our analysis, we emphasize the following points:

- We compared a much larger number of proteins targeted by both platforms (1,848 proteins in 1,514 individuals) than reported by Pietzner (871 proteins in 485 individuals), Katz (591 proteins in 568 individuals and 1,137 proteins in 219 individuals) and Raffield (427 proteins in 48 samples from 10 individuals).

- Measurements in plasma from 50K individuals from the UK Biobank enable the detection of *cis* pQTLs for 2,093 proteins using the Olink platform, or more than twice the number of proteins reported in previous studies using Olink in smaller sample sizes. This allowed the comparison of pQTLs detected on Olink and SomaScan (as we previously reported) for a much larger number of proteins than previously possible.
- We demonstrate through examples the value of cross-ancestry analysis based on Olink in UKB.

While using one platform does shed light on specific biology, we emphasize that by performing a comparison of these two platforms we can reveal some characteristics of the platforms with implication for the study of disease.

We have discussed the fact that despite being targeted by either platform a number of proteins may have too low abundance in plasma to be detected and measured with confidence by these platforms.

We also emphasize that a given gene may give rise to multiple proteoforms and these may be differentially targeted between platforms even if the protein is annotated with the same gene name on both platforms. This is indicated by the low correlation of measurements between platforms despite both claiming to measure the same protein and both having a *cis* pQTL.

We have also added further examples of disease associations to illustrating how the choice of platform can influence the results and thus the biological interpretation.

We have structured our discussion to more clearly show the novelty and biological importance as discussed in our reply to Q1.3.

Q1.16: (11.) Expand on implications for study of proteomics/genomics in general

As a part of our comprehensive revision of the Discussion section, we have added general implications of our findings for the field.

One of our results is that there is currently evidence for about the same number of proteins being measured and having evidence of *cis* pQTL on the SomaScan v4 and Olink Explore 3072 platforms. However, when studying the proteins targeted by both platforms (n=1,848) there is evidence of *cis* pQTL for more Olink assays than SomaScan ones (80% vs 60%). We emphasize in the discussion that the current versions of the platforms target a number of proteins that appear to be difficult to detect and measure in plasma based on the analysed data.

We point out that proteins with evidence of *cis* pQTLs in plasma on one platform can guide the selection of proteins for the other towards ones that have the highest chance of successful measurement in that medium.

Q1.17: (12.) The manuscript ends in a rather abrupt fashion. Limitations and Conclusions are needed.

We have now completely rewritten the discussion as mentioned in our response to Q1.3. We have made sure to include limitations and conclusions in the revised discussion.

Minor:

Q1.18: (13.) It is probably more accurate to say “for casual inference” than “to determine causality” on line 46.

This has been changed to “causal inference”.

Q1.19: (14.) Lines 52 to 55 are better suited for methods.

As suggested by the reviewer, these sentences have been moved to the methods section, while comparison of the normalization methods has been moved to Supplementary Information.

Q1.20: (15.) In the introduction the authors qualitatively describe differences in the proportion of intracellular/secreted proteins captured by each assay in comparison to the overall proteome reported in the Human Protein Atlas. Are the authors able to statistically compare these differences (Eg. by chi-square)?

As suggested by the reviewer, we have added p values to this statement.

Q1.21: (16.) The detail in lines 61-68 is probably better for the discussion while a short summary that previous cross replication attempts have had moderate success but been largely limited by sample size and the targeted nature of the Olink platform.

These points are now covered in the Discussion section. The details of prior replication efforts have been moved to Supplementary Information S5.

Q1.22: (17.) The results presented on page 5 are interesting, but a bit less detail is needed in the main text. Also the paragraphs would flow better in the following order “Comparison of precision..”, “Effect of dilution on precision..” and then “Between-platform correlation...”.

As the discussion of dilution groups affects both precision and correlation, we believe it fits better after each of them has been discussed on its own.

In order to address the results of the recent publication by Katz et al., this section has been comprehensively restructured, which we believe improves the flow of the text. Further simplifying the section, discussion of the SMP normalization of the SomaScan data has been largely moved to Supplementary Note S2.

Q1.23: (18.) The authors use correlation and association interchangeable in many instances throughout the paper where better precision in language could be employed. For example on line 109-112 they use the term correlation to refer to the association between protein levels and both age and sex. While one could do a formal correlation with both, it seems more likely that linear and logistic regression were used rather than correlation methods. They should be more precise in the results and more detailed in the text.

As suggested by the reviewer, the word ‘correlation’ has been changed to ‘association’ throughout, except where it refers to correlation tests.

Q1.24: (19.) A more important example of the above issues is on line 130-142 (paragraph on “correlations between levels of proteins measured using the two platforms...”) – the authors again refer to testing correlations between protein levels and incident heart failure and Alzheimer’s disease; however, as the authors are implying a causal association (eg. protein levels impact disease incidence), the variables are not interchangeable and should be reported as “associations” rather than “correlations.”

The word ‘correlation’ has also been changed to ‘association’ in this context.

Q1.25: (20.) In the same analysis, it is best to avoid the use of prediction unless the authors have conducted AUC analysis. Rather they appear to simply be referring to magnitude of the effect estimate and should just say so.

As suggested by the reviewer, references to 'prediction' have been changed to 'effect'.

Q1.26: (21.) In Figure 1, consider adding the number of sequence variants tested in the Iceland 1K population (presumably 33M?).

As suggested by the reviewer, this number has been added to the figure.

Q1.27: (22.) Given the differences in the analysis methods and thresholds used between the current analysis and the UKB-PPP analysis, which the authors point out, this comparison is informative but not really that interesting. It should be moved to the supplement.

This discussion has been greatly reduced and moved to Supplementary Note S5.

Q1.28: (23.) One line 174-175 it is probably more accurate to say that the lead variant at some of the pQTLs had a higher MAF rather than just that the pQTL was more common.

This paragraph has been rephrased to refer to the MAF of the variant instead of the pQTL as suggested.

Q1.29: (24.) In that the authors did not verify false discovery, they likely should refer to the section in Lines 190 as "theoretical" or "estimated" false discovery rates. This could also be moved to the supplement.

We have changed the text to refer to "estimated false discovery rates".

Q1.30: (25.) Line 246 – "Thus, even when cis pQTLs for a protein are detected by both platforms, they are not necessarily the same." The authors note that only 45% of cis pQTL are in high LD across platforms – can the authors speculate why this may be the case? Is this primarily driven by differences in population structure between UKB-BI and Icelanders, or due to differences in the assays?

We hypothesize that differences and low linkage disequilibrium of *cis* pQTLs is caused by differences in assays between platforms. We have now emphasized the following points and they are included in the discussion chapter:

Even in cases where *cis* pQTLs provide evidence that both platforms are measuring the targeted protein, different variants often drive the associations. This is likely explained in part by the two platforms targeting different proteoforms.

Also, some *cis* pQTLs correspond to binding artefact, specifically when the *cis* pQTL corresponds to a coding variant and not in LD with a *cis* eQTL, as we assessed on both platforms.

Q1.31: (26.) Line 248 - Several pQTLs were highly pleiotropic. At these pleiotropic loci the authors may consider performing multi-trait colocalization using HyPrColoc (PMID: 33536417) to better estimate the number of proteins which truly share each pQTL.

For a large number of pQTL we observe multiple signals at a locus (top and secondary). In order to assess the colocalization of variants in a region, associating with levels of different proteins, we determined if these variants are in the same LD class allowing us to describe pleiotropic trans pQTLs.

Our method and the HyPrColoc are very related and when it comes to pleiotropic variants our main aim is to compare them between platforms. We observed that highly pleiotropic pQTLs are in part study specific. As the difference in number of proteins associated with a pleiotropic pQTL is often extremely large between platforms, difference in methods when computing the number of proteins affected by the pleiotropic pQTLs would not change this conclusion. We have updated the chapter on pleiotropic pQTLs.

We provide the summary statistics and data will be made available enabling downstream analysis.

Q1.32: (27.) Consider describing the proteomic platforms more generically as immunoassay- and aptamer-based methods (rather than by brand name) throughout.

As there is no way of separating differences in the performance of the underlying technology (immunoassays vs. aptamers) from the implementation of the technology on these platforms, we feel it is more honest to describe the current comparison as a comparison of platforms rather than a comparison of the underlying technologies used by each platform. Another hypothetical method using either immunoassays or aptamers would have to be assessed independently. Some of the differences may be rooted in the different protocols used by the two platforms, which can be independent of the main technological differences, i.e., immunoassays vs aptamers.

However, we have tried throughout the manuscript to emphasize the underlying technology of each of these platforms, i.e., immunoassays vs. aptamers.

Referee #2 (Remarks to the Author):

In “Comparison of large-scale immunoassay- and aptamer-based proteomics through genetics and disease” Eldjarn et al. describe the comparisons of Olink and SomaLogic data sets originating from the currently largest population studies. They anchor their investigations on the genetic associations of both platforms and cohorts and dissect their comparison of cis- and trans-pQTLs, ethnicities, phenotype associations, and features of the target proteins and assay characteristics. The work uses own data and those presented by the UKB Pharma project in a pre-print from Sun et al. (DOI: 10.1101/2022.06.17.496443). The manuscript is well written and sufficiently powered. It addresses an important aspect of connecting and comparing proteomics data obtained by different methods and independent cohorts based on genetic and phenotypic information.

Q2.1: Previous studies have compared the leading affinity proteomics methods at a more focused and single cohort scale. This includes the cited work by Pietzner et al. (Ref 7), and recent work by Katz et al. (not cited; DOI: 10.1126/sciadv.abm5164), both highlighting similar differences between the technologies. In the latter work, a focus was on accuracy and precision, leading to an assessment that “protein target specificity and phenotypic associations were better for the Olink platform, but measurement precision and proteome coverage were better for SomaLogic”. Given the scale of the presented study, a bigger focus should therefore be set on aspects that cannot be accurately investigated in underpowered studies, such as the trans-pQTLs.

We thank the reviewer for their comment and summarize the main advances of the current manuscript:

a) Directly comparing levels measured by both SomaScan v4 and Olink Explore 3072 from 1.5k Icelanders. The number of proteins targeted by both platforms is over 1,800. Thus, we compared

many more targeted proteins (1,848 proteins in 1,514 individuals) than either Pietzner (871 proteins and 485 individuals) or Katz (591 proteins on 568 individuals and 1,137 proteins on 219 individuals). Specifically, such comparison results based on the Olink Expansion set of assays had not been reported before.

b) Comparing association of genotype and disease to protein levels in Iceland using Somascan (36K) and UK using Olink (46K). Such large sample sizes allow the detection of a large number of pQTL (*cis* and *trans*) and protein-phenotype associations that were not detected by Pietzner and Katz using much smaller sample sizes. Our main comparison is performed using *cis* pQTL. For *cis* pQTLs a small fraction of the associations has significance close to the threshold. Thus, with the current sample sizes we are probably approaching a plateau of assays with *cis* pQTL in contrast with previous studies. For *trans* pQTL, there is a high fraction of associations with significance close to the threshold, and we expect that larger sample size will bring more association. Of note we identify a large fraction of pleiotropic *trans* pQTL (association with large number of protein) that are specific to one of the studies, and might reflect difference in the platforms, sample handling and processing.

We are now quoting the relevant Katz reference that was not published at the time of submission.

Concerns to be addressed: While SomaLogic's approach uses a slow-off rate and a single binder selected due to its capabilities of holding on to an intended target, the Olink approach uses two complementary antibodies that reveal a signal when the antibodies bind a common target of interest (in proximity). Besides high-performance reagents, the stringency of an assay's specificity depends on two factors that govern the accessibility to the target of interest:

- 1) Aspects relating to the target proteins themselves - such as abundance, structure, interaction, or modifications.**
- 2) Aspect relating to external factors (sample matrix), such as abundance and frequency of other proteins with similar epitopes.**

Since neither Olink nor SomaLogic provides data on the actual epitopes and interfering factors, other means are needed to infer sources of off-target binding.

Q2.2: It would be highly interesting to investigate if the available genetic data be used to gain more insights into off-target interference, e.g., by studying the pleiotropy or *trans*-pQTLs that won't match expected pathways or processes.

When focusing on pleiotropic *trans* pQTLs we noted that the concordance between the number of associated proteins is modest between platform and have discussed some of the discordant pleiotropic pQTLs and their possible interpretation (see section *Pleiotropic pQTLs*):

"A number of the pQTLs that are pleiotropic on Somascan but not on Olink are close to complement factor genes (e.g., C3, CFD, CFH), this might indicate that the Somascan processing or measurement might interplay with complement factors as previously speculated for CFH. Other pQTLs are pleiotropic on Olink but not on Somascan such as variants in PNPLA3 and FADS1, which are genes involved in regulation of fat in liver."

In addition, we understand that the richness of these two datasets will enable further downstream analyses. The availability of summary statistics will give the opportunity to everyone to investigate the data.

Q2.3: As observed by the authors, the two platforms concur better for higher abundant proteins measured in diluted samples. This can be explained by a lower complexity of the sample matrix

that leads to a decrease in external interference factors while target protein-related factors become dominant. It is known that the more abundant plasma proteins are secreted (DOI: 10.1126/scisignal.aaz0274). Hence there is an intrinsic bias in why secreted proteins appear to perform better. The authors should therefore evaluate their data based on the organ that serves as the main source of the proteins in the blood. This could be very informative when connecting the blood protein data to specific diseases.

We are now acknowledging the possibility mentioned by the reviewer in the chapter *Effect of dilution and protein sub-cellular location on precision and correlation*. We have added the following sentence and reference:

“We observe that the abundance of proteins reflected by the dilution group is associated with the precision of measurements (Extended Data Table 7, Extended Data Figure 3, Extended Data Table 8). Previous reports have described the effect of decreased matrix complexity on measurement, which may along with the abundance affect the precision³.”

Following the reviewer comments, we have systematically annotated the tissue of preferential expression for each protein according to the Human Protein Atlas to the Supplementary table for both platforms. To answer the point of the reviewer we have also calculated the median correlation of levels between platforms when stratifying by tissue based on the proteins with an enriched tissue expression and quoted it in the result section. Only tissues with at least 10 proteins were considered. We have added the following to the results section (*Effect of dilution and protein sub-cellular location on precision and correlation*):

“We noted a wide range of median correlation between levels when stratifying by tissue based on enriched expression (r between 0.05 and 0.64). The tissue with the lowest median correlation was the pituitary gland followed by testis, fallopian tube, retina, skin and brain (all with $r \leq 0.15$). The tissue with the highest median correlation of levels was gallbladder, followed by smooth muscle, cervix, endometrium, pancreas and salivary gland (all with $r > 0.45$; Supplementary Table S4).”

Q2.4: As another useful outcome, the authors should list and rank all proteins they deem comparable on both platforms. Hence they could be annotated as platform-independent protein assays. These targets should become known to the community so they can be used and prioritized in studies aiming to expand on the current proteomics datasets. The Sun et al. work mentioned expanding on their work by using MS to quantify proteins in an affinity-free manner. Such efforts will likely inform about some of the high confidence protein assays, while lower abundant proteins will less likely be reached from the get-go.

We thank the reviewer for this practical suggestion. We agree that the information derived from this paper will be useful to a large audience.

We have now added a supplementary table with the relevant information (Supplementary Table S20). For each protein studied in plasma in the manuscript (i.e. 5,814 targeted by at least one of the two platforms) we have indicated if it is targeted by Olink Explore 3072 or SomaScan v4 and at what dilution it is tested on each platform. We annotate the cellular location of the protein as well as the tissue specificity and the enriched tissue of expression. Additionally, we report on the presence of cis pQTLs and if the cis pQTLs are correlated with PAV or cis eQTL. For the 1,848 proteins targeted by both platforms, we report the correlation of levels between platforms and its significance that we have assessed based on 1,514 Icelanders with measurements on the same sample. The table also contains the number of platforms targeting the protein, the number of platforms with cis pQTLs for the protein, and an indicator for platform consistency if the protein is targeted by both, with cis

pQTLs on both and over 0.5 correlation (577 pairs of assays). We also report on r^2 between the top *cis* pQTLs on the two platforms.

We note that while the exact definition of “platform independence” is open to interpretation, the minority of proteins targeted by both platforms show consistency across all dimensions listed above in plasma.

The results we observed and the conclusions we derived might be limited to plasma and not extend to other matrices or sample types that would have to be assessed separately.

Q2.5: In the section following line 143, the authors discuss NFL as a case with inverse directions of the effect. Did these two assays overlap in their pQTLs or differ in assay performance? In a previous study, Hong et al. presented a similar case where two antibodies were used in the same multiplexed assay against the abundant protein HRG. They found effects of PAVs, inverse trends of genetic associations, and consequently, low correlation of the data with differences in phenotypic association (DOI: 10.26508/lsa.202000817). Hence, the NFL case would benefit from more details to understand why differences have been observed.

As mentioned, the NFL levels were not correlated between the two platforms. We have now added to the text that we did not observe a *cis* pQTL for NFL on either platform.

While in the case referred by the reviewer, Hong et al. describe two assays demonstrating inverse trends of genetic association for HRG, they trace their results to a PAV correlated with the *cis* pQTL. However, no *cis* pQTLs were detected for NFL, rendering such explanation implausible. The manuscript has been updated to reflect this absence of *cis* pQTL.

NFL Levels measured by both platforms associated strongly with AD with good ranks but in opposite direction. It has been reported in multiple studies that plasma levels of NFL measured by Olink and SIMOA assay correlate very well (over 0.90 for two studies) (PMID: 34032364, 33087494). In contrast, levels of NFL that we measured using the SomaScan platform did not correlate with SIMOA measurements in a set of 231 Icelanders (Spearman $r=0.00$, $p=0.95$). Whereas SIMOA and Olink appear to measure the same proteoform of NFL, it remains to be understood what proteoform Somascan is measuring.

These points have been added to the chapter where NFL is discussed (*Protein-phenotype associations*, page 8).

Q2.6: Beyond looking at proteins 1-by-1, other means to compare data are by investigating the inter-protein correlation of all proteins in each platform. Did the authors observe any shift from zero, and if particular proteins or dilution sets present differently? Considering that proteins with common pQTLs do not always correlate well on a protein level, it would be interesting to test why this is so. Would, on the other hand, highly correlating proteins always reveal common qQTLs? Can the authors simulate a protein correlation threshold at which the genetic association becomes distinct from another or common for both?

We had already reported on the median inter-protein correlation of all proteins in each platform and now have attempted to make it even clearer. Page 6: “[...] the median pairwise correlation between assays from the two platforms for all possible assay pairs was -0.01, and median within-platform correlation between assays was 0.08 for the Olink platform and -0.01 for the SomaScan platform (0.42 if omitting SMP normalization).”

On Olink, we report a modest shift from 0 for correlation between levels of all assays (median=0.08). On Somascan we observe a large shift from 0 (median=0.42) on non-normalized data, but not on SMP normalized data (median =-0.01).

Of the 1,984 pairs of assays targeting the same protein (n=1,848) on Olink and Somascan the average correlation between platforms is 0.32. When *cis* pQTLs are detected on both platform the correlation is substantially higher (n=1,091 pairs and r=0.48) than when detected on neither of them (n=306 pairs and r=0.17).

We have also demonstrated that levels of protein targeted by both platforms have a lower correlation if the *cis* pQTL observed on one of the platforms is in high LD with a PAV, potentially reflecting a binding artefact. We also speculate that difference between platforms in specificity for different proteoforms encoded by the same gene may explain low correlation of levels despite having *cis* pQTLs on both.

We report that of the 1,848 proteins targeted with both SomaScan and Olink, 1,030 (56%) had a *cis* pQTL detected using both platforms. However, we underline that only for 434 of these 1,030 proteins (42%), the sentinel *cis* pQTLs detected using SomaScan and Olink were in high LD ($r^2 > 0.80$). Thus, even when *cis* pQTLs for a protein are detected with both platforms, they are not necessarily the same. When the *cis* pQTLs are in high LD the correlation is higher than when the *cis* pQTL are not in high LD (median correlation of 0.55 vs 0.49, Mann-Whitney $p = 6.9 \times 10^{-6}$). (Supplementary Table S20).

We note that out of 307 pairs of assays targeting the same protein and with pQTL on neither platform, there is still 50 targeted proteins with levels that are well correlating ($r > 0.50$).

We have added these points to the manuscript in the chapter *Inter-platform correlation of protein levels for shared targets*.

Q2.7: a) One concern is using multiple N=2 comparisons to determine CV values for each platform. Even though a large number of comparisons (N> 400) gives some confidence, it misses judging variability in repeated measurement of single samples with N > 3 and preferably across multiple batches of the data. Batch size and numbers have likely influenced the outcome of the analysis that - partly - contradicts those made by Katz et al.

We note that while the article by Katz et al. was not published at the time of submission of the initial manuscript, we have updated the current manuscript to address the contradictory results. In short, the coefficient of variation (CV) is not generally a measure of precision, as depending on the scale of possible measurements, if a platform were to produce random values in a tight range not including 0, it could have low CV but also no precision.

While we replicate the results by Katz et al. in our data, we note that by factoring in the range of potential measurements from the assay, the Olink assays are on average more precise than the SomaScan assays.

For evaluation of the relative strength of batch effects on each platform, the following has been added as Supplementary Note S1:

“Both Olink and SomaScan use repeated measurements of control samples, specific to the platform, for quality control. Computing the variation in the measurements of the control samples using either pairs of measurements from the same plate or pairs from different plates allows us to evaluate to

what extent the variation is inherent in the assay itself and to what extent due to variability between plates, and how those variations relate to the total variation of the assay.

We evaluated the variation in measurement of control samples in the Icelandic data set (measured by SomaLogic) and the UKB data set (measured by Olink) using the MAD to estimate the standard deviation, calculating the variance as $\sigma^2 = (1.4825 * MAD)^2$. Assuming that the variance on the control samples inherent in the assay σ_a^2 and the variance due to batch effect σ_p^2 are not correlated, we can decompose the total variance of the control sample measurements as $\sigma_M^2 = \sigma_p^2 + \sigma_a^2$.

In order to determine the impact of the plate effect on measurement results, the plate effect must be considered in relation to the total variance of the assay, σ^2 . We can then evaluate how much of the total variance is due to the variability inherent in the assay (σ_a^2/σ^2) and how much due to batch effects ($\frac{\sigma_M^2 - \sigma_a^2}{\sigma^2}$).

We can estimate σ_a^2 by using the difference between the two control samples measured on each plate, while we can estimate σ_M^2 by considering the control samples without regards to plate, and σ^2 from all values for the assay in question.

The fraction of assay variance explained by the variability inherent in the assay was 0.23 on the Olink platform and 0.10 on the SomaScan platform, while the fraction of variance explained by batch effects was 0.06 on Olink and 0.02 on SomaScan.

The discrepancy between the assay precision evaluated using the CV ratio of duplicate measurements of samples and the variation of measurements of control samples may be in part due to the use of different control samples between the platforms.”

Furthermore, the following references in the main text (section *Comparison of the precision of the two platforms*, page 5) stating the differences in batch effect between the two platforms:

“To evaluate to what extent the variability of an assay is inherent in the assay itself and to what extent due to variability between plates, and how those variations relate to the total variation of the assay, we can use pairs of measurements of control samples. The fraction of assay variance explained by the variability inherent in the assay was 0.23 on the Olink platform and 0.06 on the SomaScan platform, while the fraction of variance explained by batch effects was 0.10 on Olink and 0.02 on SomaScan (Supplementary Note S1, Extended Data Figure 2).

The discrepancy between the assay precision evaluated using the CV ratio of duplicate measurements of samples and the variation of measurements of control samples may be in part due to the use of different control samples between the platforms.”

Q2.8: B) No longitudinal samples have been studied to determine the variability of resampling, which might be an even more important aspect when considering where the value of proteomics may lie over genomics.

While we agree with the reviewer that the stability of plasma protein measurements with regards to time is certainly an important subject, the data sets analysed in this manuscript do not offer the opportunity to compare the platforms with regard to this aspect. However, this has been to some extent studied individually for each platform (PMID 35734478, 29079756). Furthermore, in the context of diseases, associations between proteins and diseases may still be detectable in studies on large populations even if the variation within subject over time exceeds the variation in levels associated with the disease.

Q2.9: C) The authors should also perform inter-class correlations between the duplicated samples to evaluate the shift from the line of identity and discuss how the scales of the two methods affect their assessments.

As suggested by the reviewer, to evaluate the inter-class correlation between the duplicated samples, we computed the correlation coefficient between all measurements for the duplicated samples on each platform.

The following has been added to the section *Comparison of precision of the two platforms* and Supplementary Note S2:

“The median correlation of the different assays for the same PN was lower for the Olink platform than the SomaScan platform (Spearman $r=0.78$ vs 0.99 , respectively, Mann-Whitney $p=8.4 \times 10^{-210}$). However, when adjusting for differences in data processing by shifting the median for each assay on each platform to 0, the median correlation was higher for Olink than for SomaScan (Spearman $r=0.78$ vs 0.64 , respectively, Mann-Whitney $p=1.4 \times 10^{-23}$, Extended Data Figure 3, Supplementary Note S2).”

“S2: Both platforms emphasize the relative quantification within each assay, and neither of the platforms makes a claim towards the comparability of inter-class measurements. In fact, the normalization process for the Olink data shifts the median measurement for each assay to 0, and the difference between the levels of any two proteins on the Olink assay are therefore purely due to the position of the individual sample within the distribution of the assay. Conversely, for the SomaScan platform, the levels measured by two different assays can have vastly different medians, to the extent that the variation within each assay is dominated by the difference in medians. This ensures that for such SomaScan assays, any two measurements done with the assay with the higher median will always be higher than any two measurements done with the other, leading to stronger inter-class correlation between them. Shifting the measurements for all SomaScan assays to have mean 0, similar to the Olink assays, reduces the median inter-class correlation accordingly to 0.64 (Extended Data Figure 3).”

Q2.10: Another informative measure to judge protein-level data is to investigate the heterogeneity of variance. The authors should present the variance from Olink and SomaLogic data for a few representative cases with concordant data and pQTL mapping. The authors should also discuss how differences in scales affect these measures.

Following the suggestion of the reviewer, for the matching assays, we have plotted the variance of the assay on each platform stratified by presence of cis pQTL on either platform, and color-coded by the strength of the correlation between them. The following has been added to the manuscript (section *Inter-platform correlation of levels for shared targets*, page 5):

“The overall variance of the levels for an assay was more concordant between matching assays on the two platforms where pQTLs were detected on both platforms (Spearman correlation 0.57 for pQTLs on both platforms, otherwise <0.28) but did not show a clear trend for individual platforms depending on the presence or absence of pQTLs on that platform (Extended Data Figure 3).”

As the two platforms use different scales, both of which are arbitrary and only meant for relative quantification of protein levels, comparing the scales of the two does not give meaningful results.

Q2.11: The authors present and discuss the SomaLogic as normalized and unprocessed. Even though this is a relevant topic, it distracts from the core comparison and should therefore be

relocated to the supplementary. Olink data can also be processed and used as inverse transformed data, including or excluding data below LOD. This possibility has though not been covered.

As suggested by the reviewer we have now moved most of the discussion about SMP normalization to Supplementary material. Discussion of LOD of the Olink assays and its relation to other metrics has been added to the results section (sections *Effect of dilution and protein sub-location on precision and correlation*, *Detection of pQTLs*) and discussion. As the correlation presented in the manuscript between the different assays is rank-based, the inverse normal transform does not affect the correlation of individual assays. Association studies to phenotype and genotype data is done on inverse normal transformed data.

Q2.12: Instead, the authors should present how pre-analytical factors, such as age-of-sample, influence the two assays. More interestingly, can the authors investigate how the data comparisons would appear if both data sets were adjusted by age, sex, and BMI?

The correlation of the assays with age and sex is discussed in section *Protein-phenotype associations* (page 7), to which correlations with BMI and age of sample have been added. We have compared these correlations between platforms. Furthermore, we have added Extended Data Table 9, describing differences in population and sample handling between the data sets.

Both pQTL analysis and protein phenotype association are done on data which is age-adjusted and rank-transformed stratified by sex, which then accounts for sex as well.

The relevant text in section *Protein-phenotype associations* now reads:

“The levels of 86% of SomaScan assays and 47% of Olink assays associated with sample age, however the effects were generally small for both assays (Supplementary Tables S5 and S6). The Spearman correlation of sample age effects between pairs of matching assays was 0.08. The levels of 77% of SomaScan assays associated with age, 64% with sex, and 69% with BMI (Supplementary Tables S5-S8). In UKB-BI, the levels of 60%, 68%, and 78% of Olink assays associated with age, sex, and BMI respectively, and the effects correlated well between ancestries in UKB (pairwise Spearman correlation greater than 0.70, 0.71, and 0.87 for age, sex, and BMI respectively; Extended Data Figure 5; Supplementary Tables S7 and S8). The Spearman correlation of age and sex effects between the 2,021 pairs of matching assays on Olink and SomaScan was 0.52 for age, 0.56 for sex, and 0.43 for BMI.”

Q2.13: A core of the comparison is understanding why differences and commonalities were observed. The presented study does not provide further insights or molecular evidence into such aspects. Using an antibody-based dual binder ELISA to validate the exemplified target is a good effort. Still, it favors one platform over the other due to the nature of the assay and affinity reagents used. Enriching proteins from plasma via a pulldown is an alternative strategy to corroborate affinity binding, but caution has to be taken considering common contaminants that will co-precipitate. It is worth discussing that the rebinding effects occur when surface-bound affinity enrichment, such as for SomaLogic’s assays, is being used and that this differs from in-solution binding, such as those used in Olink’s proximity assays.

Following the reviewer’s suggestion we have added the following section to the Discussion chapter:

“The Olink proximity extension assay (PEA) platform is based upon an in-solution binding of two polyclonal antibody pools to a target protein and subsequent hybridization and enrichment of two unique single stranded DNA probes to create a double stranded barcode unique for the antigen. The SomaScan platform utilizes a surface bound enrichment of proteins alongside a universal polyanionic

competitor to prevent transient non-specific interactions. Thus, these two platforms differ in nature and this may affect how proteins are quantified in complex tissue such as plasma. The biochemical properties of orthogonal assays used for validation may need to be considered in the context of the properties of the two platforms.”

Other points:

Q2.14: Many figures were pasted into the documents at a highly inconvenient resolution. It was difficult to judge these on correctness and detail. Instead of Excel, all figures should be reproduced in the more appropriate software, like R.

We have made every effort to ensure that the figures are now of sufficient quality.

Q2.15: Line 26: revise to “1,075 samples were also...”

We have revised that sentence accordingly.

Referee #3 (Remarks to the Author):

Summary of the key results

The authors present a technical tour de force comparing the currently most comprehensive assays to measure plasma proteins, namely the SomaScan v4 platform and the Olink Explore 1536 platform. They present consistency and divergence of both technologies based on a large set of phenotypic and genetic associations in more than 30k participants, representing the largest study currently performed. They highlight technical factors to account for missing replication of phenotypic and genetic associations, and further, that while the Olink assays seems to provide better performance for targeted proteins, the SomaScan platform distinguishes by the larger protein coverage.

Q3.1: While these results are interesting, they do not really represent an advance in our understanding of how those protein technologies can inform biology or clinical translation of genetic findings.

We have updated the manuscript to encompass analysis of the full Olink Explore 3072 (1536 + Expansion) as well as added comments in the discussion chapter describing how those protein technologies can inform biology or clinical translation of genetic findings.

Levels of plasma proteins as measured by these platforms and their derived pQTLs can shed light on biological mechanism mediating disease associated risk through their colocalization, and we have added some more examples of this in the manuscript (section *Cis pQTLs on Olink and SomaScan and their colocalization with disease-associated variants*, page 14). We have now made our best efforts to clarify that when attempting clinical translation of genetic findings, the difference in results based on the two platforms may drastically change the conclusions drawn depending on which platform is used for the analysis. We now include examples for both Olink and Somascan where *cis* pQTLs for a protein in plasma colocalize with a disease-associated variant on one platform but not the other despite the protein being targeted by both.

Association of protein level with disease and other traits points to possible utility and clinical translation as diagnostic tools and biomarkers in the clinical trial context. We have added further examples illustrating how association between protein levels in plasma and phenotypes are either concordant or discordant between platform when targeted by both and emphasized this in the rewritten discussion chapter.

Validity

The conclusions drawn from the study are mostly well justified by the data presented. My greatest concerns are:

Q3.2: (1)) The presentation of cis-pQTLs as orthogonal validation for target specificity sounds obvious, cis-pQTLs may equally likely strongly affect assay performance and it is currently unclear to which extend this explains the reproducibility of phenotypic associations.

We agree with the reviewer that cis pQTLs can serve as validation of “target specificity”, but while this may appear obvious, we still want to emphasize this for the large audience of readers.

We also agree that some cis pQTLs may affect “assay performance” and add that in these cases the pQTL variant would likely be a protein altering variant causing a binding artefact.

We report that of the 1,848 proteins targeted with both SomaScan and Olink, 1,030 (56%) had a cis pQTL detected using both platforms. However, we underline that only for 434 of these 1,030 proteins (42%), the sentinel cis pQTLs detected using SomaScan and Olink were in high LD ($r^2 > 0.80$). Thus, even when cis pQTLs for a protein are detected with both platforms, they are not necessarily the same. When the cis pQTLs on the two platforms are in high LD, the correlation between levels is higher than when the cis pQTL are not in high LD (median correlation 0.55 vs 0.49, Mann-Whitney $p = 6.9 \times 10^{-6}$) (Supplementary Table S20).

In addition to the possibility of binding artefact, differences in cis pQTLs may indicate that the two platforms have affinity for different proteoforms. The proteoforms can encompass different cleaved or bound forms, post translational modifications and different isoforms resulting from differential transcript splicing. Proteoforms encoded by the same gene may inherently reflect different biological processes and therefore have different association with diseases and traits.

Taken together, this shows complementarity of conclusions based on the two platforms and suggests that the future expansion of these and other proteomics platforms can to an extent be guided in the selection of targeted proteins and proteoforms by validated assays through cis pQTL on the other platforms.

To address the reviewer’s comments, we have emphasized and reworded the sections about PAVs in high LD with cis pQTLs both in the results and discussion chapters. About one third of cis pQTLs on both platforms correspond to PAV, and we have previously shown that those cis pQTLs that are not in high LD with cis eQTLs are more likely to be artefactual especially when the PAV is a missense variant (Ferkingstad et al., PMID: 34857953). The remaining two third of cis pQTLs, with no PAV in high LD, are much less likely to be artefactual.

We have also placed emphasis on the fact that cross-platform replication is lower when cis pQTLs are in high LD with a PAV compared to low LD.

Q3.3: (2)) The authors, correctly, claim that cis-pQTLs can help establish causality, but present only a rather simple cross-referencing exercise to variants reported in the GWAS catalog. This is one of essential messages of this paper and greatly determines its value to the scientific community. The authors would need to present more rigorous statistical tests, including statistical colocalization and Mendelian randomization to validate their claims.

We consider the GWAS catalogue to be the most exhaustive library for systematic assessment of the existing association with diseases and traits. However, the GWAS catalogue comes with the limitation that most often only the most significantly associated marker is indexed since only it was presented in the supporting publication. Per phenotype and locus, we restrict ourselves to test the most significant markers out of any publications listed in the GWAS catalogue. The approach we took, which is based on LD between the variants associating most significantly with diseases and protein levels, should be viewed as a screening method. Analysis based on high LD has been shown to be consistent with statistical colocalization. It has been reported that most pQTLs in high LD ($r^2 > 0.8$) to disease associated variants have a colocalization with them. In 2018, Sun concluded from a proteomics scan that colocalization was “highly likely” (PP > 0.8) for 89% of the pQTL-disease association pairs in high LD, and “most likely” explanation (PP > 0.5) for 98% of these pairs. We now have added a reference to this result.

In addition to the above LD-based approach, we have now also included a comparison based on credible sets of pQTL variants. For each pQTL, we created a 95% credible set of variants, using methods described by Mallar *et al.*⁴ and checked if any of the variants included in the credible set was listed in the GWAS catalogue list of top markers. We further restricted to variants in the credible set in high LD ($r^2 > 0.8$) with the top pQTL variants in order to provide a more conservative set.

Finally, for all examples discussed in the manuscript, where the top markers for pQTL and disease are not the same, we have added a statistical colocalization (posterior probability estimated using the R package COLOC) based on genotype-phenotype association results available to us⁵.

Currently a very large fraction of association between sequence variants and diseases or traits have been observed in Europeans only. With larger set covering multiple ancestries being made available, the number of associations in populations other than Europeans will increase (Our Future Health in UK, All of Us in US). The UKB includes subpopulations of different ethnicities and will also allow colocalization to signals discovered among non-Europeans.

Q3.4: Further, it is unclear how replication of pQTLs across assays has been defined when comparing between DeCODE and UKB.

For replication between platforms, the p-value threshold is 0.05, with the requirement that initial and replication associations are in the same direction. We have added a sentence to this effect to the methods.

Minor:

Q3.5: - The reporting of pQTL fractions assumes that the additional proteins on the SomaScan assay would be similarly beneficially measured using Olink, which is unlikely to be the case, pls report such fractions only for matching pairs of proteins

As suggested by the editor and reviewer, we have since the initial submission added data for Olink Explore Expansion in addition to the Explore 1536, doubling the number of studied proteins targeted

by Olink (2,931 assays). With the addition of the Expansion set of assays, a further 739 proteins previously only targeted by Somascan v4 are now also targeted by Olink.

Considering first the full set of assays tested on each platform, *cis* pQTLs were present for 2,101 out of 2,931 Olink assays and 2,120 out of 4,907 SomaScan assays (Table 1). While a very similar number of proteins had a *cis* pQTLs detected by either platform, fewer proteins are targeted by Olink than SomaScan. Thus, we observe *cis* pQTLs for 71% of proteins using Olink 3072 and 43% using SomaScan v4.

Importantly, when restricting our analysis to the matching assays for 1,848 unique proteins targeted by both platforms, the Olink assays were more likely to have a *cis* pQTLs (80% of 1,832) than the SomaScan assays (58% of 1,954) (chi-squared $p=1.8\times 10^{-42}$, Table 1).

The difference between the platforms in the fraction (and number) of proteins with a *cis* pQTLs we reported in the initial submission can still be observed in the set of assays added since. Considering only assays targeting the 739 proteins currently targeted by both platforms and included in the Expansion set of assays, the number of assays with *cis* pQTLs was 499 on Olink (67% of 745) and 395 on SomaScan (49% of 799) (chi-squared $p=4.3\times 10^{-12}$).

In addition, we observe a lower fraction of proteins with *cis* pQTLs for proteins targeted by the Expansion set of assays than the by the initial 1536 set (56% vs 85%). Similarly, a lower fraction of novel proteins targeted by the SomaScan v4 platform compared to the ones targeted by a previous generation of the platform (SomaScan 1.2K, Supplementary Table S2) have a *cis* pQTL in our data (40% vs 56%). We conclude that on both platforms, the fraction of proteins with *cis* pQTL trends down with more recent versions.

We have now added numbers corresponding to these results throughout the manuscript, both in Results and Discussion sections.

Originality and significance

Q3.6: While this work is certainly the largest and most comprehensive of its kind, the conclusions drawn from the data largely align with what has been reported previously (e.g., Raffield et al. *Proteomics* 2020; Pietzner et al. *Nature Communications* 2021; Katz et al. *Science Advance* 2022), but the authors sometimes miss to acknowledge those findings.

While we do reference Pietzner et al., the article by Katz et al. was not yet published when the initial manuscript was submitted. The omission of Raffield et al.'s results was an oversight which has been corrected in the resubmitted manuscript. We thank the reviewer for pointing this out and are now acknowledging these three studies and comparing them to ours.

The following has now been added to the Discussions section:

“We compare a much larger number of proteins targeted by both platform (1,848 proteins in 1,514 individuals) than Pietzner (871 proteins and 485 individuals), Katz (591 proteins on 568 individuals and 1137 proteins on 219 individuals) and Raffield (427 proteins in 48 samples from 10 individuals).”

In the manuscript, we are now directly comparing levels measured both with Somascan v4 and Olink Explore 3072 from 1.5k Icelanders.

Specifically, comparison results based on the overall Olink Explore 3072 had not been reported before since the Olink Expansion set is recently released.

We are comparing association of genotype or disease with the levels of 1,848 proteins targeted by both platforms using Somascan (36K Icelanders) and Olink (44K UKB participants). In both these groups the sample size is large and allows the detection of a large number of pQTLs (*cis* and *trans*) and protein-phenotype associations that could not be detected and compared when using much smaller sample sizes as reported by Pietzner and Katz.

Q3.7: It is further unclear to me, why this work has been submitted separately and was not combined with what has already been published by the UKB-PPP. It appears highly redundant to have both papers separately. The method comparison, ancestry-specific analysis and a substantially improved disease follow-up would have addressed the major shortcomings of the work by Sun et al. (Sun et al. BioRxiv 2022) and would in its totally, as a single paper, provide the conceptual advance that both papers are currently lacking.

Whereas we understand that the reviewer found shortcomings in the submitted version of the UKB-PPP that are addressed in our manuscript, we want to note that the authors of the submitted manuscripts are not the same. We are now addressing the points expressed by the reviewer and the editor by listing clearly the distinction between the two manuscripts with regard to data used, proteomics analyses in the context of genotype and phenotype and are emphasizing these points throughout the manuscript introduction, results and discussion. The commonalities between our paper and the one by UKB-PPP is that they in part use the same data.

The following points underline the main attributes of our paper that we are convinced distinguish it from that of the UKB-PPP consortium.

Some of these distinctions originate in the association of protein levels based on Olink and Somascan with genotypes and phenotypes:

- For Olink in UKB, we emphasized the value of both studying multiple ancestries and exhaustively testing the full genome, including the X chromosome and rare variants when searching for pQTLs. We further develop these themes as suggested by the reviewers. This includes the detection of pQTLs involving ancestry-specific variants as well as demonstration of the refinement of signal across ancestries.
- For SomaScan, we present an updated analysis of our previously published analysis of pQTLs among Icelanders.
- We report a large number of associations between protein levels and phenotypes on both platforms.

Even more importantly, other distinctions are due to our effort to perform the largest comparison of the Olink and SomaScan platforms to date. The comparison is not limited to correlation of levels, but also leverages pQTLs and protein-phenotype association.

- Using a set of 1,514 Icelanders with data from both platforms on the same samples, we perform a direct comparison of protein levels between the two platforms for 1,848 proteins targeted by both.
- We compare the number and list of targeted proteins that have pQTLs on the Olink and SomaScan platforms focusing on *cis* pQTLs and the determinants of their presence. In

particular, we perform this comparison for the 1,848 proteins that are targeted by both platforms, using a comparable sample size in the two populations.

- We assess the correspondence of the pQTLs (LD) detected using each platform while targeting the same protein.
- We compare the relationship of protein levels measured with both platforms with age, sex, and a large number of diseases and other traits. We give examples pointing to differences in results using the two platforms when studying diseases.
- We illustrate how different pQTLs, based on the two platforms, can lead to different colocalization with disease-associated variants.
- The fraction of proteins with cis pQTLs is lower for proteins expected to be less abundant in plasma than others and their levels are less correlated between platforms. Consistently we note that less abundant proteins have a higher fraction of measurements below the limit of detection (LOD) on Olink where such information is available. We conclude that some proteins are not abundant enough in plasma to be detectable.

Data & methodology

Q3.8: While the authors have certainly access to the largest data set to compare both platforms, the influence of differences in sample handling and study characteristics between UKB and DeCODE is not sufficiently presented, investigated and discussed, which strongly limits the conclusion drawn from the data.

Regarding differences due to the country (i.e., Iceland versus UK), good concordance within the Somascan platform has previously been established between the Icelandic 36K data set and a separate UK-based population, with 83% of the reported pQTLs in the INTERVAL data set replicated in the Icelandic 36K data set¹. Despite good overall concordance it was noted that some of the highly significant trans pQTL, including pleiotropic variants did not replicate between these two studies while using the same platform.

It was speculated by us and others that some differences in sample handling and processing may explain difference in pQTL². In particular, we have previously reported that when investigating associations that did not replicate from the INTERVAL study (UK), also using SomaScan, the majority of non-replicating trans pQTL associations were with a single variant, rs62143194 on chromosome 19, and these associations also did not replicate in the Fenland study (UK)^{1,2}. The lack of replication might be explained by standardization and/or sample handling².

We have now added a chapter highlighting examples of discordant of pleiotropic trans pQTLs between platforms:

“A number of the pQTLs that are pleiotropic on Somascan but not on Olink are close to complement factor genes (e.g. C3, CFD, CFH), this might indicate that the Somascan processing or measurement might interplay with complement factors as previously speculated for CFH². Other pQTL are pleiotropic on Olink but not on Somascan such as variants in PNPLA3 and FADS1, which are genes involved in regulation of fat in liver.

Other such differences between studies have been noted and attributed to variants associated with platelet counts, and have been suggested to be at least partially because of differences in sample handling and storage (Supplementary Note S8).”

We have now also added a table comparing the demographics for the two groups as well as features of sample handling and storage (Extended Data Table 9). On average, the demographic dimensions are very similar.

The Icelandic 36K and UKB-BI data sets are drawn from two relatively close North-Western European populations. However, the Icelandic and UKB data sets are differently biased. The healthy-donor bias in UKB has been documented (cite Fry 2017) and the UKB sample used here is less than 0.1% of the UK population and is expected to be depleted of diseases (cite Fry 2017). In contrast the Icelandic data set is enriched for certain diseases (e.g. cancers) at the time of sampling and represents over 10% of the Icelandic population.

Concerning the large difference observed on many of the pleiotropic trans pQTL, we suspect that the differences are unlikely because of age, sex, or disease enrichment but most likely rooted in difference in sample handling and processed independent or dependent of the technology used.

We have now made a note of these when discussing pleiotropic pQTL. These differences are not expected to affect *cis* pQTL to the same extent (*Detection of pQTLs*, page 10; *Pleiotropic pQTLs*, page 14, *Discussion*).

Q3.9: Statistical analyses have been overall well defined and executed. However, as already outlined above, linkage between protein and phenotypes using genetic instruments is certainly a subject that needs improvement. For example, the term ‘colocalization’ is used without proper justification of a statistical test.

See detailed answer to Q3.2 of the same reviewer.

Q3.10: The comparison between this study and the previous UKB-PPP preprint is rather confusing, as both analyze the very same protein data and I think it needs more information on why a surprisingly large fraction of pQTLs do not agree (using large genomic, $\pm 2\text{Mb}$, is a poor definition of replication). What role do rare variants play? How do those differences affect phenotypic investigations? What explains the massive difference in trans-pQTLs (16k vs 9k)?

The detailed comparison between our results and those of the UKB-PPP has now been removed from the manuscript.

With regards to the results from the initial submission, there are multiple differences between how we and the UKB-PPP consortium analysed the data that might explain difference in the number of pQTL detected. In the initial submissions the number of proteins with *cis* pQTLs on Olink was very similar in our and UKB-PPP manuscript. The difference in the number of pQTLs was mainly rooted in the trans pQTL (16k vs 9k).

We are not correcting for the number of assays tested but rather decided to display the estimated False Discovery Rate. Whereas we used for discovery all the data in UKB for each of the 3 ancestral groups, the UKB-PPP only used two third of them and solely from European in their initial submission.

We tested and reported variants on the X chromosome as well as very rare variants, whereas the consortium did not.

When it comes to phenotypic impact of these differences it is clear that colocalization with rare pQTLs or ancestry specific pQTLs will be missed if these pQTLs are excluded, as exemplified in the case of ancestry specific pQTLs for PCSK9 and HBB. Additionally, we now show how the ancestry

analysis allows substantial refinement of the pQTL LD class size when moving between ancestral populations.

Q3.11: Please provide coloc pairs of protein and phenotype by unique variant or highly related cluster thereof and further separated by cis and trans to judge the impact of pleiotropic variants. Further, were previous proteomic studies excluded from the GWAS catalog data? It is particularly irritating that the authors begin the section correctly noting the evidence for cis but the vast majority of the presented numbers afterwards relates to trans associations.

For the colocalization chapter, in addition to the number of pairs where variants associated with protein levels and phenotype, we are reporting the suggested number of variants or highly related clusters, both in section *Cis pQTLs on SomaScan and Olink and their colocalization with disease-associated variants* (page 14) and Supplementary Tables S26 and S27. We are now also splitting the count into *cis* and *trans* to have a clearer representation of the data.

Previous proteomic studies were excluded from the GWAS catalog data and this is now reflected in the text.

We have now also reworded the chapter *Cis pQTLs on SomaScan and Olink and their colocalization with disease-associated variants*:

“Colocalization of a variant associating with a disease and a cis pQTL makes it likely that the variant is mediating risk through the associated protein. We identified all variants reported in the NHGRI-EBI Catalog of human genome-wide association studies (GWAS catalog) in high LD ($r^2 > 0.80$) with sentinel pQTLs based on Olink UKB-BI data and Icelandic SomaScan data (Supplementary Tables S26 and S27).

For the Olink UKB-BI data, there were 669,045 such pairs where variants in high LD associated with levels of a particular protein and a phenotype in the GWAS catalog (excluding proteomics studies); 4,439 pairs in cis and 664,606 pairs in trans. For the Icelandic SomaScan data, there were 300,067 such pairs; 3,360 in cis and 296,707 in trans. Some pQTLs associate with many proteins and phenotypes. On Olink and SomaScan, counting only the unique pQTLs yields 622 and 544 cis pQTLs and 3,147 and 1,992 trans pQTLs, respectively, where at least one disease colocalizes with the levels of at least one protein.

For each sentinel pQTL association, we also identified a 95% credible set of variants (variants that most parsimoniously explain regional association) likely to include the causal variant, as previously described by Maller et al.³⁵. We then checked whether GWAS catalog variants in high LD with the pQTL variant (with $r^2 > 0.8$ with the pQTL) were included in the credible set. For the pQTL associations with GWAS catalog entries detected using LD in the Olink UKB-BI data, 2,409 of 4,439 (54%) of the cis pairs and 529,604 of 669,045 (79%) of the trans pairs also have the GWAS catalog variant included in the credible set of the pQTL association. In the Icelandic SomaScan data, the same was true for 1,597 of 3,360 (48%) of the cis pairs and 196,836 of 296,707 (66%) of the trans pairs (Supplementary Tables S26, S27 and S28).

Furthermore, for the selected examples below, we estimated the posterior probability (PP) of statistical colocalization for the variants associating with disease and protein levels when they were not identical and when we had access to the necessary statistics.”

We also added the following section on *trans* pQTL examples to section *Proteins targeted on both platforms*.

“Based on either or both platforms, trans pQTLs can point to potential biomarkers of diseases. Using SomaScan data, we have previously noted that variants associating with psoriasis also associate with levels of DEFB4A, a protein highly expressed in skin, pointing to a potential disease biomarker. Of these variants, the variant most significantly associated with disease and protein levels at the IL12B locus (rs12188300) was also detected as trans pQTL of DEFB4A in plasma based on Olink. The levels of DEFB4A are highly correlated between the two platforms ($r=0.81$) and cis pQTLs are observed on both platforms.

Several trans pQTLs for PRSS2 at different loci are in high LD with diabetes-associating variants. PRSS2 encodes trypsinogen, a protein highly expressed in exocrine pancreas, and the trans pQTLs may reflect damage of the pancreas among diabetic patients (Supplementary Tables S26 and S27). The levels of PRSS2 are highly correlated between the two platforms ($r=0.78$) and cis pQTLs are observed on both platforms.”

Q3.12: A fundamental statistical difference between the UKB-PPP and the present effort is the identification of secondary signals. Although the authors use the same method as previously described, I have some concerns how valid exact conditional analysis in the context of massive effect sizes work. How strongly are secondary signals correlated with sentinel signals, and how much do marginal effect sizes drop once the authors compute a joint model for the entire locus? Given that many downstream analyses using the provided data will include a MR framework of some sort, validity of instrument selection is an important issue.

As the reviewer suggested, we have added to the pQTL tables the r^2 to the sentinel signal as well as the effect and significance based on a joint model for the locus.

We have now added the following to the conditional analysis methods section.

“We observe that 92% and 97% of secondary variants have an r^2 below 0.2 and 0.5 to the primary variant on Olink, respectively (based on r^2 calculated in the UK Biobank British/Irish set).

In addition, we estimated significance and effect based on a joint model of all variants at the locus to the phenotype for the variants selected in the stepwise model. When jointly estimating the effect on a protein at a locus, and examining pQTLs at loci that contains more than 1 associated variant to a protein, 96% and 92% of the associations detected using SomaScan and Olink respectively, remained significant when using the same genome wide significance threshold as in the stepwise model (i.e. 1.8×10^{-9}).”

Minor:

Q3.13: - I do not understand the very first sentence of the results. Isn't the CV already a measure of precision? This data is also somewhat contradicting to a recent study by Katz et al. (Science Advance 2022) who showed higher precision of the SomaScan assay and it remains unclear how much differences in sample handling and cohort characteristics might have contributed to this data. How does this comparison looks within DeCODE only?

Precision refers to how similar repeated measurements will be, while CV is the standard deviation of measurements divided by their mean. As such, CV is not a measure of precision – if a platform were to produce random values in a tight range not including 0, it would have low CV but also low precision depending on the scale of the potential measurements.

The recent study reported by Katz et al., published after our initial submission of the manuscript, compares the CV of the assays without taking into account the expected distribution of values if the measurements were not correlated. Using the set of 1,514 Icelandic samples measured using both Olink and SomaScan we replicate the results of Katz et al., demonstrating the lower CV of the SomaScan assays than the Olink assays (Extended Data Table 1). However, we also demonstrate the difference in assay CV between the platforms if the samples were not correlated, leading to the conclusion of lower CV ratio for the Olink assays than the SomaScan assays.

Q3.14: - Please do not use the term ‘predicted’ for effect estimates in Cox-models as it is not clear whether protein measurements really help to identify more future patients.

In order to simplify the comparison between phenotype associations for the two data sets, we now use logistic regression for all protein-phenotype associations. Thus, we do not use the term ‘prediction’ any more.

Q3.15: - When reporting the fraction of trans-pQTLs, it would be important to understand how large the contribution from unspecific trans-pQTLs might be, that are the sole associations for possibly very many SOMAmers.

As suggested by the reviewer, we have calculated the corresponding statistics and added the following to the manuscript (section *Detection of pQTLs*):

“Based on Olink, 52% of sentinel trans pQTL associations are with a variant associating with more than 10 proteins (unspecific pQTL), while based on SomaScan, 63% of *trans* pQTL associations are with such a variant. Consequently, 211 out of all 2,616 proteins with at least one pQTL on Olink (8%) only have association with unspecific *trans* pQTLs and 1,282 out of 4,574 proteins with at least one pQTL (28%) only had such an association on SomaScan.”

Conclusions

Q3.16: The conclusions drawn by the authors are mostly well justified. I would, however, suggest refining the statement about orthogonal validation of protein targets by cis-pQTLs. Strong cis-pQTLs might very well obscure protein measurements as has been shown previously (Pietzner et al. 2021).

The discussion chapter has been extensively rewritten to reflect the incorporation of the Olink Expansion set of assays as well as the changes due to the revision of the manuscript. We are now including a substantial discussion of the impact of *cis* pQTLs on protein measurement.

Specifically in the Results and the Discussion we now mention the following:

In the Results section:

“As we have previously observed, the effect of the alternative allele of *cis* pQTLs not in high LD with PAV is on average close to zero². However, when a PAV is in high LD with a *cis* pQTL, we observe a negative effect for the alternative allele and the negative effect tends to be even stronger when no *cis* eQTL is in high LD. When the PAV in high LD is annotated as high impact (stop gained, frameshift, splice donor or acceptor), the negative effect is consistent with expected nonsense mediated decay, in particular when supported by *cis* eQTL. However, when the PAV in high LD is annotated as moderate impact (missense, inframe indel or splice region), the negative effect we observe for the alternative allele is consistent with binding artefact. We observe very similar results on both

platforms. We conclude that the 23% of sentinel Olink cis pQTLs and 24% of sentinel SomaScan cis pQTLs that are in high LD with a moderate impact PAV but not with a cis eQTL are strong candidates for being caused by binding artefacts (Figure 6, Extended Data Figure 8)."

In the Discussion section:

„Even where cis pQTLs provide evidence that both platforms are measuring the targeted protein, in more than half of the cases, the top associated variants were in low LD. While cis pQTLs provide strong evidence that the protein being measured is in fact encoded by the gene of interest, they do not indicate which proteoform is being measured. The difference in pQTLs between the two platforms is consistent with proteoforms being differentially targeted by the platforms, as suggested by previous work on smaller sample sizes, both in terms of individuals and proteins⁷. Proteoforms encoded by the same gene may participate in different biological processes and therefore have different associations with diseases and other traits.

Furthermore, some cis pQTL, may correspond to binding artefacts rather than protein levels, particularly when the cis pQTL correlates with a coding variant (occurring in around 30% of cis pQTLs) and in the absence of a correlated eQTL^{2,7}. These artefacts may affect the platforms differently. We have not systematically performed assays such as ELISA or other methods for each of the proteins or proteoforms studied, as such validation is currently difficult to perform at scale.“

Suggested improvements

1) A general weakness of the study are the limited insights from these amazing data sets.

Q3.17: Further analysis on 1) what characterizes genuine pQTLs,

We have increased the analysis on that issue and have more results for both platforms.

As we had concluded before, *cis* pQTLs are less likely to be artefactual if not in LD with protein altering variant and if in LD with a *cis* eQTL for the gene encoding the protein. This is particularly true if the protein altering variant is a missense variant in contrast to a high-impact variant (i.e., stop-gained, essential splice or frame shift).

We have compared the direction of the effects of alternative alleles of *cis* pQTLs between sets of pQTLs to identify artefacts. For each of the two platforms, we have assessed this metric stratified by whether the pQTLs are in high LD with *cis* eQTLs, moderate impact PAVs, or high impact PAVs. pQTLs in high LD with moderate impact PAVs but not *cis* eQTLs are much more likely to have alternative alleles with a negative effect on protein levels than other pQTLs, indicating that these are highly enriched for artifacts.

For the same gene, multiple proteoforms might coexist and thus pQTL for these separate proteoforms might have distinct cis pQTLs.

We emphasized these points in the revised results and discussion, and have added supporting figure:

“As we have previously observed for SomaScan, the effect of the alternative allele for cis pQTLs not in high LD with a PAV is on average close to zero. However, when a PAV is in high LD with a cis pQTL, we observe a negative effect for the alternative allele and the negative effect tends to be even stronger when no cis eQTL is in high LD. When the PAV in high LD is annotated as high impact (stop gained, frameshift, splice donor or acceptor), the negative effect is consistent with expected nonsense mediated decay, in particular when supported by cis eQTL. However, when the PAV in high

LD is annotated as moderate impact (missense, inframe indel or splice region), the negative effect we observe for the alternative allele is consistent with binding artefact. We observe very similar results on both platforms. We conclude that the 23% of sentinel Olink *cis* pQTLs and 24% of sentinel SomaScan *cis* pQTLs that are in high LD with a moderate impact PAV but not with a *cis* eQTL are likely to be caused by binding artefacts (Figure 6, Extended Data Figure 8)."

Q3.18: (2)) how can differences between both assays highlight relevant biology, or affect downstream phenotypic studies would be needed to provide an advance in our understanding in which situations to trust which assay, but more importantly how those assays can yield biologically and clinically relevant insights.

For proteins targeted by both methods, we show that for a large fraction, the levels measured by the two platforms are not correlated at all. We also show that pQTLs (including *cis*) are sometimes only present on only one platform and when *cis* pQTLs are present on both platforms, we show that they often are not the same.

Additionally, phenotype-protein associations sometimes differ between the platforms.

A consequence of this is that colocalization between pQTLs and disease-associated variants can lead to different biological insights (e.g., IL10 and IBD, CD58 and MS, IL2RB and asthma on Olink and SomaScan).

The protein-phenotype associations will be influenced by similarity or differences between platforms as demonstrated through the examples of ADM, NEFL, and PTGDS with cardiac-, neurological-, and immunological diseases, respectively.

Q3.19: (2)) The availability of different ancestry is a clear strength of this study, but the current presentation of results remains superficial showing only two well-known examples. This section can certainly improve.

We have drastically expanded the section on different ancestry groups as suggested by the reviewers. We have added overall statistics in text and tables discussing the pQTLs with variants enriched or only present in different ancestries and show that one third of pQTLs in African involve variants absent or much rarer in Europeans (ancestry-specific variants). We also show that whereas sometimes the pQTL in different ancestries involve variants in high LD in these groups, they may represent LD classes ($r^2 > 0.8$ to sentinel variant in each ancestry) of different size between ancestries, allowing refinement of the signal. This refinement of pQTLs can aid refinement of disease associating variants if colocalizing with the pQTLs in the ancestral group with less resolution.

The section *pQTL analysis in different ancestry groups* now reads as follows:

"Studying genetics in different ancestry groups allows the assessment of greater sequence diversity and possibly perform fine mapping by distilling the signal to fewer markers¹⁴. Recently, Katz and co-workers used the Olink Explore 1536 platform to analyze the levels of 1,472 proteins in the plasma of 489 individuals of African ancestry¹¹. Using the same cutoff for significance ($p < 1.8 \times 10^{-9}$) for the same 1,472 proteins we here report *cis* pQTLs in the UKB-AF data set ($n = 1513$ individuals) for 628 proteins whereas they detected only 307 using an approximately three times smaller sample size. We detected *cis* pQTLs for 301 of these 307 proteins. Furthermore, Zhang and co-workers have reported the analysis of plasma protein levels of 1,871 individuals of African ancestry using the SomaScan platform (4,437 targeted proteins analyzed). Of these, 1,746 proteins are also targeted by Olink Explore 3072, which we used to analyze the UKB-AF group ($n = 1513$). Using the same cutoff for significance ($p < 1.8 \times 10^{-9}$), and considering the 1,746 overlapping targets, in these two sets of similar

population size we detect *cis* pQTLs for a similar number of proteins using Olink (667) as they did using SomaScan (671) (chi-sq $p=0.92$), of which only 417 are common.

Of the top *cis* pQTLs identified in the African or South Asian ancestry groups, around 32% and 4% of variants respectively were absent from or extremely rare in the European ancestry group (Supplementary Tables S17 and S18). For example, the predicted loss-of-function variant Cys679Ter in PCSK9, which has been associated with low levels of LDL cholesterol (-0.92 SD per copy)^{14,25}, was carried by 1 in 50 participants of African ancestry but was almost absent from other participants and associated with 2.1 SD lower levels of PCSK9 ($p=6.1 \times 10^{-33}$). This is consistent with Cys679ter preventing the secretion of PCSK9²⁶. Also, the sickle cell anemia variant Gly7Val in HBB (hemoglobin sub-unit beta)²⁷ was almost exclusively seen among participants of African ancestry (MAF=7.5%) where it associated in trans ($p=2.9 \times 10^{-12}$) with 0.50 SD higher levels of HMOX1 (heme oxygenase-1), an enzyme that degrades free hemin. Hemin is released intravascularly in sickle cell disease and is a known inducer of HMOX-1²⁸.

In the African ancestry group, *cis* pQTLs are in high LD with fewer variants on average (12) than in the European or South Asian ancestry groups (37 and 29, respectively), consistent with less LD in populations of African ancestry than other populations. Of 893 proteins with *cis* pQTLs in both European and African ancestry groups, for 324 (36%), the top *cis* pQTLs are in high LD ($r^2 > 0.8$) in the European ancestry group. Of these 324, there was substantial refinement of the *cis* pQTL locus of 62 proteins where the top *cis* pQTL in the African ancestry group is in high LD with 5 or fewer variants but with 15 or more variants in the European ancestry group (Figure 3a). For example, rs6794768 is the top *cis* pQTL for SERPINI2 in the European and African ancestry groups, but in the latter the signal is refined to drastically fewer markers than in the former (Figure 3b). At the CD58 locus, the sentinel *cis* pQTLs in the European and African ancestry groups were in high LD ($r^2=0.96$ in the European ancestry group). The number of highly correlated markers is much smaller in the African ancestry group than in the European (3 vs 37 variants). Since the pQTL in the European ancestry group associates with multiple sclerosis, and the pQTL in the African ancestry group allows refinement, this means that the disease association has been similarly refined (Figure 3c).“

Q3.20: (3)) Although the authors compare the two largest sets available, some of the difference might be driven by ancestry-specific effects between deCODE and UKB, and comparing UKB-PPP results to a matching European study with SomaScan might help to understand those differences better.

In order to isolate the differences driven by the two distinct populations, we have measured a set of 1,514 Icelanders with Olink on the same samples as we previously had with SomaScan. While we have already used this set to investigate the replication of the pQTLs detected in the UKB-BI, this can also be used to investigate what the reviewer refers to as ‘ancestry-specific effects between deCODE and UKB’.

For *cis* pQTLs, the replication observed is close to what would have been expected by the tested sample size in Iceland, indicating that for *cis* pQTLs, differences between the populations do not substantially affect the results.

However, for *trans* pQTLs, the fraction of pQTLs that replicate is less than for *cis* pQTLs, and may suggest that *trans* pQTLs are more sensitive to population differences such as we observe in the fraction of individuals with disease, as well as differences in sample handling and processing as has been previously reported (Ferkingstad et al., PMID: 34857953; Pietzner et al., PMID: 33328453).

It has been noted that a small fraction of invitees has participated in UKB with a bias towards healthy individuals, while the Icelandic data set includes samples collected for the study of specific diseases (such as cancer) and is therefore somewhat enriched with regards to those.

The chapter describing the replication of pQTLs detected in UKB-BI in the Icelandic Olink data now reads as follows (section *Detection of pQTLs*, page 10):

“We tested the sentinel pQTLs we discovered using the UKB-BI data set for association using Olink data from 1,514 Icelanders and found 74% and 19% of the *cis* and *trans* pQTL associations to be nominally significant ($p < 0.05$) and with consistent directions (Supplementary Table S16). In comparison, a power analysis resulted in expected proportions of replicated associations of 85% and 35% for *cis* and *trans* pQTL associations, respectively. The ratio between expected and observed replication was 0.87 and 0.54 for *cis* and *trans* pQTLs, respectively, and while the power analysis represents a best-case scenario, the replication rate may also be impacted by differences in population and sample handling and processing (Extended Data Table 9).”

Q3.21: (4)) A major message of this study is that the higher coverage of the SomaScan assays is its surplus. With another 1.5k proteins on the horizon for UKB, to which the authors will have preferential access, I am wondering whether some of the conclusions drawn here will still hold and whether it might not be better to wait and have a fairer comparison also in terms of coverage.

As suggested by the reviewer and the editor, we have now revised and changed the paper to include Olink Explore 3072 (1536 and Expansion) instead of Olink Explore 1536 only.

The relevant sections of the comparison chapter (section *pQTL comparison between Olink and SomaScan*, page 11-12) now read:

“Cis pQTLs were present for 2,101 (71%) Olink assays and 2,120 (43%) SomaScan assays (Table 1). Thus, while a very similar number of proteins with cis pQTLs were detected with the two platforms (2,093 on Olink and 2,044 on SomaScan), a higher fraction of proteins on Olink than on SomaScan had cis pQTLs. On both platforms, most assays had pQTLs: trans pQTLs were present for 2,528 (86%) and 4,716 (95%) Olink and SomaScan assays, respectively. There were more trans than cis sentinel pQTLs associations on both platforms, but a larger number of secondary associations in cis than in trans (Table 1, Supplementary Tables S9 and S12).

[...]

When we restricted our analysis to matching assays for 1,848 unique proteins targeted with both platforms, the Olink assays were more likely to have a cis pQTL (80% of 1,864) than the SomaScan assays (58% of 1,994) (Table 1).”

Q3.22: (5)) Some text of the Supplemental Notes is repetition from the main text.

The Supplemental Notes expand upon results presented in brief in the main text. In some cases, the results presented in the main text need to be restated for context or for comparison purposes. We have gone through the Supplementary Notes in order to minimize the unnecessary redundancies.

Minor:

Q3.23: - Many figures are of poor quality or miss proper legends. For example, the black background on figure 4 does not really help to distinguish points or figure 5 should rather be a stacked locus zoom plot instead of a 2x2 panel.

We have attempted to improve the clarity of figures and have updated them accordingly.

References

Q3.24: References look appropriate, although the authors seem to overly emphasize their own previous work.

We have added several references to the works of others, some of which were pending publication at the time of initial submission.

Clarity and context

Q3.25: The authors have done a good job in pulling many statistical analyses together. However, this is also a general weakness of the work, as the reader is left with this number without meaningful interpretation. Most importantly, to really represent a conceptual advance this work needs to dig deeper into phenotypic and ancestry-specific links towards the proteins investigated.

We have now made large modification to the manuscript in order to represent the changes in the coverage of the Olink platform. Furthermore, as suggested by the reviewer we have added to the results on phenotypes and analyse and discuss them in more detail.

Concerning the analysis of different ancestries, we have made vast efforts and changes in order to assess the ancestry specific association as well as the locus refinement enabled by cross-ancestry analysis in UKB.

Q3.27: (1) The presentation and discussion around the normalized vs non-normalized data for SomaScan is somewhat vague and would benefit from a clearer presentation. At least to me, it looks like presenting data on normalized results only and discussing non-normalized results in the Supplement would improve the paper.

As suggested by the reviewer, discussion of the impact of the SMP normalization suggested for the SomaScan data has been moved to Supplementary Note S2, where we have made an effort to further clarify the presentation.

Reviewer Reports on the First Revision:

Referees' comments:

Referee #1 (Remarks to the Author):

The significant work the authors put into the revisions is appreciated and the manuscript is much improved. A number of critical issues remain. Most of these can be addressed with textual changes or minor computational work. It is not expected that significant new analyses will need to be done to address the issues which are detailed below:

1. On lines 90-95 the authors seem to have conflated precision and accuracy. Precision is how consistently the value is measured as compared to accuracy which reflects how closely the values agree with the true accepted value. Their sentence "if a platform were to produce random values in a tight range it would have a low CV but no precision" is wrong; such a set of values would have a high precision but low accuracy and is frequently cited as the definition of such a scenario. This section needs to be rewritten.
2. In the section on "Protein-phenotype associations" the authors refer to associations with "sample age." This is confusing. I think they mean participant age and not the sample age – which would be how old the biological sample is (i.e. time from blood draw to analysis). Additionally, it is not clear why they report 2 sets of results for the percent of proteins associating with age – perhaps one is on univariate and the other multivariable analysis. This needs to be clarified.
3. To what degree are differences in LD structure from UKB-BI and Icelander populations responsible for the lower-than-expected rate of replication of UKB-BI sentinel pQTLs in the Icelanders and for differences in the lead variants between the analyses on the two platforms? It is possible that the tagging variants are different in the two populations and the underlying causal variant is the same (or not). This could be addressed by comparing credible sets rather than lead variants.
4. On lines 279-282 authors comment that "On the Olink platform, the majority of proteins already have a cis pQTL, suggesting that increased sample size may yield diminishing returns in terms of the number of additional proteins with a cis pQTL." In that the goal of pQTL analysis is to understand how genetic variation affects protein expression, this seems misguided. Finding a single pQTL is not the goal – rather it is fully understanding the landscape of cis-pQTLs at a locus. Are the authors able to comment on how sample size may affect the number of cis-pQTL for a protein? This idea that the goal is finding any cis-pQTL is pervasive throughout and the paper would benefit from some interation around this language.
5. The authors have nicely addressed concerns about differential sample sizes in the section on "Detection of pQTLs" for both its effect on discovery in UKB and in the replication of signals from Olink UKB-BI in Icelanders with Olink data. The section on "pQTL comparison between Olink and SomaScan" does not seem to acknowledge or adjust for the differences in sample size and the impact of that on discovery across the platforms. In some sense, it is this later section where it is most important to reckon with the differential sample size. For example, sample size seems to be ignored on line 330 where the authors are discussing the factors that influence the likelihood of finding a cis-pQTL.
6. It is appreciated that the authors have now attempted formal statistical colocalization in some instances. The current description of colocalization in the Methods now refers to statistical colocalization, but in the manuscript continues to be used more generally (variant in high LD). It is key that the terminology is differentiated and noted explicitly in the methods and results section.
7. There are lots of "results" in the Discussion section – would try to de-emphasize re-iterating specific numbers/statistics and focus on higher-level conclusions.
8. Please add a specific "Limitations" section to the Discussion

Referee #2 (Remarks to the Author):

The authors of "Comparison of large-scale immunoassay- and aptamer-based plasma proteomics through genetics and disease" have revised and improved the initial submission. They addressed my concerns accordingly and satisfactorily. My recommendation would be to accept this important and timely manuscript for publication after considering these final comments (editorial review only):

1) The substantial number of supplementary information should be made more accessible via a web interface where the community can browse the summary statistics for their favorite targets.

2) One key message of the paper includes the 500+ high-confidence proteins (tier 1) that pass all set criteria. These proteins deserve a figure that includes the criteria, estimated abundance, and organ of origin. The 3129 minus 500 protein (tier 2) set could include those where one of the two platforms provides high-confidence information and Tier 3, those with a lack of supportive evidence. These lists will benefit the community and could serve as starting proteins for validation by mass spectrometry methods.

3) Besides many associations and correlations, one valuable outcome is the rules that define the experimental data and analysis that increase the likelihood of providing platform-independent information. That is to say, that data from such proteins are not due to applying a particular method. Please prepare a systematic and guiding figure for this.

4) The discussion about proteoforms is excellent. I think caution must be taken when extrapolating the effects of proteoforms in the context of affinity-binding assays because alterations of a protein's backbone may have wider-reaching effects. In this context: The authors use "binding artifacts" to describe "epitope effects". As discussed in PMID: 32860016, affinity binding to a target of interest depends on many factors. Alterations of the antigen sequence may (1) reduce the binding affinity and detected levels while maintaining a binder's specificity. Alterations can influence (2) post-translational modifications, (3) structure, and (4) change in the portfolio of interaction partners. These will govern the accessibility of the epitope for the affinity reagent. In addition, differences in the abundance of the intended on-target and the most probable off-target matters for specificity assignment. Hence I would interpret the term artifact as a residual binding to any target. Given the link to proteoforms, I suggest reconsidering the used terminology.

Other:

- Check the platforms' IQRs for each protein data in the two modes of Fig 2b. IQR could provide another informative measure for judging whether a protein is measured reproducibly, above LOD, and with high specificity.
- Fig 5 was still challenging to read and interpret.
- The abstract and manuscript are still lengthy and could benefit from being shortened to keep the focus on the most informative items.

Thank you for the opportunity to review this work.

Jochen Schwenk

Referee #3 (Remarks to the Author):

Reviewer Comments

The study by Eldjarn and Ferkingstad et al. clearly improved upon review, in particular by the

inclusion of the full Olink Explore 3072 assay. The authors present a detailed comparison between both technologies, including paired measurements, observational associations, and genetic discovery, with interesting insights about potential disagreements and a putative synergistic nature of both assays, and the authors should be congratulated for this service to the community. The improved trans-ethnic analysis is further to be highlighted.

However, the study still reads as a merely technical report with little conceptual advance as core conclusions have already been outlined by others or the authors themselves, e.g., the effect of differential cis-pQTLs, poor correlations and divergent findings. Also, the pQTL discovery aspect is not put into context, in terms of how many variants have already been reported or, most importantly, the rather underwhelming link of cis-pQTLs to disease outcomes, with ~500 reported for each platform, which is very similar to previous efforts and the added value appears incremental. The most important question about the relevance and role of all the newly discovered trans-pQTL and how they can guide biological investigations remains largely untouched. Which is also exemplified by the resistance of the authors to perform more appropriate statistical analysis for causal inference. In its current form the paper still reduces to an exercise in reporting numbers, which are certainly of technical value, but I find it hard to derive a biological breakthrough from the presented results. Some more specific comments below.

- 1) The authors should stop using the term 'colocalisation' for the overlap done between the GWAS catalog and pQTLs, the added fine-mapping probabilities certainly help, and the corresponding section should be reduced to variants with evidence for both, LD and fine-mapping, and numbers reported accordingly.
- 2) The authors present the GWAS in Icelandic data almost like a novel discovery effort, although they clarify that the data has been published early, I am wondering whether this 'correction' of their early Nature Genetics paper deserves credit in such an outstanding journal like Nature.
- 3) The supplemental material is poorly organized with missing headers and annotations.
- 4) The criteria to define cross-platform replication is not sufficient, since it is easily imaginable that very strong effects dilute along an LD gradient, that leads even to nominal significant p-values even for unrelated SNPs. The authors should adopt a more rigorous approach in comparing respective lead and secondary signals using at least LD values.
- 5) Line 394: The authors introduce the section with the biological relevance of cis-pQTLs but then go on to report mostly trans-pQTL "colocalisations" (4k in cis vs 600k in trans; which almost certainly includes double counting). It is completely unclear what trans-pQTLs in high LD with GWAS catalog variants might imply and the authors need to do more digging on bringing sense to these associations. In particular, since overlap between cis-pQTLs and GWAS catalog variants has been shown by multiple other groups already and is not even clear whether any of the findings highlighted are novel.
- 6) Even for cis-pQTLs and given the fairly large coverage by the Olink Explore 3072 platform, it remains to be determined how often cis-pQTLs are indeed uniquely identifying causal genes. For example, variants like rs10774624 that are annotated as cis-pQTL for a number of proteins (e.g., BRAP, PPP1CC, or ERP29)?
- 7) The authors should make the code available, including how power calculations have been performed.

Minor

- line 115 what is PN? The very first section is very long and might be better shortened to a section of not more than a small paragraph.
- Line 136: Does the very strong median within platform correlation for SomaLogic imply that w/o SMP normalization, the data should rather not be used, given that there is a strong underlying latent factor?
- The section on tissue-based correlations should be omitted, since it is not clear whether the proteins indeed originate from each (e.g., testis in women)

- The head-to-head comparison of sample age between Olink and SomaScan is misleading, since they originate from different cohorts as far as I understand.
- Why are observational associations stratified by ethnicity? I get the reason for genetic analysis but would assume that one would need to provide evidence of an interaction effect to justify splitting groups. Also how were the analysis in UKB adjusted for the sampling character, that is, random vs consortia selected cases?
- The section "Protein-phenotype associations" is very long with little information overall and written in a very repetitive manner.
- What explains the drop in the number of pQTLs discovered with Olink Explore and Expansion?
- Line 319 onwards: the conclusion for MS does not hold, since the authors do not establish colocalization of the MS signal in people of African American descent
- Line 349: Does this imply that SomaScan is the more reliable technology?
- Line 362: I do not understand this sentence. Why is the effect of the alternative allele for a cis-pQTL close to zero when it is not in LD with a PAV? Is this because true effects of the alternative allele cancel out across proteins having positive and negative effects? Please clarify.
- Line 532: The conclusion that the absence of a cis-pQTL indicates that the protein is not well measured in plasma is not quite right. I might be wrong but peptide hormones such as insulin do not have a cis-pQTL either and we have no doubt that they are actively secreted and well-measured.
- Line 597: This section is somewhat redundant to the introduction.
- Line 772 Why is the testing threshold adjusted to a different number of variants compared to what is shown in line 694?
- Prior settings for COLOC should be given, possibly adapting what has been recently proposed (Wallace 2020 Plos Genetics)

Figures:

- Fig 4 adds little and should go into the Supplement.
- Fig 5 is very low resolution and hard to read, colouring bins to indicate spearman correlations would work better than a black to red gradient.
- Why is a prettier version of the top panel of EDF 7 not in the main text?
- The colour gradient in stacked locus zoom plots is hard to distinguish due to the thick frame around dots, pls omit this.

Misc

- Extended table 9 is not well formatted (age all lower case) and bias 'health' / 'disease' is not really a description of a population; also what does 'delayed' sample handling in UKB imply?

Author Rebuttals to First Revision:

Referee #1 (Remarks to the Author):

The significant work the authors put into the revisions is appreciated and the manuscript is much improved. A number of critical issues remain. Most of these can be addressed with textual changes or minor computational work. It is not expected that significant new analyses will need to be done to address the issues which are detailed below:

1. On lines 90-95 the authors seem to have conflated precision and accuracy. Precision is how consistently the value is measured as compared to accuracy which reflects how closely the values agree with the true accepted value. Their sentence “if a platform were to produce random values in a tight range it would have a low CV but no precision” is wrong; such a set of values would have a high precision but low accuracy and is frequently cited as the definition of such a scenario. This section needs to be rewritten.

We thank the reviewer for pointing this out.

We have rewritten the chapter “Comparison of precision of the two platforms” to emphasize the interpretation of CV ratio, while pointing out both our replication of prior results using CV of control samples and the discrepancy between CV on control samples and CV ratio. The relevant sections now read as follows (pg. 5, l. 85-105):

“We evaluated the repeatability of assays using the ratio of the coefficient of variation (CV) of repeated measurements of the same sample to the CV of the assay (CV ratio). The CV ratio is similar to the ratio of the standard deviation of the repeated measurements to the standard deviation of the assay. A CV ratio of zero means that repeated measurements of a sample always give the exact same result, while a CV ratio of one means that the repeated measurements of the same sample are no more similar than measurements of unrelated samples.

We calculated the CV ratio for the assays using duplicate measurements of 1,474 samples in the UKB data set (Olink Explore 3072) and 419 samples in the Icelandic data set (SomaScan v4) (Figure 2; Supplementary Tables S1 and S2; Extended Data Figure 1). Based on all assays, the median CV ratio for Olink was lower than for SomaScan (0.35 and 0.50, respectively, Mann-Whitney $p=1.1\times 10^{-135}$) (Extended Data Table 3). Restricting to assays that target the same protein on both platforms, the

median CV ratio remained smaller for Olink than SomaScan (0.33 and 0.49, respectively, Mann-Whitney $p=4.7\times 10^{-93}$). Thus, the Olink assays are on average more precise than the SomaScan assays.

Both the Olink and SomaScan platforms include repeated measurements of control samples for quality control purposes. Prior research using CV of such control samples as a measure of precision has demonstrated lower CV for assays on a previous version of the SomaScan platform (1.2K) than on the Olink Explore 1536 platform^{12,13,15}. We observe similar results in our data: on control samples, the CV of SomaScan assays is on average smaller than the CV of Olink assays (Extended Data Table 2)¹². This is in contrast with SomaScan assays having higher CV ratios than Olink assays, indicating that the CV of assays on control samples does not necessarily reflect assay precision, most likely because direct comparison of CVs depends on accurate estimation of mean protein levels which CV ratios do not. ”

2. In the section on “Protein-phenotype associations” the authors refer to associations with “sample age.” This is confusing. I think they mean participant age and not the sample age – which would be how old the biological sample is (i.e. time from blood draw to analysis). Additionally, it is not clear why they report 2 sets of results for the percent of proteins associating with age – perhaps one is on univariate and the other multivariable analysis. This needs to be clarified. We have reworded the text to remove this ambiguity. Protein levels in plasma have previously been demonstrated to correlate with sample age (PMID: 27596149). Therefore, in line 174, “sample age” does in fact refer to the time from the blood draw until the proteomics measurement of the sample, it corresponds to the duration of storage of the biological sample. The distribution of the sample age for the two data sets is described in Extended Data Table 9. The section “Protein-phenotype associations” has now been updated to explicitly refer to “subject age” and “sample age”, as well as include a reference to the previously cited article. The “subject age” being defined as the age of the subject at the time of the sample collection.

The relevant parts of the section now read (pg. 7-8, l. 174-186):

“Plasma protein levels have previously been shown to correlate with sample age¹⁷. Sample age, time from blood draw to measurement, correlated significantly with protein levels for a considerable number of assays on both platforms, although differences in the distribution of sample age between cohorts make direct comparison difficult (Supplementary Tables S5 and S6, Extended Data Table 9). The effects of sample age were generally small for both platforms and the correlation between effects was low (Spearman $r=0.08$). The levels of 77% of SomaScan assays associated with subject age at sample collection, 64% with sex, and 69% with BMI (Supplementary Tables S5-S8). In UKB-BI, the levels

of 60%, 68%, and 78% of Olink assays associated with subject age, sex, and BMI respectively, and the effects correlated well between ancestries in UKB (pairwise Spearman correlation greater than 0.70, 0.71, and 0.87 for subject age, sex, and BMI respectively; Extended Data Figure 6; Supplementary Tables S7 and S8). The Spearman correlation of subject age and sex effects between the 2,021 pairs of matching assays on Olink and SomaScan was 0.52 for subject age, 0.56 for sex, and 0.43 for BMI. “

3. To what degree are differences in LD structure from UKB-BI and Icelander populations responsible for the lower-than-expected rate of replication of UKB-BI sentinel pQTLs in the Icelanders and for differences in the lead variants between the analyses on the two platforms? It is possible that the tagging variants are different in the two populations and the underlying causal variant is the same (or not). This could be addressed by comparing credible sets rather than lead variants.

As suggested by the reviewer, we have analyzed the two cohorts to see if the observed difference in replication between the cohorts might be caused by differences in LD structure.

For sentinel pQTLs found in Iceland based on Somascan, we selected all the variants in the credible set in high LD ($r^2 > 0.8$) with the pQTL and assessed their r^2 in the UKB-BI to the same variant. For sentinel pQTLs found in UKB-BI based on Olink, we selected all the variants in the credible set in high LD ($r^2 > 0.8$) with the pQTL and assessed their r^2 in Iceland to the same variant.

Of the variants in high LD with the sentinel variant in Iceland ($r^2 > 0.8$), 92% were also in high LD in the UKB-BI cohort and a further 5% had r^2 between 0.6 and 0.8. Of the variants in high LD with the sentinel variant in the UKB-BI ($r^2 > 0.8$), 89% were also in high LD in Iceland and a further 8% had r^2 between 0.6 and 0.8. Thus, for both groups, 97% of the variants in high LD with the sentinel reported pQTL had $r^2 > 0.6$ in the replication cohort. It is therefore unlikely that the difference in replication rate between the platforms is because of difference in LD structure between the groups.

We have added this information as Supplementary Note S9 (pg. 80).

In our definition of replication of a pQTL, i.e., the significance of the association of the variant in the replication cohort exceeding a predefined threshold, we follow established practice (e.g., Sun 2018, Pietzner 2021). However, we are aware that this approach has some drawbacks, particularly in the presence of multiple secondary signals. While the direct replication of pQTLs between the platforms is not one of the primary aims of the article, we have further deemphasized the replication while still keeping it in the paper for comparison with other works. Instead, we have increased emphasis on the

correspondence of pQTLs between platforms, which we have expanded to include LD between not only sentinel pQTLs in both cohorts, but also between sentinel pQTL in the source cohort and all signals (sentinel and secondary) in the replication cohort.

To this end, the section “Cross platform replication” has been renamed “Relationship of pQTLs between platforms”. The section describing the replication has been moved to the latter part of the section, which now reads (pg. 14-15, l. 393-421):

“The presence of multiple independent signals at the same locus makes the comparison of pQTLs complicated, as the sentinel signal in one cohort may be a secondary signal in the other. To establish correspondence between pQTLs on the Olink and SomaScan platforms, we checked if the sentinel variant detected on one platform was in high LD ($r^2 > 0.8$) with any of the pQTLs (sentinel or secondary) at the same locus (within 5MB) on the other platform. In the UKB-BI Olink data, 581 (40%) out of 1,468 sentinel *cis* pQTL signals had a corresponding pQTL in the Icelandic SomaScan data, and in 434 cases (30%) the pQTL was the sentinel signal at the locus. In the SomaScan data, 559 (48%) out of 1,164 sentinel *cis* pQTL signals had a corresponding pQTL in the UKB-BI Olink data, and in 449 cases (39%) the pQTL was the sentinel signal at the locus. Of the sentinel *trans* pQTL signals detected in the UKB-BI Olink data, 1,855 (10%) out of 18,578 had a corresponding pQTL in the Icelandic SomaScan data, and in most cases (1,777; 10%) the pQTL was the sentinel signal at the locus. Of the sentinel *trans* pQTL signals detected in the Icelandic SomaScan data, 1,918 (19%) out of 10,352 had a corresponding pQTL in the UKB-BI Olink data, and in 1,828 cases (18%) the pQTL was the sentinel signal at the locus (Supplementary Table S21). Thus, even when pQTLs for a protein are detected on both platforms, they are not necessarily the same. When the sentinel *cis* pQTLs on the two platforms are in high LD, the correlation between levels is higher than when they are not (median correlation 0.55 vs 0.49, Mann-Whitney $p=6.9 \times 10^{-6}$) (Supplementary Table S20).

In addition to the correspondence of pQTLs, we checked the replication of pQTLs detected on each of the platforms on the other platform^{6,7}. Of the pQTLs we identified using Olink UKB-BI, we nominally ($p < 0.05$) replicated 70% of sentinel *cis* pQTLs and 54% of sentinel *trans* pQTLs in the Icelandic SomaScan data (Supplementary Table S21). Conversely, we replicated 84% of sentinel *cis* pQTLs and 54% of sentinel *trans* pQTLs detected using SomaScan data in the UKB-BI Olink data (Supplementary Table S22). Thus, the replication rate for *cis* pQTLs detected using SomaScan in Olink data was somewhat higher than for those detected using Olink and tested using SomaScan data. Assays targeting proteins with *cis* pQTLs that did not replicate between platforms had lower correlation between measured

protein levels than assays for proteins with replicating *cis* pQTLs (Figure 5 a, b, d, e; Supplementary Tables S21, S22, S10 and S3; Supplementary Note S8). The difference in replication cannot be explained by difference in LD structure between the two populations (Extended Data Figure 11, Extended Data Table 10, Supplementary Note S9)."

4. The on lines 279-282 authors comment that "On the Olink platform, the majority of proteins already have a *cis* pQTL, suggesting that increased sample size may yield diminishing returns in terms of the number of additional proteins with a *cis* pQTL." In that the goal of pQTL analysis is to understand how genetic variation affects protein expression, this seems misguided. Finding a single pQTL is not the goal – rather it is fully understanding the landscape of *cis*-pQTLs at a locus. Are the authors able to comment on how sample size may affect the number of *cis*-pQTL for a protein? This idea that the goal is finding any *cis*-pQTL is pervasive throughout and the paper would benefit from some iteration around this language.

As well as explaining how genetic variation affects protein expression, the analysis of pQTLs can serve other purposes. The primary focus of our manuscript is the comparison of the two most common high throughput proteomics platforms currently available.

In the context of comparing the performance of the platforms, the detection of a *cis* pQTL for a particular protein measured using a particular assay is relevant as it indicates that the assay is detecting the correct protein. We have added this clarification of our intent to the pQTL result section, as well as emphasizing the reviewer's point that a single pQTL in most cases explains only a part of the genetically determined variation in protein expression.

Furthermore, we have added a section discussing the number of *cis* pQTLs for a protein.

The relevant section now reads (pg. 11, l. 291-302):

"Secondary *cis* pQTL associations were detected for 1,702 of 2,102 sentinel *cis* pQTLs on the Olink platform (81%) and for 1,594 of 2,120 on the SomaScan platform (75%). Secondary *trans* pQTL associations were detected for 3,340 of 24,824 sentinel *trans* pQTL associations on the Olink platform (13%) and 4,065 out of 22,616 on the SomaScan platform (18%).

While secondary signals help to understand how genetic variation affects protein expression, the mere existence of a *cis* pQTL for a protein on a particular platform provides evidence that the assay

is measuring the correct protein. On the Olink platform, the majority of proteins already have a *cis* pQTL. Furthermore, the significance of most *cis* pQTLs is well above the genome-wide threshold, suggesting that the number of *cis* pQTLs is unlikely to drastically change with increased sample size, although they may still provide valuable insights into the genetic control of protein expression. However, as most *trans* pQTL associations have significance rather close to the genome-wide threshold, expanding sample size is likely to reveal more *trans* pQTLs (Extended Data Figure 9)."

5. The authors have nicely addressed concerns about differential sample sizes in the section on "Detection of pQTLs" for both its effect on discovery in UKB and in the replication of signals from Olink UKB-BI in Icelanders with Olink data. The section on "pQTL comparison between Olink and SomaScan" does not seem to acknowledge or adjust for the differences in sample size and the impact of that on discovery across the platforms. In some sense, it is this later section where it is most important to reckon with the differential sample size. For example, sample size seems to be ignored on line 330 where the authors are discussing the factors that influence the likelihood of finding a *cis*-pQTL.

In the section "pQTL comparison between Olink and SomaScan", the focus is on the existence of a *cis* pQTL for a particular assay as evidence that the assay is in fact measuring the protein intended. Both the data sets under consideration are of sufficient size such that the number of assays with a *cis* pQTL does not vary much with sample size, as demonstrated by Extended Data Figure 9. We have added to the section a prefacing sentence emphasizing this (pg. 12, l. 344-346):

"In both the Icelandic and UKB-BI cohorts, the sample size is sufficiently large so that the number of proteins with a *cis* pQTL is not likely to change much by increasing it (Extended Data Figure 9)."

The discussion in line 330 was supposed to refer to the data sets under consideration, correlating the presence or absence of a *cis* pQTLs for a given assay with several factors in those data sets in particular. We have tried to make this clearer. The relevant sentence now reads (pg. 13, l. 353-355):

"The fraction of assays with *cis* pQTLs varied depending on several factors, including dilution group, sub-cellular location, and CV ratio, in both the Icelandic and UKB-BI set (Figure 4; Supplementary Table S19)."

6. It is appreciated that the authors have now attempted formal statistical colocalization in some instances. The current description of colocalization in the Methods now refers to statistical

colocalization, but in the manuscript continues to be used more generally (variant in high LD). It is key that the terminology is differentiated and noted explicitly in the methods and results section.

In the interest of clarity, we have renamed the relevant chapter “Relationship between pQTLs on Olink and SomaScan and disease-associated variants” and added a sentence outlining the various methodologies used.

The relevant section now reads (pg. 15-16, l. 437-460):

“The establishment of a relationship between a variant associating with a disease and a *cis* pQTL makes it likely that the variant is at least in part mediating risk through the associated protein. We use three methods to establish such relationship: high LD ($r^2 > 0.8$) between a pQTL and a disease-associated variant, inclusion of a disease-associated variant in the subset of variants in the credible set in high LD with the pQTL, and for specific examples where the necessary statistics are available, statistical colocalization.

We identified all variants reported in the NHGRI-EBI Catalog of human genome-wide association studies (GWAS catalog³⁴, excluding proteomics studies) in high LD ($r^2 > 0.80$) with sentinel pQTLs based on Olink UKB-BI data and Icelandic SomaScan data (Supplementary Tables S26 and S27). For each sentinel pQTL association, we also identified a 95% credible set of variants (variants that most parsimoniously explain regional association³⁵) likely to include the causal variant³⁶. We then checked whether GWAS catalog variants in high LD with the pQTL variant (with $r^2 > 0.8$ with the pQTL) were included in the credible set. For the Olink UKB-BI data, there were 2,409 pairs of GWAS catalog variants and *cis* pQTLs where the GWAS catalog variant was in high LD with the *cis* pQTL and included in the 95% credible set for the pQTL, while for the Icelandic SomaScan data there were 1,597 such pairs. In addition, there were 529,604 and 196,836 such pairs for *trans* pQTLs detected in the Olink UKB-BI data and the Icelandic SomaScan data, respectively (Supplementary Tables S26, S27 and S28, Supplementary Note S11).

On Olink and SomaScan, counting only the unique pQTLs yields 403 and 359 *cis* pQTLs, respectively, and 2,830 and 1,782 *trans* pQTLs, respectively, where at least one disease is related as described above to the levels of at least one protein.

In addition to high LD between the disease-associated variant and both the pQTL and a variant in the credible set, for the selected examples below, we estimated the posterior probability (PP) of statistical

colocalization for the variants associating with disease and protein levels when they were not identical and when we had access to the necessary statistics³⁷.”

7. There are lots of “results” in the Discussion section – would try to de-emphasize re-iterating specific numbers/statistics and focus on higher-level conclusions.

To address the reviewer’s comment, we have tried to remove unnecessary reiteration of previously presented results from the Discussion section.

8. Please add a specific “Limitations” section to the Discussion

We have added a subsection to the Discussion chapter. It reads as follows (pg. 23-24, l. 665-682):

Limitations

While both the Olink and SomaScan platforms are affinity-based, they differ in nature, as one is based on antibodies and the other on aptamers. This may affect how proteins are quantified in complex samples like plasma. The biochemical properties of orthogonal assays used for validation may need to be considered in the context of the properties of the two platforms.

Protein concentration varies between tissues and sample types⁹. The current work is limited to plasma and the results and conclusions may be specific to it. The analysis of other sample types using these platforms requires separate assessment which has begun for some of those, such as cerebrospinal fluid^{59,60}. Analysis of pQTLs in different sample types in large data sets is likely to be highly informative.

Whereas the current study attempts to assess the proteome of individuals of non-European ancestry, the sample size is still limited. The differences in genetic association with protein levels between the ancestries are of high interest and our results suggest that larger sample sizes in the cohorts of non-European ancestries will further our understanding of these differences.

While our study suggests that a part of the difference between platforms may lie in their sensitivity to different proteoforms, the contribution of the various proteoforms remains to be studied.

The platforms may be differently affected by epitope effects. We have not systematically performed assays such as ELISA or other methods for each of the proteins or proteoforms studied, as such validation is currently difficult to perform at scale.

Referee #2 (Remarks to the Author):

The authors of “Comparison of large-scale immunoassay- and aptamer-based plasma proteomics through genetics and disease” have revised and improved the initial submission. They addressed my concerns accordingly and satisfactorily. My recommendation would be to accept this important and timely manuscript for publication after considering these final comments (editorial review only):

1) The substantial number of supplementary information should be made more accessible via a web interface where the community can browse the summary statistics for their favorite targets.

To address the reviewer’s comment and increase the accessibility of the Supplementary Tables, we have added to the supplementary information a file containing a detailed description of the contents of each of the Supplementary Tables. Furthermore, summary statistics will be made available via the deCODE web page upon publication.

2) One key message of the paper includes the 500+ high-confidence proteins (tier 1) that pass all set criteria. These proteins deserve a figure that includes the criteria, estimated abundance, and organ of origin. The 3129 minus 500 protein (tier 2) set could include those where one of the two platforms provides high-confidence information and Tier 3, those with a lack of supportive evidence. These lists will benefit the community and could serve as starting proteins for validation by mass spectrometry methods.

Following the reviewer’s suggestion, we have added a column to Supplementary Table S20 with confidence tier, corresponding to if the protein has a *cis* pQTL on both platforms and high correlation in levels (tier 1), if it is not in tier 1 but has a *cis* pQTL on at least one platform (tier 2) or if it has *cis* pQTL on neither platform (tier 3).

Furthermore, we have added Extended Data Figure 17, outlining both the use of genomic information and inter-platform correlation to assess the performance of the matching assays, as well as the tissue of enriched expression, subcellular location and expected abundance (as represented by dilution group), both for the tier 1 proteins and all proteins targeted by both platforms. Consistent with what we found in each set, the tier 1 proteins are more often secreted and more abundant as reflected by dilutions on both platforms compared to all proteins targeted by both platforms.

We have added the following to the “Discussion” chapter (pg. 21-22, l. 607-615):

Using the presence of *cis* pQTLs on the two platforms, and the correlation in protein levels between them, the proteins targeted by the platforms can be organized into tiers by confidence (Extended Data Figure 17, Supplementary Table S20). Of all proteins targeted by either platform, about 500 had *cis* pQTLs on both platforms and strong correlation between protein levels as measured by the two platforms. These can be said to be measured with high confidence on these two platforms (tier 1). About 2,600 had a *cis* pQTL on at least one of the platforms but either lacked a *cis* pQTL on one platform, were not highly correlated between platforms, or both (tier 2). Finally, about a further 3,000 had a *cis* pQTL on neither platform (tier 3). We believe this classification of proteins can be useful in prioritizing the orthogonal validation of the assays, e.g., by mass spectrometry.

3) Besides many associations and correlations, one valuable outcome is the rules that define the experimental data and analysis that increase the likelihood of providing platform-independent information. That is to say, that data from such proteins are not due to applying a particular method. Please prepare a systematic and guiding figure for this.

We have included in Extended Data Figure 17, generated to address Q2.3, the accompanying caption, and the text added to the Discussion chapter (as quoted in the response to Q2.3) the criteria for each class of supporting evidence. We have furthermore emphasized the role of Supplementary Table S20 in assembling this information. However, we urge caution in the use of the term ‘platform-independent information’, as the results only imply independence for the two platforms considered here. Future results may very well be different for other platforms, in particular platforms using different methodologies (e.g., non-affinity-based). While the methodology should be generally applicable, the results cannot be expected to be general in the same way.

4) The discussion about proteoforms is excellent. I think caution must be taken when extrapolating the

effects of proteoforms in the context of affinity-binding assays because alterations of a protein's backbone may have wider-reaching effects. In this context: The authors use “binding artifacts” to describe “epitope effects”. As discussed in PMID: 32860016, affinity binding to a target of interest depends on many factors. Alterations of the antigen sequence may (1) reduce the binding affinity and detected levels while maintaining a binder's specificity. Alterations can influence (2) post-translational modifications, (3) structure, and (4) change in the portfolio of interaction partners. These will govern the accessibility of the epitope for the affinity reagent. In addition, differences in the abundance of the intended on-target and the most probable off-target matters for specificity assignment. Hence I would interpret the term artifact as a residual binding to any target. Given the link to proteoforms, I suggest reconsidering the used terminology.

Following the reviewer's suggestion, we have throughout the paper changed references to “binding artefact” to “epitope effect”.

Other:

- Check the platforms' IQRs for each protein data in the two modes of Fig 2b. IQR could provide another informative measure for judging whether a protein is measured reproducibly, above LOD, and with high specificity.

The correspondence between assay IQR and correlation with the matching assay on the other platform is discussed in Supplementary Note S4, to which we have added Extended Data Figure 17, showing the distribution of correlation coefficients and IQRs along with the corresponding histograms.

- Fig 5 was still challenging to read and interpret.

To improve clarity, Figure 5 has been regenerated with different colouring and larger text.

- The abstract and manuscript are still lengthy and could benefit from being shortened to keep the focus on the most informative items.

We have tried to remove redundancies from the manuscript in order to reduce its length and improve the focus. We are happy to work with the editor to further focus the manuscript on the key messages.

Thank you for the opportunity to review this work.

Jochen Schwenk

Referee #3 (Remarks to the Author):

Reviewer Comments

The study by Eldjarn and Ferkingstad et al. clearly improved upon review, in particular by the inclusion of the full Olink Explore 3072 assay. The authors present a detailed comparison between both technologies, including paired measurements, observational associations, and genetic discovery, with interesting insights about potential disagreements and a putative synergistic nature of both assays, and the authors should be congratulated for this service to the community. The improved trans-ethnic analysis is further to be highlighted.

However, the study still reads as a merely technical report with little conceptual advance as core conclusions have already been outlined by others or the authors themselves, e.g., the effect of differential cis-pQTLs, poor correlations and divergent findings. Also, the pQTL discovery aspect is not put into context, in terms of how many variants have already been reported or, most importantly, the rather underwhelming link of cis-pQTLs to disease outcomes, with ~500 reported for each platform, which is very similar to previous efforts and the added value appears incremental. The most important question about the relevance and role of all the newly discovered trans-pQTL and how they can guide biological investigations remains largely untouched. Which is also exemplified by the resistance of the authors to perform more appropriate statistical analysis for causal inference. In its current form the paper still reduces to an exercise in reporting numbers, which are certainly of technical value, but I find it hard to derive a biological breakthrough from the presented results. Some more specific comments below.

While others have discussed the lack of correlation between levels of some proteins measured using the two platforms and extended here to cover more proteins, analysis of the impact of differential detection

of pQTLs between the two platforms is only made possible by the existence of data sets of sufficient size. To the best of our knowledge, prior studies have indeed *not* discussed the effect of differential *cis* pQTLs between platforms, not having had the statistical power to do such analysis.

The fact that even in cases where proteins have pQTLs on both platforms the sentinel pQTL is the same in only 40% of cases has not been previously published.

The focus of the current study is not only the detection of pQTLs in the UKB data set, but also the comparison of the two proteomics platforms, and how the difference between the platforms can translate into different results and thus different interpretation of the relationship between protein levels in plasma and diseases and other traits.

1) The authors should stop using the term ‘colocalisation’ for the overlap done between the GWAS catalog and pQTLs, the added fine-mapping probabilities certainly help, and the corresponding section should be reduced to variants with evidence for both, LD and fine-mapping, and numbers reported accordingly.

As requested by the reviewer, we have removed the word colocalization except when referring to the statistical approach itself. The section now has the heading “Relationship between pQTLs and disease-associated variants”. We have added a brief description of the three methods we use to assess the relationship between pQTLs and disease-associated variants to the start of the section.

Furthermore, we have removed from the main text the section stating the number of pairs of variants in high LD, leaving the number of pairs where the variants are in high LD and the GWAS catalog variant is in the credible set. As we believe that the extent to which the two concur may be of interest to the reader, we have moved the number of variant pairs in high LD to Supplementary Note S11.

The relevant section now reads (pg. 15-16, l. 437-460):

“The establishment of a relationship between a variant associating with a disease and a *cis* pQTL makes it likely that the variant is mediating risk through the associated protein. We use three methods to establish such relationship: high LD ($r^2 > 0.8$) between a pQTL and a disease-associated variant, inclusion of a disease-associated variant in the subset of variants in the credible set in high LD with the pQTL, and for specific examples where the necessary statistics are available, statistical colocalization.

We identified all variants reported in the NHGRI-EBI Catalog of human genome-wide association studies (GWAS catalog³⁴, excluding proteomics studies) in high LD ($r^2 > 0.80$) with sentinel pQTLs based on Olink UKB-BI data and Icelandic SomaScan data (Supplementary Tables S26 and S27). For each sentinel pQTL association, we also identified a 95% credible set of variants (variants that most parsimoniously explain regional association³⁵) likely to include the causal variant³⁶. We then checked whether GWAS catalog variants in high LD with the pQTL variant (with $r^2 > 0.8$ with the pQTL) were included in the credible set. For the Olink UKB-BI data, there were 2,409 pairs of GWAS catalog variants and *cis* pQTLs where the GWAS catalog variant was in high LD both with the pQTL and included in the 95% credible set for the pQTL, while for the Icelandic SomaScan data there were 1,597 such pairs. In addition, there were 529,604 and 196,836 such pairs for *trans* pQTLs detected in the Olink UKB-BI data and the Icelandic SomaScan data, respectively (Supplementary Tables S26, S27 and S28, Supplementary Note S11).

On Olink and SomaScan, counting only the unique pQTLs yields 403 and 359 *cis* pQTLs, respectively, and 2,830 and 1,782 *trans* pQTLs, respectively, where at least one disease is related as described above to the levels of at least one protein.

In addition to high LD between the disease-associated variant and both the pQTL and a variant in the credible set, for the selected examples below, we estimated the posterior probability (PP) of statistical colocalization for the variants associating with disease and protein levels when they were not identical and when we had access to the necessary statistics³⁷."

2) The authors present the GWAS in Icelandic data almost like a novel discovery effort, although they clarify that the data has been published early, I am wondering whether this 'correction' of their early Nature Genetics paper deserves credit in such an outstanding journal like Nature.

It is not our intention to present the updated Icelandic SomaScan GWAS results as novel discovery. The discussion of the updated results in lines 237-240 only serves as a short summary. Updating these results is necessary to ensure a fair comparison between the two platforms.

The updated results differ from the initial ones in two respects: they include a larger number of tested variants and are also done on data transformed by SomaScan's recommended SMP normalization. As discussed in Supplementary Note S3, the SMP normalization adjusts to some extent out the total amount of protein in plasma, thus affecting the GWAS results. Specifically, as outlined in lines 1,204-1,212, while

this normalization can increase the power to detect pQTLs, it can also obscure their direction of effect, as in the case of ASGR1. The analysis of SMP normalized data should be considered complementary to the unnormalized data.

3) The supplemental material is poorly organized with missing headers and annotations.

Following the reviewer's comment, we have gone through the supplementary material to ensure the uniform formatting of headers and to make sure that title of each table is included in the header. Furthermore, we have added detailed annotation of column contents to the list of tables.

4) The criteria to define cross-platform replication is not sufficient, since it is easily imaginable that very strong effects dilute along an LD gradient, that leads even to nominal significant p-values even for unrelated SNPs. The authors should adopt a more rigorous approach in comparing respective lead and secondary signals using at least LD values.

The presence of multiple signals per locus makes the interpretation of replication of pQTLs difficult. We align ourselves with prior work (e.g., Sun 2018, Pietzner 2021) in reporting replication. However, we acknowledge the concerns of the reviewer and address them below.

We believe that the data that best address this comment can be obtained through the analysis of correspondence of sentinel and secondary pQTLs obtained in conditional analysis. We have therefore in the current manuscript shifted the primary focus to the correspondence of pQTLs, as a more thorough comparison of the pQTL signals. We have expanded our notion of correspondence to not only cover the LD (r^2) between the sentinel pQTLs but also LD between the sentinel pQTL in the source cohort and all pQTLs (sentinel or secondary) in the replication cohort. As the pQTL signals are determined using conditional analysis, this would serve to filter out spurious correlations of the type described by the reviewer. The conditional analysis performed in each group (stepwise as well joint model) reflects the LD structure in each of them.

To facilitate comparison with prior work, as mentioned in the text, we retain the replication results in the text but add the caveat that some of the apparent replications may be shadows of other signals.

The section header has furthermore been changed from “Replication of pQTLs” to “Relationship of pQTLs between platforms”, and the relevant parts of the section now read as follows (pg. 14-15, l. 393-421):

“The presence of multiple independent signals at the same locus makes the comparison of pQTLs complicated, as the sentinel signal in one cohort may be a secondary signal in the other. To establish correspondence between pQTLs on the Olink and SomaScan platforms, we checked if the sentinel variant detected on one platform was in high LD ($r^2 > 0.8$) with any of the pQTLs (sentinel or secondary) at the same locus (within 5MB) on the other platform. In the UKB-BI Olink data, 581 (40%) out of 1,468 sentinel *cis* pQTL signals had a corresponding pQTL in the Icelandic SomaScan data, and in 434 cases (30%) the pQTL was the sentinel signal at the locus. In the SomaScan data, 559 (48%) out of 1,164 sentinel *cis* pQTL signals had a corresponding pQTL in the UKB-BI Olink data, and in 449 cases (39%) the pQTL was the sentinel signal at the locus. Of the sentinel *trans* pQTL signals detected in the UKB-BI Olink data, 1,855 (10%) out of 18,578 had a corresponding pQTL in the Icelandic SomaScan data, and in most of the cases (1,777; 96%) the pQTL was the sentinel signal at the locus. Of the sentinel *trans* pQTL signals detected in the Icelandic SomaScan data, 1,918 (19%) out of 10,352 had a corresponding pQTL in the UKB-BI Olink data, and in most of the cases (1,828; 95%) the pQTL was the sentinel signal at the locus (Supplementary Table S23). Thus, even when pQTLs for a protein are detected on both platforms, they are not necessarily the same. When the sentinel *cis* pQTLs on the two platforms are in high LD, the correlation between levels is higher than when they are not (median correlation 0.55 vs 0.49, Mann-Whitney $p=6.9 \times 10^{-6}$) (Supplementary Table S20).

In addition to the correspondence of pQTLs, we checked the replication of pQTLs detected on each of the platforms on the other platform^{6,7}. Of the pQTLs we identified using Olink UKB-BI, we nominally ($p < 0.05$) replicated 70% of sentinel *cis* pQTLs and 54% of sentinel *trans* pQTLs in the Icelandic SomaScan data (Supplementary Table S22). Conversely, we replicated 84% of sentinel *cis* pQTLs and 54% of sentinel *trans* pQTLs detected using SomaScan data in the UKB-BI Olink data (Supplementary Table S23). Thus, the replication rate using Olink for *cis* pQTLs detected using SomaScan was somewhat higher than the replication rate using SomaScan for *cis* pQTLs detected using Olink. Assays targeting proteins with *cis* pQTLs that did not replicate between platforms had lower correlation between measured protein levels than assays for proteins with replicating *cis* pQTLs (Figure 5 a, b, d, e; Supplementary Tables S22, S23, S10 and S3; Supplementary Note S8). The difference in replication

cannot be explained by difference in LD structure between the two populations (Extended Data Figure 11, Extended Data Table 10, Supplementary Note S9)”

5) Line 394: The authors introduce the section with the biological relevance of cis-pQTLs but then go on to report mostly trans-pQTL “colocalisations” (4k in cis vs 600k in trans; which almost certainly includes double counting). It is completely unclear what trans-pQTLs in high LD with GWAS catalog variants might imply and the authors need to do more digging on bringing sense to these associations. In particular, since overlap between cis-pQTLs and GWAS catalog variants has been shown by multiple other groups already and is not even clear whether any of the findings highlighted are novel.

Following the reviewer’s suggestion, we have rewritten the relevant introductory section to put more emphasis on the number of *cis* pQTLs. The section now reads (pg. 16, l. 448-453):

“For the Olink UKB-BI data, there were 2,409 pairs of GWAS catalog variants and *cis* pQTLs where the GWAS catalog variant was in high LD with the pQTL and included in the 95% credible set for the pQTL, while for the Icelandic SomaScan data there were 1,597 such pairs. In addition, there were 529,604 and 196,836 such pairs for *trans* pQTLs detected in the Olink UKB-BI data and the Icelandic SomaScan data, respectively (Supplementary Tables S26, S27 and S28, Supplementary Note S11).”

We have also added to the chapter a discussion about the possible ways in which trans pQTLs may be associated with diseases.

The association of a *trans* pQTL with a disease-associated variant can be interpreted in three ways. First, the change in protein levels may be a consequence of the disease predisposed by the variant. Second, the variant may be affecting the disease risk through a protein encoded by the gene at the variant locus affecting another protein in the same pathway, reflected by plasma protein levels and the *trans* pQTL. Third, the variant may affect the protein levels and the disease risk independently of each other.

We propose that the existence of many *trans* pQTLs for the same protein, associating with the disease proportionally with the levels of the protein, indicates that the change in protein levels is a consequence of the disease.

Furthermore, to identify cases where the variant may be affecting the protein levels through interaction between proteins in the same pathway, we assessed if each protein affected by a *trans* pQTL interacts with the protein encoded by the closest gene to the variant according to the STRING database. For about

9% of the *trans* pQTLs in high LD with a disease-associated variant, the two proteins of interest are known to interact.

We take several examples of cases where such evidence exists for the mode of action of the *trans* pQTLs.

The relevant section now reads (pg. 18-19, l. 536-567):

“In addition to our observation of *cis* pQTLs associating with diseases and other traits possibly shedding light on pathological mechanisms, we also noted such association for a large number of *trans* pQTLs. The associations of a *trans* pQTL with a variant associating with a disease can be interpreted in three ways. First, the change in protein levels may be a consequence of the disease predisposed by the variant. Second, the variant may be affecting the disease risk through a protein encoded by the gene at the variant locus affecting another protein in the same pathway, reflected by plasma protein levels and the *trans* pQTL. Thirdly, the variant may affect the protein levels and the disease risk independently of each other.

When all variants associating significantly with the levels of a given protein also associate proportionally with the risk of a particular disease, we propose that the protein plays a role in the pathogenesis of the disease. When all variants associating significantly with a given disease also associate proportionally with the levels of a particular protein in plasma (*trans* pQTLs), and in the absence of the conditions described above, we propose that the change in protein levels is a consequence of the disease. We note that often, the *trans* pQTLs associate with a protein with enriched expression in the tissue affected by the disease.

The proteins associated with these *trans* pQTLs can point to potential biomarkers of diseases. Using SomaScan data, we have previously noted that variants associating with psoriasis also associate with levels of DEFB4A, a protein highly expressed in skin, pointing to a potential disease biomarker². Of these variants, the variant most significantly associated with disease and protein levels at the *IL12B* locus (rs12188300) was also detected as *trans* pQTL of DEFB4A in plasma based on Olink. The levels of DEFB4A are highly correlated between the two platforms ($r=0.81$) and *cis* pQTLs are observed on both platforms.

Similarly, several *trans* pQTLs for PRSS2 at different loci are in high LD with diabetes-associating variants. PRSS2 encodes trypsinogen, a protein highly expressed in exocrine pancreas, and the *trans* pQTLs may reflect damage of the pancreas among diabetic patients⁵³ (Supplementary Tables S26 and

S27). The levels of PRSS2 are highly correlated between the two platforms ($r=0.78$) and *cis* pQTLs are observed on both platforms.

In order to assess which *trans* pQTLs associating with disease are likely to correspond to an interaction between proteins in the same pathway, we assessed if each protein affected by a *trans* pQTL interacts with the protein encoded by the closest gene to the variant according to the STRING database⁵⁴. For about 9% of the *trans* pQTLs in high LD with a disease or trait associated variant, the two proteins of interest are known to interact. For instance, we note that a variant in TLR3 associates with autoimmune thyroid disease⁵⁵ (AITD) and levels of IFNL1 in *trans*, consistent with TLR3 being known to activate IFNL1⁵⁶. The *trans* pQTL with IFNL1 strongly supports the role of TLR3 at the variant locus in AITD pathogenesis.”

Furthermore, the thrust of the manuscript involves demonstrating how the choice of platform can impact the conclusions drawn from proteomics data. In this case, the novelty of the associations is secondary to the fact that the results differ between platforms.

6) Even for *cis*-pQTLs and given the fairly large coverage by the Olink Explore 3072 platform, it remains to be determined how often *cis*-pQTLs are indeed uniquely identifying causal genes. For example, variants like rs10774624 that are annotated as *cis*-pQTL for a number of proteins (e.g., BRAP, PPP1CC, or ERP29)?

To address the reviewer’s point, we have added to Supplementary Tables S10, S11 and S12 columns indicating for each *cis* pQTL to how many proteins it associates in *cis*. While 98% of pQTLs detected on both Olink and SomaScan only associate with one protein in *cis*, 68 pQTLs on one of the two platforms associate in *cis* with more than one protein. In most of the 68 instances (34 on Olink and 34 on SomaScan) where pQTL associated with more than one protein in *cis* we observe that this pQTL did associate with less than 25 proteins in *cis* or *trans*. In contrast, two pQTLs on Olink (including the pQTL associating with BRAP, PPP1CC, ERP29 and 765 other proteins proteome wide) and five on SomaScan are highly pleiotropic when considering *cis* and *trans* associations, and could therefore be interpreted in the same manner as *trans* pQTLs, in that their proximity to the gene may well be coincidental. Thus in the rare instances of highly pleiotropic variants, their association to protein in *cis* is difficult to interpret.

Our data are consistent with some variant influence Co-regulation of expression in *cis* previously observed at RNA level (Larson et al. 2015, PMID 25983244).

We have added the following to manuscript (pg. 10, l. 257-265):

“In both the UKB-BI Olink data and Icelandic SomaScan data, over 98% of the *cis* pQTLs only associate with one protein in *cis*. However, 34 pQTLs detected in the Olink data had multiple *cis* associations. Of these, 32 associated with less than 25 proteins in *cis* or *trans*, mostly two or three, while two associated with more than 25 (768 and 388). On SomaScan, 34 pQTLs had multiple *cis* associations. Of these, 29 associated with less than 25 proteins in *cis* or *trans* while five associated with more than 25 (ranging from 40 to 786). Co-regulation of expression in *cis* has been observed at RNA level and does not in itself detract from the *cis* pQTL as evidence for the performance of the assay²⁶. However, the *cis* location of a highly pleiotropic variant could be a matter of chance and should therefore be considered in the same way as a *trans* pQTL.”

7) The authors should make the code available, including how power calculations have been performed.

We provide in the Methods section thorough documentation of the computations we have performed. As our code makes extensive use of inhouse infrastructure, we believe that this is more useful and transparent than making the code available without the required infrastructure to run it.

Minor

- line 115 what is PN? The very first section is very long and might be better shortened to a section of not more than a small paragraph.

We have changed this to refer to ‘sample’ and thank the reviewer for pointing this mistake out.

As the results from this section adds information to prior work (e.g., by Katz et al.), we believe it is worth some effort to discuss the difference in methodology to explain the difference in results. The section on inter-class correlation were added at the request of another reviewer.

- Line 136: Does the very strong median within platform correlation for SomaLogic imply that w/o SMP normalization, the data should rather not be used, given that there is a strong underlying latent factor?

The high median correlation between protein levels on the SomaScan platform without SMP normalization probably reflects true biological variation (e.g., because of cell counts, BMI, liver function, etc.) that the SMP normalization adjusts out. Furthermore, as has been previously described, the SomaScan measurements show stronger correlation with Olink measurements without SMP normalization than with SMP normalization (Pietzner 2021, PMID: 34819519).

As discussed in Supplementary Note S3, the SMP normalization proposed by SomaScan affects the downstream analysis. It affects the results of GWAS analysis, both by increasing the power to detect pQTLs, and by occasionally affecting their direction, as in the case of ASGR1. It affects the association of protein levels to phenotypes, for instance by vastly reducing the association with BMI.

Therefore, neither method is clearly superior and they can lead to different results and interpretations.

- The section on tissue-based correlations should be omitted, since it is not clear whether the proteins indeed originate from each (e.g., testis in women)

The section on tissue-based correlation was added at the request of another reviewer. As implied in lines 165-166, the proteins are classified based on enriched expression as annotated in the Human Protein Atlas (HPA). HPA notes that while “enriched protein expression” is defined as being at least four-fold compared to any other tissue, very few proteins are strictly tissue specific. Therefore, the tissue of enhanced expression cannot be assumed to be the source of the protein in blood, nor is it claimed to be.

- The head-to-head comparison of sample age between Olink and SomaScan is misleading, since they originate from different cohorts as far as I understand.

The analysis of sample age on protein levels measured using the Olink and SomaScan platforms was added at the request of another reviewer. Previous research has demonstrated that sample age does affect plasma protein levels (PMID: 27596149). We have rephrased this paragraph to reduce the emphasis on the comparison between the platforms, and added a sentence noting that the differences in the distribution of sample age may affect the number of associations reaching the significance threshold.

The section now reads as follows (pg. 7, l. 174-179):

“Plasma protein levels have previously been shown to correlate with sample age¹⁷. Sample age, time from blood draw to measurement, correlated significantly with a considerable number of assays on both platforms, although differences in the distribution of sample age between cohorts make direct comparison difficult (Supplementary Tables S5 and S6, Extended Data Table 9). The effects were generally small for both platforms and the correlation between effects was low (Spearman $r=0.08$).”

- Why are observational associations stratified by ethnicity? I get the reason for genetic analysis but would assume that one would need to provide evidence of an interaction effect to justify splitting groups. Also how were the analysis in UKB adjusted for the sampling character, that is, random vs consortia selected cases?

There are two reasons why we choose to perform the phenotype associations stratified by ancestry.

First, as this is the first large-scale plasma proteomics study on this type of cohort, with three fairly distinct sub-populations by ancestry, we cannot rule out the presence of interaction effect with the ancestry. Indeed, we know of several sequence variants with large effects that are ancestry specific. Performing the analysis stratified by ancestry allows the reader to determine the extent of such interaction effect for themselves.

Secondly, as the sample size in UKB-BI is such that statistical power is not an issue, performing the association stratified by ancestry allows a closer comparison of effect sizes with those in the Icelandic cohort measured using SomaScan.

Performing the analysis in individual strata allows combination in meta-analyses and assessment of heterogeneity which would not be possible otherwise. Furthermore, due to the difference in size, the analysis would be dominated by the UKB-BI ancestry group if all the ancestry groups were analyzed together.

All analysis in UKB was done on the randomly selected samples. This information has been added to the Methods section (pg. 25, l. 696-698):

“A large majority of the samples were randomly selected across the UK Biobank, and only those were used for the analysis presented here.”

- The section “Protein-phenotype associations” is very long with little information overall and written in a very repetitive manner.

In the interest of clarity, the presentation of all protein-phenotype associations and pQTL-disease colocation examples follows the same structure. However, should the need arise in the editorial process, we are open to rewriting those sections to minimize repetition.

- What explains the drop in the number of pQTLs discovered with Olink Explore and Expansion?

Within both sets of proteins targeted with Olink Explore 1536 and Expansion, non-secreted proteins and proteins expected to have low abundance in plasma are less likely to have pQTLs detected in plasma than others. As the 1536 set is enriched for secreted and abundant proteins compared to the Expansion set, this results in a lower fraction of proteins targeted on the Expansion set having a *cis* pQTL than of those targeted on the 1536 set.

We have restructured the paragraph in lines 602-611 to make this clearer. The paragraph now reads (pg. 22, l. 616-625):

“A greater fraction of the Olink Expansion set of assays targets intracellular and less abundant proteins (undiluted) than of the 1536 set. Both these characteristics are associated with lower fraction of assays having a *cis* pQTL within each set of assays. Consequently, we observe a lower fraction of proteins with *cis* pQTLs for proteins targeted in the Expansion set of assays than by the initial 1536 set (56% vs 85%). Similarly, proteins targeted on a previous generation of the SomaScan platform (SomaScan 1.2K, Supplementary Table S2) are more likely to have a pQTL in our data than proteins targeted on the SomaScan v4 platform and not the older version (40% vs 56%). Given the enrichment of secreted and abundant proteins in the collection of proteins currently targeted on the platforms compared with HPA, we expect proteins currently targeted by neither of the two platforms to be even more challenging to measure in plasma.”

- Line 319 onwards: the conclusion for MS does not hold, since the authors do not establish colocalization of the MS signal in people of African American descent

Following the reviewer comment, we have changed the presentation to put forward the refinement of the pQTL signal. The text now reads (pg. 12, l. 338-341):

“Since the pQTL in UKB-BI associates with multiple sclerosis, the refinement allowed in UKB-AF of the pQTL signal indicates the potential gain from investigating disease correlation with the same variant in a population of African origin (Figure 3c). This would require proteomics measurements of MS samples from individuals of African ancestry.”

- Line 349: Does this imply that SomaScan is the more reliable technology?

Lines 349-351 state that pQTL replication succeeded in a higher fraction of cases using Olink than SomaScan. We have tried to rephrase this sentence to make it clearer. The sentence now reads (pg. 15, l. 415-416):

“Thus, the replication rate using Olink for *cis* pQTLs detected using SomaScan was somewhat higher than the replication rate using SomaScan for *cis* pQTLs detected using Olink.”

While we have deemphasized replication in this version of the manuscript, these results do not imply that SomaScan is more reliable.

- Line 362: I do not understand this sentence. Why is the effect of the alternative allele for a *cis*-pQTL close to zero when it is not in LD with a PAV? Is this because true effects of the alternative allele cancel out across proteins having positive and negative effects? Please clarify.

In general, if a variant is truly associated with the expression of a protein, it should be equally likely to associate with increased levels of the protein as with decreased levels. Therefore, the average effect should be close to zero. Conversely, if the association is due to epitope effects in the measurement caused by the variant, it would be more likely that the alternate allele correlates with lower measured levels of the protein than higher.

We have added the following clarification to the manuscript (pg. 14, l. 378-381):

“In general, a *cis* pQTL would be expected to be equally likely correlate with increased levels of a protein as decreased. Therefore, the average effect of *cis* pQTLs would be expected to be close to zero. However, if the pQTL is due to epitope effects, the alternate allele would be more likely to associate with decreased levels of the protein as measured by the assay.”

- Line 532: The conclusion that the absence of a cis-pQTL indicates that the protein is not well measured in plasma is not quite right. I might be wrong but peptide hormones such as insulin do not have a cis-pQTL either and we have no doubt that they are actively secreted and well-measured.

Following the reviewer's suggestion, the relevant sentence has been rewritten to emphasize that not all proteins are expected to have a *cis* pQTL. The sentence now reads (pg. 21, l. 600-602):

"While not all proteins are expected to have a *cis* pQTL in plasma, the large fraction of proteins with no *cis* pQTLs, around half, suggests that not all proteins can easily be measured in plasma."

- Line 597: This section is somewhat redundant to the introduction.

Following the reviewer's comment, to minimize redundancy, we have moved the sentences in the Discussions section detailing the differences in operating principles between the two platforms to the Methods section, where the differences were already outlined to some extent.

- Line 772 Why is the testing threshold adjusted to a different number of variants compared to what is shown in line 694?

For comparison with our previous study using the Icelandic SomaScan data, the threshold for significance was chosen to be the same as in that study. The number of variants tested in the data sets and the significance threshold are both described in the Methods section. The paragraph describing the effects of the different multiple-testing adjustments on the pQTL detection has been simplified (pg. 11, l. 280-282):

Correcting for the number of variants tested in UKB (57.7 million) would result in a multiple testing correction threshold in UKB of 8.7×10^{-10} instead of 1.8×10^{-9} . A total of 153 (1%) of the *cis* pQTLs have significance between those two thresholds and 1,608 (5%) of the *trans* pQTLs.

The paragraph containing line 772 has been rewritten to reflect this and now reads (pg. 30, l. 851-852):

"We computed *p*-values using a likelihood-ratio test and adjusted for multiple testing by using the same significance threshold (1.8×10^{-9}) as in our previous study on the Icelandic data set²."

- Prior settings for COLOC should be given, possibly adapting what has been recently proposed (Wallace 2020 Plos Genetics)

Following the reviewer's suggestion, the prior settings for COLOC have been added to the relevant Methods section.

The prior probabilities used for COLOC were the software defaults. As the posterior probabilities for shared signals were much higher than for independent signals in all the reported cases, the proposed choices of prior are unlikely to change the conclusions.

Figures:

- Fig 4 adds little and should go into the Supplement.

The purpose of Figure 4 is to emphasize that the presence of evidence for the correct targeting of an assay is highly dependent on several factors, including dilution group (representing expected abundance in plasma) and subcellular location (which also relates to the expected abundance in plasma). We feel that this is one of the key messages of the paper and believe the figure serves the purpose of underscoring this message.

- Fig 5 is very low resolution and hard to read, colouring bins to indicate spearman correlations would work better than a black to red gradient.

Following the reviewer's recommendations, we have regenerated Figure 5 with larger text and different colouring scheme.

- Why is a prettier version of the top panel of EDF 7 not in the main text?

As the primary focus of the manuscript is comparison between the Olink and SomaScan proteomics platforms, the main purpose of Extended Data Figure 7 is to allow the reader to contrast the observations between the two platforms. Therefore, displaying only one of the panels in Extended Data Figure 7 would not serve the purpose of the paper. While such graphical comparison can be helpful, we

believe that the figures included in the main text serve the purpose of the manuscript better than including all of EDF7.

In addition, a manuscript focusing primarily on the detection of pQTLs in the UKB data set is under review as well. This manuscript will almost certainly include a version of the UKB panel of Extended Data Figure 7 in the main text.

- The colour gradient in stacked locus zoom plots is hard to distinguish due to the thick frame around dots, pls omit this.

We thank the reviewer for pointing this out. The thick lines around the points on the locus zoom plots are a PDF conversion issue which we will make sure does not affect the final version of the manuscript.

Misc

- Extended table 9 is not well formatted (age all lower case) and bias 'health' / 'disease' is not really a description of a population; also what does 'delayed' sample handling in UKB imply?

We thank the reviewer for pointing out that the bias towards healthy donors and diseased donors is a factor of the cohort, not the population, and have changed the legend accordingly. Following the reviewer suggestion, we have also reformatted Extended Data Table 9 to include capitalization of rows and columns.

“Delayed” sample handling refers to the delay between blood draw and the aliquoting of the sample. After centrifuging, UKB samples are kept at 4°C until they are shipped to processing center for aliquoting, preferably within 24h. Icelandic samples are aliquoted immediately following centrifuging. This information has been added to the table legend.

Reviewer Reports on the Second Revision:

Referees' comments:

Referee #1 (Remarks to the Author):

The authors have addressed all of my concerns.

Referee #3 (Remarks to the Author):

Reviewer Comments

The revised version of the paper has clearly improved, and I am grateful to the authors for incorporating my comments and suggestions. The current version starts demonstrating the power, but also hurdles to turn pQTLs into knowledge for human biology and diseases. I only have some comments left that will hopefully help to further increase the accessibility of the paper, most importantly to move the emphasis from a merely technical report towards the joy of biological insights that can be gained from such technologies.

Rebuttal:

- 1) I appreciate that the scale and breadth of the pQTLs compared between both technologies is unprecedented, but the claim of the authors, that they are the first to report differential cis-pQTLs is not true. Pietzner et al. 2021 Nat Comms already demonstrated not only differential cis-pQTLs between both platforms, but also highlighted examples with different phenotypic consequences. The authors might want to acknowledge this observation, even if done at much larger scale and that the current study goes way beyond these early findings.
- 2) The comment on my request to make code public is simply not acceptable, since the underlying data is in principle available to a large community, which would be able to replicate and expand upon findings. This is of relevance given the rather low replication rate compared to the UKB-PPP paper presented in previous version of the manuscript.
- 3) I don't think that the answer about the intercorrelation of SL proteins is quite right. The authors report an intercorrelation among all assays of 0.08 for Olink, which is supposedly different from the 0.42 reported by SL w/o SMP normalization. At least to, this would suggest that some level of normalization is certainly needed to work with SL data. However, it is currently completely unclear how this may look like, and the authors may want to acknowledge this lack of knowledge in the limitation section of the paper.

Paper

- 1) Line: 252-256: This is an important observation not highlighted by previous studies, and maybe the authors can add one or two brief sentences, that explains those findings and whether the field should consider dropping pleiotropic, non-specific variants from any reporting.
- 2) Some sections would still benefit from more concise writing. For example, lines 277-284 might be simply summarized with "Down sampling and accounting for the different number of analysed SNPs did not affect conclusions drawn..." along with a reference to the Supplemental Methods. This is only one example, but the current draft offers more such potential to present core messages and analysis in a more concise manner.
- 3) Lines 391 – 394: While I agree with the overall impact of PAVs and the potential of eQTLs to support absence of epitope effects, I do not think that the opposite holds true. Why would one expect a PAV to have an effect on mRNA abundance, while it may well have a strong effect on translation, folding, secretion, etc. for the protein. It might be better to rephrase this conclusion. Same applies to Lines 597-599.
- 4) Line 395-411: I appreciate that discussing reciprocal replication of cis-pQTLs is the most

important part, but did the authors also check, in particular given the massive sample size, if for protein targets with at least some level of agreement, also trans-pQTLs did line up, or if there were even examples where cis-pQTLs do not line up, but almost all trans-pQTLs for the same protein target? Previous studies did already show differential cis-pQTLs, including with distinct phenotypic consequences, but the authors are here uniquely suited to address this important question, this would also help to lift the USP of the project in terms of platform comparison. However, the overall message from this section is really powerful, with major relevance for the use of cis-pQTLs and the results might even be featured in the abstract.

5) I missed the description of how fine-mapping has been performed.

6) The authors might want to also acknowledge in the limitations, that we still do not know which assay to believe if results are discordant. While the authors provide important criteria for cis-pQTLs, discrepancies for observational associations are left undiscussed. The same applies to the massive influence of normalization on the SomaLogic results.

Minor

1) Lines 297-299: This statement might benefit from rephrasing, since cis-pQTLs are also a strong source of measurement bias if they change epitope regions, potentially rendering the assay measurements useless among homozygous carriers of the alternative allele.

2) Line 458: Pls replace 'disease' with 'trait' or similar, since not all of the linked outcomes are diseases.

3) Line 546-552: This is effectively rephrasing MR assumptions and should be stated as such.

Author Rebuttals to Second Revision:

Referee #3 (Remarks to the Author):

Reviewer Comments

The revised version of the paper has clearly improved, and I am grateful to the authors for incorporating many comments and suggestions. The current version starts demonstrating the power, but also hurdles to turn pQTLs into knowledge for human biology and diseases. I only have some comments left that will hopefully help to further increase the accessibility of the paper, most importantly to move the emphasis from a merely technical report towards the joy of biological insights that can be gained from such technologies.

Rebuttal:

1) I appreciate that the scale and breadth of the pQTLs compared between both technologies is unprecedented, but the claim of the authors, that they are the first to report differential cis-pQTLs is not true. Pietzner et al. 2021 Nat Comms already demonstrated not only differential cis-pQTLs between both platforms, but also highlighted examples with different phenotypic consequences. The authors might want to acknowledge this observation, even if done at much larger scale and that the current study goes way beyond these early findings.

We agree that other studies have pointed out discordant pQTLs between the two platforms. The Pietzner et al. 2021 Nat Comms paper is referenced in the context of the platforms targeting different proteoforms in the discussion chapter (I517-519, p20):

“The difference in pQTLs between the two platforms is consistent with proteoforms being differentially targeted by the platforms, as suggested by previous work on smaller sample sizes, both in terms of individuals and proteins⁷.”

However, using the presence or absence of pQTLs to quantify the performance of the platforms is only made possible through the existence of sufficiently large data sets from both platforms.

2) The comment on my request to make code public is simply not acceptable, since the underlying data is in principle available to a large community, which would be able to replicate and expand upon findings. This is of relevance given the rather low replication rate compared to the UKB-PPP paper presented in previous version of the manuscript.

The association analysis is done using a combination of open-source tools and has been extensively described elsewhere (e.g., Halldorsson 2021, Ferkingstad 2021). The code for the power analysis and

computation of CV ratios will be uploaded to github at
http://github.com/DecodeGenetics/proteomics_comparison

3) I don't think that the answer about the intercorrelation of SL proteins is quite right. The authors report an intercorrelation among all assays of 0.08 for Olink, which is supposedly different from the 0.42 reported by SL w/o SMP normalization. At least to, this would suggest that some level of normalization is certainly needed to work with SL data. However, it is currently completely unclear how this may look like, and the authors may want to acknowledge this lack of knowledge in the limitation section of the paper.

Following the reviewer's suggestion, we have added the following to the 'Limitations' section (I556-559):

"The SMP normalization of the SomaScan data has a considerable impact on downstream analysis, as reflected by lower correlation with Olink measurements, higher fraction of assays with *cis* pQTLs, and differences in associations (Supplementary Note SN4). The full extent of the effects of the SMP normalization warrants further study."

Paper

1) Line: 252-256: This is an important observation not highlighted by previous studies, and maybe the authors can add one or two brief sentences, that explains those findings and whether the field should consider dropping pleiotropic, non-specific variants from any reporting.

We have added the following to the manuscript (I209-211):

"When an assay has no *cis* pQTLs and only non-specific *trans* pQTLs, it is possible that the targeting or measurement are not accurate. We believe that non-specific pQTLs should be interpreted with caution."

2) Some sections would still benefit from more concise writing. For example, lines 277-284 might be simply summarized with "Down sampling and accounting for the different number of analysed SNPs did not affect conclusions drawn..." along with a reference to the Supplemental Methods. This is only one example, but the current draft offers more such potential to present core messages and analysis in a more concise manner.

We thank the reviewer for the suggestion. Significant parts of the manuscript have been moved to Supplementary Notes, including the one described above.

3) Lines 391 – 394: While I agree with the overall impact of PAVs and the potential of eQTLs to support absence of epitope effects, I do not think that the opposite holds true. Why would one expect a PAV to have an effect on mRNA abundance, while it may well have a strong effect on translation, folding, secretion, etc. for the protein. It might be better to rephrase this conclusion. Same applies to Lines 597-599.

We agree with the reviewer that the presence of a PAV makes it more likely that a *cis* pQTL is due to an epitope effect, thus potentially rendering the protein level measurement unreliable except stratified on genotype. We have tried to rephrase the lines quoted to make this clearer. In lines 391-394 we have added a caveat to the conclusion, which now reads (l318-321):

“We conclude that the 23% of sentinel Olink *cis* pQTLs and 24% of sentinel SomaScan *cis* pQTLs that are in high LD with a moderate impact PAV but not with a *cis* eQTL are likely to be caused by epitope effects and may not in fact reflect variation in protein levels.”

In lines 597-599 we have replaced the phrase ‘the platform is measuring the designated protein’ with ‘the assay is binding to the designated protein’ to make the distinction clearer.

4) Line 395-411: I appreciate that discussing reciprocal replication of cis-pQTLs is the most important part, but did the authors also check, in particular given the massive sample size, if for protein targets with at least some level of agreement, also trans-pQTLs did line up, or if there were even examples where cis-pQTLs do not line up, but almost all trans-pQTLs for the same protein target? Previous studies did already show differential cis-pQTLs, including with distinct phenotypic consequences, but the authors are here uniquely suited to address this important question, this would also help to lift the USP of the project in terms of platform comparison. However, the overall message from this section is really powerful, with major relevance for the use of cis-pQTLs and the results might even be featured in the abstract.

We have added to our existing discussion of correspondence of *trans* pQTL associations between the Icelandic SomaScan data and the UKB-BI Olink data by subdividing the associations according to whether the protein has *cis* pQTLs on neither, only one, or both platforms, noting that on either platform a much larger proportion of *trans* signals have corresponding primary *trans* pQTLs on the other platform when *cis* pQTLs are present on both platforms. We have added the following sentence to the relevant section (l.339-341):

“Proteins having a *cis* pQTL on both platforms were more likely to have corresponding sentinel *trans* pQTLs (Supplementary Table ST28, Supplementary Note SN13).”

Supplementary note SN13 reads:

“SN13A. Replication of trans pQTLs conditional on cis pQTLs

For pQTLs detected in UKB-BI, subdividing signals into proteins having *cis* pQTLs on neither, only one, or both platforms resulted in 3%, 2%, and 13%, respectively, of signals having a sentinel corresponding *trans* pQTL in the Icelandic SomaScan data. Similarly, for pQTLs detected in the Icelandic SomaScan data, subdividing into proteins having *cis* pQTLs on neither, only one, or both platforms resulted in 3%, 4%, and 25%, respectively, of signals having a sentinel corresponding *trans* pQTL in the UKB-BI Olink data (Supplementary Table ST28).”

5) I missed the description of how fine-mapping has been performed.

We thank the reviewer for pointing out this mistake in terminology. We have removed the term ‘fine-mapping’ from the text (l. 307) to avoid confusion, instead referring to the refinement of the association signal as in other places in the manuscript.

6) The authors might want to also acknowledge in the limitations, that we still do not know which assay to believe if results are discordant. While the authors provide important criteria for cis-pQTLs, discrepancies for observational associations are left undiscussed. The same applies to the massive influence of normalization on the SomaLogic results.

Following the reviewer’s suggestion, the following sentence has been added to the ‘Limitations’ section (l562-564):

“Where the results from the platforms are discordant, further studies are required to determine which platform to believe, although evidence such as orthogonal validation or the existence of *cis* pQTLs can provide some insight.”

Minor

1) Lines 297-299: This statement might benefit from rephrasing, since cis-pQTLs are also a strong source of measurement bias if they change epitope regions, potentially rendering the assay measurements useless among homozygous carriers of the alternative allele.

We have added the caveat that *cis* pQTLs may be due to epitope effects rather than actual protein levels. The beginning of the section now reads (l237-240):

“While secondary signals help to understand how genetic variation affects protein expression, the mere existence of a *cis* pQTL for a protein on a particular platform provides evidence that the assay is binding to the correct protein, even though the pQTL may in fact be an effect of the sequence variation and not reflect actual variation in protein levels.”

2) Line 458: Pls replace ‘disease’ with ‘trait’ or similar, since not all of the linked outcomes are diseases.

We thank the reviewer for pointing this out and have replaced ‘disease’ with ‘disease or trait’.

3) Line 546-552: This is effectively rephrasing MR assumptions and should be stated as such.

We have modified the section to highlight the similarities to Mendelian randomization. The beginning of the section now reads (l447-449):

“Similar to the logic underlying Mendelian randomization, when all variants associating significantly with the levels of a given protein also associate proportionally with the risk of a particular disease, we propose that the protein plays a role in the pathogenesis of the disease⁴⁷.”